# Satellite observations of aerosols and clouds over southern China from 2006 to 2015: analysis of changes and possible interaction mechanisms

Nikos Benas[1], Jan Fokke Meirink[1], Karl-Göran Karlsson[2], Martin Stengel[3], Piet Stammes[1]

[1]Royal Netherlands Meteorological Institute (KNMI), De Bilt, the Netherlands

[2]Swedish Meteorological and Hydrological Institute (SMHI), Norrköping, Sweden

[3]Deutscher Wetterdienst (DWD), Offenbach, Germany

*Correspondence to*: Nikos Benas (benas@knmi.nl)

**Abstract.** Aerosol and cloud properties over southern China during the 10-year period 2006-2015 are analysed based on observations from passive and active satellite sensors and emission data. The results show a strong decrease in aerosol optical depth (AOD) over the study area, accompanied by an increase in liquid cloud cover and cloud liquid water path (LWP). The most significant changes occurred mainly in late autumn and early winter months: AOD decreased by about 35%, coinciding with an increase in liquid cloud fraction by 40% and a near-doubling of LWP in November and December. Analysis of emissions suggests that decreases in carbonaceous aerosol emissions from biomass burning activities were responsible for part of the AOD decrease, while inventories of other, anthropogenic emissions mainly showed increases. Analysis of precipitation changes suggests that an increase in precipitation also contributed to the overall aerosol reduction. Possible explanatory mechanisms for these changes were examined, including changes in circulation patterns and aerosol-cloud interactions. Further analysis of changes in aerosol vertical profiles demonstrates a consistency of the observed aerosol and cloud changes with the aerosol semi-direct effect, which depends on their relative heights: fewer absorbing aerosols in the cloud layer would lead to an overall decrease in evaporation of cloud droplets, thus increasing cloud LWP and cover. While this mechanism cannot be proven based on the present observation-based analysis, these are indeed the signs of the reported changes.

## 1 Introduction

The role of atmospheric aerosols in climate change has been studied widely in the past. Their various effects are broadly defined based on their interactions with atmospheric radiation and clouds. The direct effect is described through scattering and absorption of radiation whereas indirect effects describe interactions with clouds, which can lead to changes in both cloud albedo (Twomey, 1977) and cloud lifetime (Albrecht, 1989). The semi-direct effect is a third category that describes aerosol-induced changes in clouds through interaction with radiation. According to the latest terminology (Boucher et al., 2013), the semi-direct effect is described as a "rapid adjustment" induced by aerosol radiative effects, and along with the direct effect it is grouped into the "Aerosol-Radiation Interactions" (ARI) category, whereas the indirect effects are termed "Aerosol-Cloud Interactions" (ACI).

Observations of these mechanisms and their effects on climate have been elusive, and the uncertainties associated with them remain high (Boucher et al., 2013). The main reasons for this lack of substantial progress originate in the high complexity of these phenomena, with multiple possible feedback mechanisms and dependences on various parameters in different regimes (Stevens and Feingold, 2009, Bony et al., 2015). Although there are continuous improvements, the mechanisms related to aerosol and cloud interactions and feedbacks are still inadequately represented in models (Feingold et al., 2016), and poorly

captured by remote sensing measurements (Seinfeld et al., 2016). Regarding the latter approach, many studies have highlighted the difficulties and limitations of remote sensing methods, which usually include limitations in spatial and temporal samplings (Grandey & Stier, 2010; McComiskey & Feingold, 2012). On the other hand, progress is steadily being made, as data sets of aerosols and clouds based on remote sensing retrievals gradually improve. Additionally, independent data sets with complementary characteristics and properties become constantly available, allowing more in-depth analyses of the aerosol and cloud conditions and opening new possibilities for combined usage, towards further constraining the effects of aerosols on clouds.

The present study builds on these developments by providing an analysis of aerosol and cloud characteristics and changes in recent years over a climatically important and sensitive area in southern China. This region (20°-25° N, 105°-115° E) was selected, being a densely populated area with intense human activities, ranging from urban and industrial to agricultural, which also constitute different sources of aerosol emissions. Furthermore, significant changes in aerosol loads during the past years over the wider surroundings have previously been reported (e.g. Zhao et al., 2017; Sogacheva et al., 2018), providing the opportunity for an analysis of possible effects on clouds. Hence, the purpose of this study is dual. The primary aim is to analyse aerosol and cloud characteristics and changes during the period 2006-2015 over southern China. Using multiple data sets, created based on different retrieval approaches, adds robustness to the results. The secondary purpose of this study is to investigate the possibilities and limitations of the combined use of this multitude of aerosol and cloud data sets for the assessment of possible explanatory mechanisms, including large-scale changes and local-scale aerosol and cloud interactions. For this purpose, data sets are analysed in combination, to either help exclude possible explanations, or provide indications of their manifestation.

The study is structured as follows: Section 2 provides a description of the aerosol, emissions and cloud data sets used, and the methodology for analysing their changes. Results of this analysis include time series and seasonal changes in aerosols and clouds, presented in Section 3, and possible effects of large-scale meteorological variability and indications of possible effects of aerosol changes on corresponding cloud changes, described in Section 4. Our findings are summarized in Section 5.

## 2 Data and methodology

### 2.1 Aerosol, emissions and precipitation data

Analysis of aerosol changes was based on MODerate resolution Imaging Spectroradiometer (MODIS), Multi-angle Imaging SpectroRadiometer (MISR) and Cloud-Aerosol Lidar and Infrared Pathfinder Satellite Observations (CALIPSO) data. MODIS is a sensor on board NASA's Terra and Aqua polar orbiters, providing aerosol and cloud data products since 2000 and 2002 from Terra and Aqua, respectively. The Aqua MODIS level 3 Collection 6 daily Aerosol Optical Depth (AOD) was used here, available over both land and ocean at $1° × 1°$ spatial resolution (Levy et al., 2013).

AOD data from MISR were also analysed. MISR flies on board NASA's Terra satellite and acquires measurements at nine viewing angles, providing information on specific aerosol types along with the total aerosol load (Khan & Gaitley, 2015). Here, MISR products of total AOD, along with fine mode, coarse mode AOD and dust (non-spherical) particles AOD were analysed on a monthly basis and at $1° × 1°$ spatial resolution, available at level 3 of version V23.

The CALIPSO level 3 monthly aerosol profile product was also used, to include information on the aerosol vertical distribution in the analysis. CALIPSO level 3 parameters are derived from the corresponding instantaneous level 2 version 3 aerosol product (Winker et al., 2009; Omar et al., 2009; Tackett et al., 2018) and include column AOD of total aerosol, available globally at $2° × 5°$ latitude/longitude resolution, along with the extinction profiles at 60 m vertical resolution, up to 12 km altitude. The standard quality filters implemented to ensure the quality of the level 3 product, described in Tackett et al. (2018), were also adopted here.

Apart from the analysis of aerosol loads and vertical distributions over the region with MODIS, MISR and CALIPSO data, aerosol sources were investigated using the Global Fire Emissions Database (GFED), and the Copernicus Atmosphere Monitoring Service (CAMS) emissions inventories. GFED provides information on trace gas and aerosol emissions from fires on a global scale. Here, organic carbon (OC) and black carbon (BC) data were analyzed, based on the latest version 4 of

the data set (GFED4s). They are available at $0.25° \times 0.25°$ spatial resolution and on a monthly basis. GFED emission estimates are based on data of burned areas and active fires, land cover characteristics and plant productivity, and the use of a global biogeochemical model (Van der Werf et al., 2017). The CAMS global anthropogenic emissions inventory provides emission information for a multitude of species and sources, including transport, industry, power generation, waste handling and agriculture, also on a monthly basis and at $0.1° \times 0.1°$ (Granier et al. 2019). It should be noted that, due to the long-range

transport of aerosols, local aerosol emissions will not always fully explain corresponding properties and characteristics of aerosol types and loads in the atmosphere of the same region. The adequacy of local emissions in this role will depend on the effect that long-range transport may have, either removing aerosols from the region or adding more loads from adjacent regions. To assess the magnitude of these effects in the study region, an air mass trajectory analysis was also performed to supplement local emissions.

While emission data records provide a useful source of possible aerosol sources, decreases in aerosol loads can also originate in other phenomena, e.g. increase in precipitation, apart from decreasing emissions. Hence, to achieve a more complete overview of the possible reasons that led to the aerosol changes reported here, rainfall data were also analyzed. For this purpose, the Global Precipitation Climatology Project (GPCP) version 2.3 data record was used (Adler et al. 2018). The GPCP monthly product integrates precipitation estimations from various satellites over land and ocean with gauge

measurements over land at a $2.5° \times 2.5°$ resolution.

## 2.2 Cloud data

Two independently derived, satellite-based cloud data sets, were used for the analysis of cloud properties and changes over southern China. The Aqua MODIS level 3 Collection 6 daily $1° \times 1°$ product was used (Platnick et al., 2017), as in the case of AOD, for the estimation of monthly averages and corresponding changes in cloud properties, including total and liquid

Cloud Fractional Coverage (CFC), in-cloud and all-sky Liquid Water Path (LWP), as well as liquid Cloud Optical Thickness (COT) and Effective Radius (REFF).

The same cloud properties were analyzed using the second edition of the Satellite Application Facility on Climate Monitoring (CM SAF) cLoud, Albedo and surface RAdiation data set from AVHRR data (CLARA-A2), a recently released cloud property data record, created based on Advanced Very High Resolution Radiometer (AVHRR) measurements from

NOAA and MetOp satellites (Karlsson et al., 2017). It covers the period from 1982 to 2015 and includes, among other parameters, CFC and cloud phase (liquid/ice), cloud top properties and cloud optical properties, namely COT, REFF and water path, separately for liquid and ice clouds. Orbital drift in NOAA satellites is an important issue regarding the stability of the CLARA-A2 time series, especially in the 80s and 90s. For the 10-year period examined in this study, CLARA-A2 level 3 data, available at $0.25° \times 0.25°$ spatial resolution from AVHRR on NOAA-18 and NOAA-19 were used. Specifically,

only the "primary" satellite was used in each month, meaning that when NOAA-19 data became available, NOAA-18 was not used any more. As a result, orbital drifts are minor.

## 2.3 Uncertainties in aerosol and cloud products

Uncertainties in pixel-based (level 2) data can in many cases be estimated by propagation of error sources through the retrieval algorithms and through validation with collocated independent reference observations. For example, Levy et al.

(2013) showed by comparison with Aerosol Robotic Network (AERONET) observations that the MODIS AOD has a 1-sigma uncertainty of about $\pm(0.05+0.15AOD)$ over land. However, the propagation of pixel-based error estimates to monthly

aggregates is difficult because it needs to separate contributions from systematic and random errors. Similarly, validation at monthly scales is cumbersome, and no level-3 validation results have been reported for the data sets used in this study.

Therefore, the use of three independent aerosol data sets and two cloud data sets, derived from different sensors is an important element of this study, which suggests that the detected changes reflect actual changes, rather than possible sensor degradations or retrieval artifacts. This is especially true in the case of aerosol data, which were obtained by different retrieval approaches.

## 2.4 Analysis of time series and changes

The analysis of all data sets and their changes was based on monthly average values. This temporal resolution is appropriate for studying both long-term interannual as well as seasonal changes. Furthermore, data from afternoon satellites were mainly used (MODIS Aqua, AVHRR on NOAA-18 and -19 and the daytime product of CALIPSO), to minimize differences due to different temporal samplings. Additionally, due to the different grid cell sizes of the products used, the analysis was based only on area-weighted averaged values over the entire study region, rather than individual grid cells. Area-weighted averages were computed based on the cosines of the latitudes of the grid cells covering the study region. However, due to the small size of the domain the ensuing differences were minor. It should be noted that, in the case of the two emission data sets GFED and CAMS, monthly values of emissions over the study area were calculated by summing the corresponding grid cell values, instead of averaging. Additionally, in the case of CALIPSO, spatial averages were weighted by the number of samples used, which is available in the level 3 data.

The quantification of changes during the study period was based on linear regression fits to the spatially averaged deseasonalized monthly time series. Deseasonalization was performed by subtracting from each month the corresponding time series average of this month and then adding the average of all months in the time series. For every variable $X$ studied, the change $\Delta X$ was calculated as $\Delta X = X_f - X_i$, where $X_i$ and $X_f$ are the initial and final monthly values of the regression line. The corresponding percent change was estimated as $\Delta X = 100(X_f - X_i)/X_i$.

Spatial and temporal representativeness of the study area and time period in the change analysis were ensured by applying thresholds to both the area covered with valid data and the number of months used in the calculations. Specifically, the following thresholds were applied: a) on a grid cell basis, a monthly average value was used only if it was computed from at least 18 daily values (10 daily values for AOD, due to sparsity of data). Application of this threshold requires the number of days used in the calculation of the monthly average. This information was available in all data sets used, except for MISR; b) a spatially averaged value was used if it was computed from at least 50% of the grid cells in the study area; c) it was required that at least 80% of monthly averages are present in the time series, for the corresponding 10-year changes to be estimated. Further analysis included a per month estimation of changes, in order to assess their seasonal variation. In this case, no deseasonalization was applied. Statistical significance of all calculated changes was estimated using the two-sided t-test, with a confidence interval of 95%.

## 3 Results

### 3.1 Aerosol and emissions characteristics and changes

Figure 1 shows the seasonal variation of AOD from MODIS, MISR and CALIPSO over southern China, based on data during 2006-2015. MODIS, MISR and CALIPSO total AOD are in relatively good agreement in most months, with the largest differences occurring in March and April, when CALIPSO deviates from the other two data sets. While the present analysis was designed to minimize discrepancies due to differences in spatial and temporal resolutions, as described in Section 2.3, some disagreement between CALIPSO and the passive sensors should be expected, considering their differences in areas sampled, overpass times and retrieval methodologies. While it was not possible to pinpoint specific reasons for the

March-April differences based on the data sets used here, this feature deserves further investigation. Based on MISR, which offers additional information on aerosol types, the fine mode and coarse mode AODs, which add up to the total AOD, follow a seasonal pattern similar to the latter. The fine mode AOD, which constitutes a large part of the total, highlights the important role that anthropogenic emissions play in the overall aerosol load over the region. On the other hand, the contribution of dust is minimal, with a small peak in spring. This is probably due to the long distance of the study region from deserts, which constitute major dust sources.

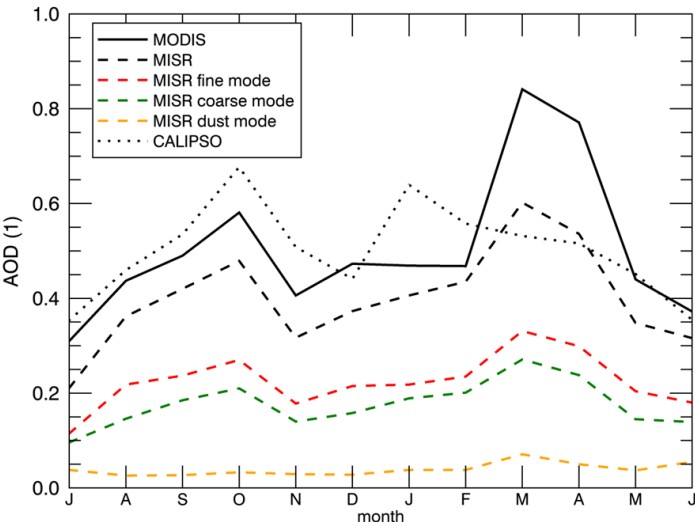

**Figure 1. Seasonal variation in Aerosol Optical Depth (AOD) over southern China, based on the period 2006-2015, from MODIS, MISR and CALIPSO, including MISR fine, coarse and dust mode AOD. Note that the horizontal axis starts in July and ends in June.**

Figure 2 shows the changes in AOD over the southern China region during the 10-year period examined, both on a grid cell basis from MODIS (Fig. 2a) and as spatially averaged time series from MODIS, CALIPSO and MISR (Figs. 2b, 2c and 2d). The grid cell-based changes in AOD (Fig. 2a) reveal an almost uniform reduction throughout the area, with stronger decreases over land. The time series of the deseasonalized spatially averaged monthly values of the AOD, separately from MODIS, CALIPSO and MISR, are shown in Figs. 2b, 2c and 2d, along with their linear regression fits and corresponding changes (in percent). The reduction in total AOD during the 10-year period is apparent and statistically significant in the 95% confidence interval in all three time series, and covers large parts of the study region (see also supplementary Figs. S1 and S2. Additional information on the time series analysis, i.e. slopes and p-values, is given in Table S3). The reduction in AOD reported here is in agreement with changes over the same region or wider Chinese regions during recent years, reported based on different satellite sensors, e.g. MODIS (He et al., 2016), MODIS and AATSR (Sogacheva et al., 2018) and MODIS and MISR (Zhao et al., 2017).

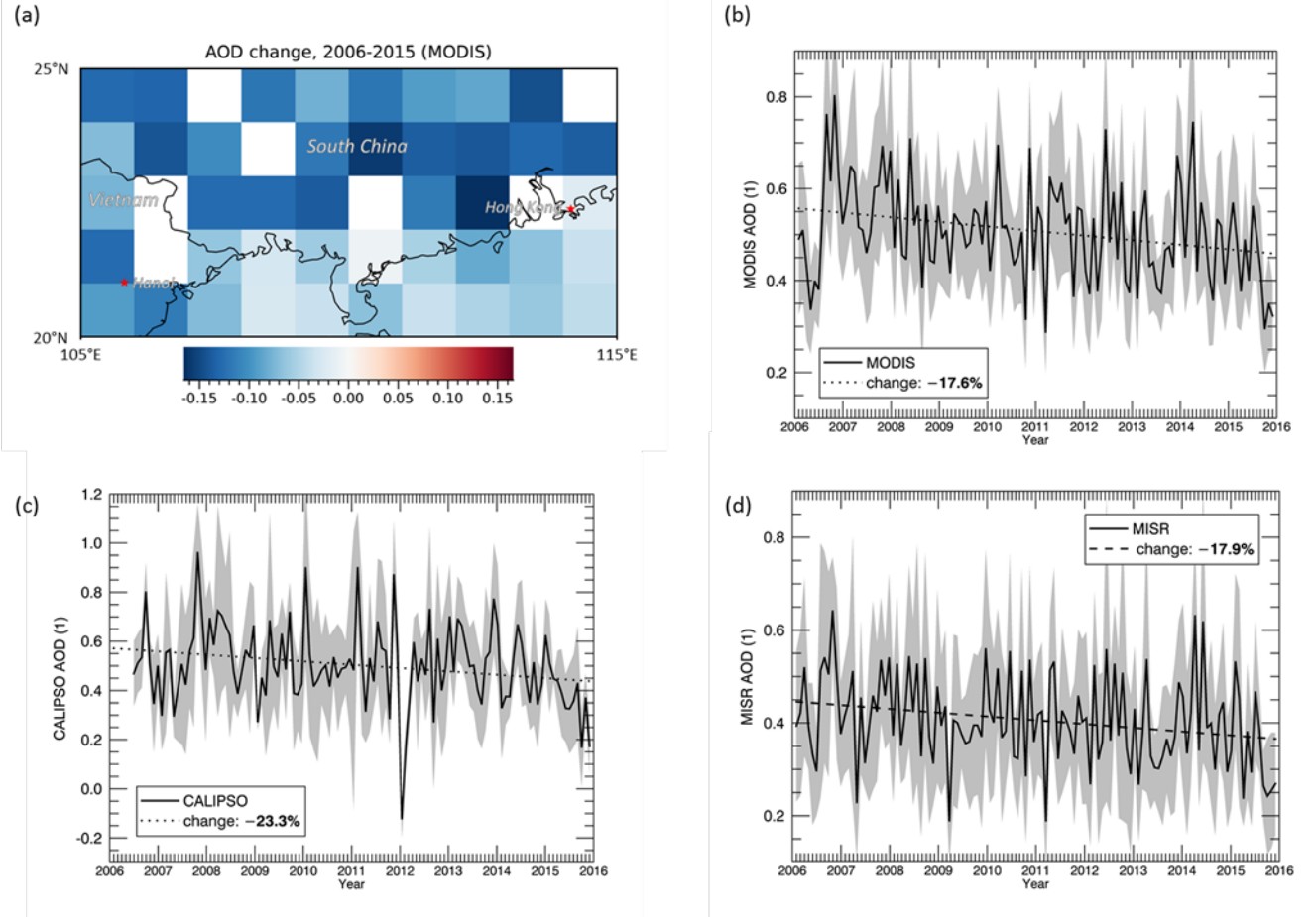

**Figure 2. Changes in AOD over southern China from 2006 to 2015. (a) Spatial distribution of AOD change over the study region deduced from MODIS data. Spatially averaged monthly deseasonalized values of AOD from MODIS (b), CALIPSO (c), and MISR (d). Shaded areas correspond to one standard deviation of the grid-scale monthly averages. Dotted lines correspond to linear regression fits. Percent changes during the period examined are also shown, with the statistically significant ones indicated in bold.**

The seasonal variability of aerosols over the study region (Fig. 1) suggests that their changes could also exhibit seasonal variations. Hence, the time series changes in AOD were further analyzed in terms of their seasonal variability. Results are shown in Fig. 3. It is apparent that the main decrease occurs in autumn and early winter. All three data sets agree well in this seasonal pattern. Based on MISR, this decrease is driven primarily by fine mode and secondarily by coarse mode aerosols, as reported earlier, while dust aerosols show no significant change.

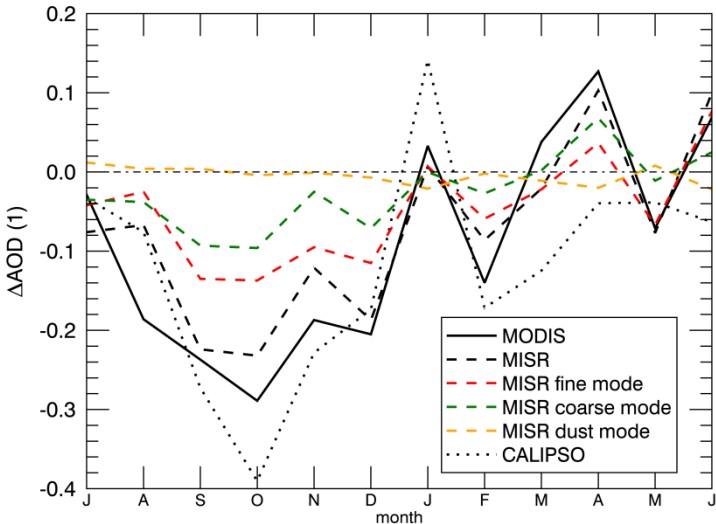

**Figure 3. Seasonal variation of changes in AOD over southern China from 2006 to 2015 deduced from MODIS, MISR and CALIPSO data. MISR data include fine, coarse and dust mode AOD.**

The same analyses of seasonal variability and changes was also performed for emission sources in the region, which could possibly explain part of the AOD characteristics. These include local fire emissions of organic carbon (OC) and black carbon (BC) particles from GFED, as well as anthropogenic emissions of OC, BC, $SO_2$, $NO_x$ and $NH_3$ from CAMS, the latter three acting as sulphate and nitrate aerosol precursor gases. Figure 4a shows that GFED emissions exhibit a strong seasonality, with the highest occurring between November and April. The emissions from CAMS exhibit almost no seasonal variation (supplementary Fig. S4), since there is no strong seasonality in the activities producing them, e.g. industrial emissions and transportation. On the other hand, biomass burning seasonality might be explained by activities which exhibit seasonal variation, such as crop residues burning, firewood consumption and agricultural waste open burnings (Chen et al., 2017). It should be noted, however, that the GFED emissions are limited to open fire events, and thus should not be regarded as representative of all biomass burning activities in the region.

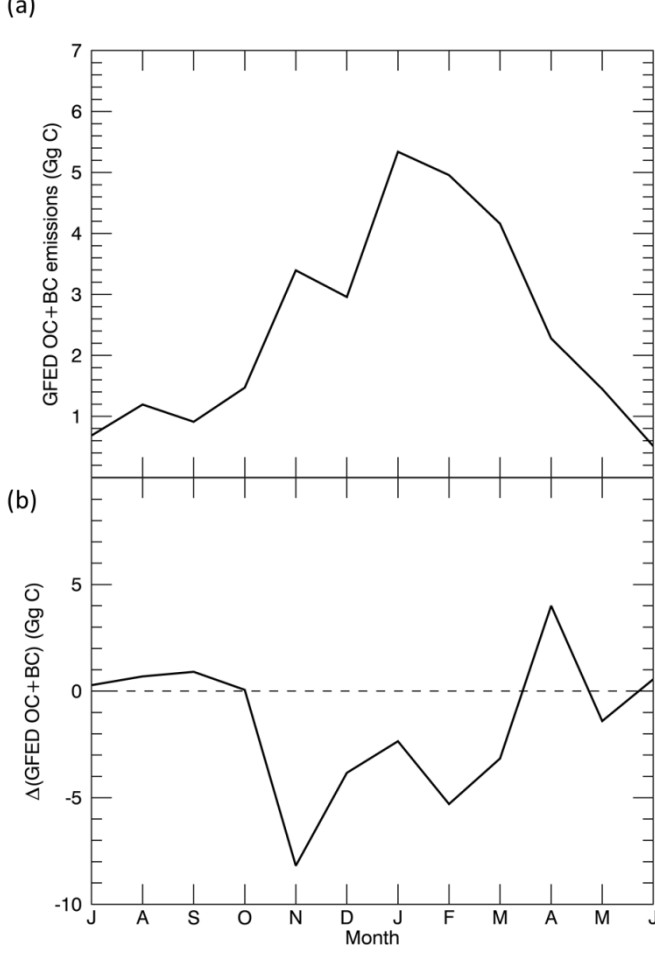

**Figure 4. (a) Seasonal variation in organic and black carbon emissions from GFED (in Gg C) over southern China from 2006 to 2015. (b) Corresponding changes on a monthly basis (in Gg C month$^{-1}$) during the same period.**

Analysis of changes in OC and BC emissions from GFED shows an overall decrease in emitted particles, with the largest occurring during late autumn to early spring (Fig. 4b), with a minimum in November. Analysis of other anthropogenic major emission sources in the region reveals increases during the period examined (Fig. 5). While these results may at first seem contradictory to the general consensus on the reduction of anthropogenic emissions in China during recent years (see e.g. Van der A et al. 2017), it should be noted that emission patterns are not uniform throughout China. Instead, differences should be expected on the provincial level, such as in this study. Furthermore, large scale implementations of emission policies in China in specific past years render similar analysis very sensitive to the time range selected.

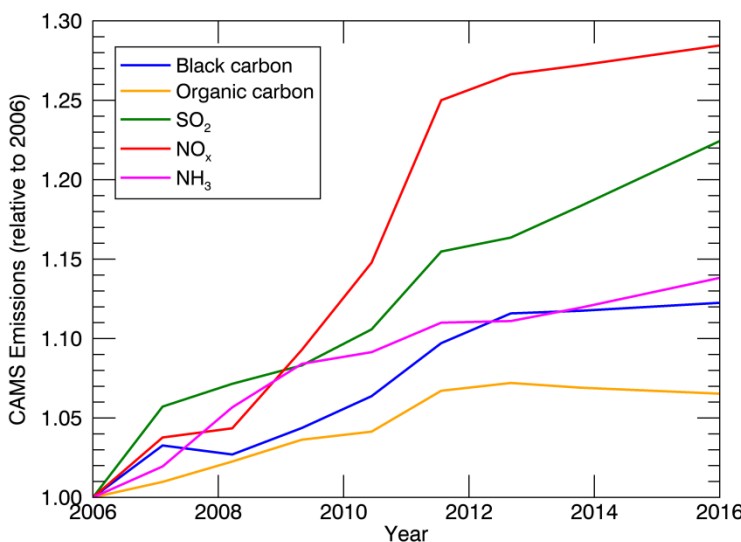

**Figure 5. Emissions of aerosols and precursor gases from the Copernicus Atmosphere Monitoring Service (CAMS). The emissions have been aggregated to annual totals over the southern China study area and plotted relative to the year 2006.**

A direct comparison of changes in AOD and GFED surface emissions, which are both satellite-based, offers additional insights into the origins of these changes: in cases where AOD and emissions changes agree well, such as in November and December, when both AOD and biomass burning decrease (Figs. 3 and 4b), it can be hypothesized that the former played a role in the latter. While this cause-and-effect mechanism cannot be proved based on observations only, this hypothesis can be further supported by examining correlations between the two data sets (this is done in Sect. 3.3). In cases where there is

an obvious disagreement between emissions and AOD changes (e.g. in September and October), additional reasons for the AOD reduction must be sought.

Two additional mechanisms that could explain the decrease in AOD in the absence of decreasing emissions were examined: precipitation and long-range transport. As mentioned in Sect. 2.1, changes in precipitation are also a factor that can lead to changes in aerosol concentrations. For this reason, a similar analysis of GPCP precipitation data was performed, with the

results shown in Fig. 6. The seasonality pattern (Fig. 6a) shows higher precipitation values appearing in summer months, compared to winter. Precipitation has overall increased by 11.2 % over the region during the study period (Fig. 6b), although not in a statistically significant sense. It should be noted, however, that increased precipitation in southern China during the same period is also reported elsewhere (see e.g. figure 3b in Zhang et al. 2019). Examination of monthly changes shows that this increase appeared mainly in autumn and early winter (September – December), largely coinciding with the decrease in

aerosols (Fig. 3), while a significant precipitation decrease occurred in June. Further correlation analysis showed that precipitation changes anti-correlate significantly with AOD changes from MODIS and MISR in September – December (see Sect. 3.3). These results suggest that wet removal played a role in the decrease in AOD reported for the same period.

A long-range transport analysis was also performed, since a change in AOD could also be caused by transportation of adjacent air masses to or from the study region. For this purpose, forward- and back-trajectory analyses were performed

using the Hybrid Single-Particle Lagrangian Integrated Trajectory (HYSPLIT) model. HYSPLIT is a public domain model (ready.arl.noaa.gov/HYSPLIT.php), suitable for analyzing air mass trajectories (Draxler and Hess, 1998). For the present study, the analysis setup was as follows: October and November were selected to be analyzed in terms of long-range transport. October is the month with the largest discrepancies between changes in AOD (large decrease, see Fig. 3) and local emissions (practically no change, see Fig. 4b), while in November large changes are reported in AOD (Fig. 3), biomass

burning emissions (Fig. 4b) and clouds (discussed later in Section 3.2). For October and November 2007 and 2015,

representing conditions close to the beginning and end of the study period respectively, HYSPLIT was run for trajectories starting every 6 hours and lasting 24 hours each, during the whole month, at two heights, 500 m and 1500 m above sea level. Both forward- and back-trajectories were analyzed, starting and ending at the center of the study region. The forward-trajectories analysis was performed to give insights into the degree of dispersion of locally-emitted aerosol loads outside of the study region, while the back-trajectory analysis reveals how frequently the air masses found in the study region originate outside of it.

The results, shown in supplementary Figs. S5 and S6 for the October forward- and back-trajectories respectively, and in Figs. S7 and S8 for the November cases, reveal that in all cases more than 90% of the forward trajectories end up within the study region and more than 90% of the back-trajectories originate inside the study region. In fact, these high-probability areas are much smaller than the study region, suggesting that even for points near the edge of the study region, instead of its center, the contributions from adjacent areas will be much lower than the local emissions.

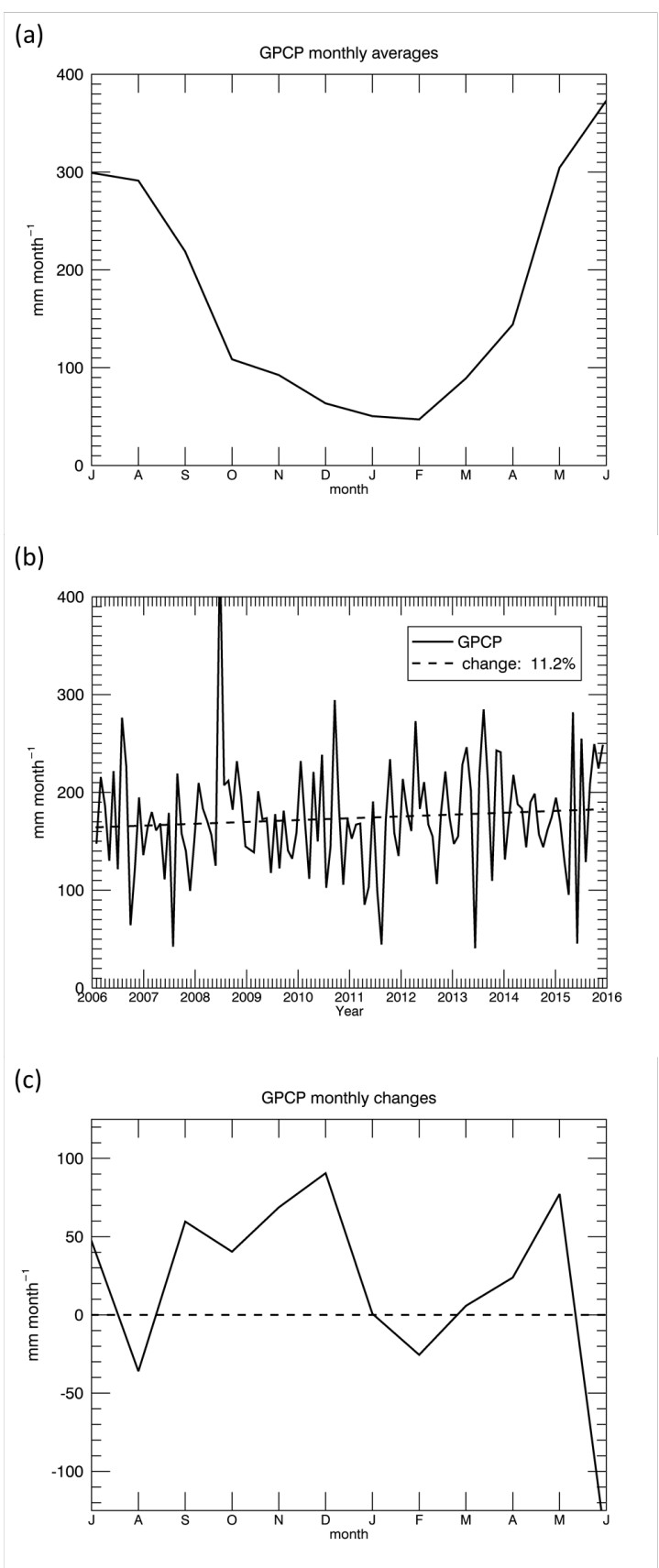

Figure 6. (a) Seasonal variation of precipitation in the southern China study region based on Global Precipitation Climatology Project (GPCP) data. (b) Corresponding spatially averaged monthly deseasonalized values. The dotted line corresponds to the linear regression fit. (c) Seasonal variation of changes in GPCP precipitation. Seasonal averages and changes in (a) and (c) are based on data from the period 2006-2015.

## 3.2 Cloud characteristics and changes

The seasonality of main cloud properties over the study region, comprising total and liquid cloud cover, and optical thickness and effective radius for liquid clouds, is shown in Fig. 7. While the total cloud cover does not exhibit strong seasonal characteristics (Fig. 7a), varying between 0.7 and 0.8 throughout the year (based on CLARA-A2 and MODIS, respectively), liquid clouds appear to prevail from late autumn to early spring (Fig. 7b). A similar seasonal pattern appears in liquid COT (Fig. 7c), which is not necessarily related to the variation in the extent of liquid clouds. Liquid REFF ranges between 10 μm and 14 μm throughout the year (Fig. 7d). The LWP, which is proportional to the product of liquid COT and REFF, also varies seasonally, with higher values in winter (not shown here). The main driving factor for the seasonality in total and liquid cloud cover is the Asian Monsoon (AM). The monsoon season in summer is characterized by a larger fraction of high clouds with ice near the top, in particular convective clouds. In winter, low stratus/stratocumulus clouds prevail. Overall, there are more clouds in summer compared to winter, but more liquid clouds in winter (Pan et al., 2015). The prevalence of low, liquid clouds in winter, which are mostly single-layer clouds, is also verified based on CALIPSO data (Cai et al., 2017). On the other hand, in summer higher ice clouds, constituting about half of the CFC, probably shield a considerable amount of low liquid clouds.

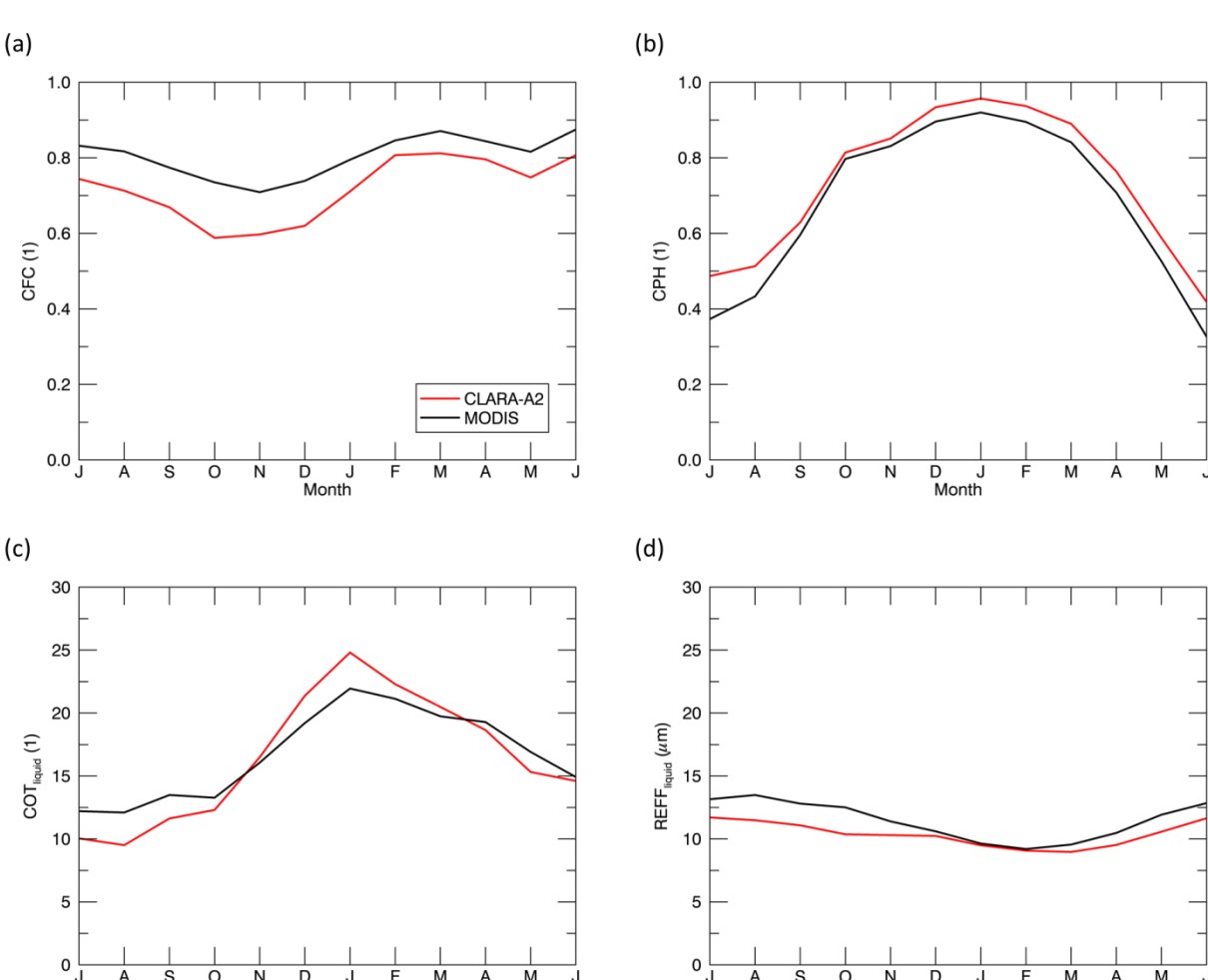

**Figure 7. Seasonal variations in cloud properties over southern China, based on CLARA-A2 and MODIS data, during the period 2006-2015. (a) Total CFC, (b) cloud phase (CPH; fraction of liquid clouds relative to total CFC), (c) COT for liquid clouds and (d) REFF for liquid clouds.**

Figure 8 shows grid cell based and spatially averaged changes in cloud properties over southern China during the period examined. The all-sky LWP and liquid CFC have increased over most parts of the land and significantly in most cases (Figs.

8a and 8b, with corresponding maps of statistical significance levels given in Fig. S9). In fact, Fig. 8 shows increases in all liquid cloud properties, with the largest increase found for the total liquid water content present in clouds (12%-14%). Liquid COT variations appear similar to those of LWP, with very good agreement between the two data sets (CLARA-A2 and MODIS). COT changes are positive, while liquid REFF changes are also positive but more ambiguous. Cloud changes appear statistically significant at the 95% level over large areas of the study region, especially over land, when studied on a grid cell basis. Analysis of spatially averaged values, however, over the entire (5° × 10°) study region, reduces this significance to levels below 95% in most cases of Fig. 8 (see also Table S3). Overall, MODIS and CLARA-A2 are in good agreement and consistent in terms of the changes reported, with biases of around 10% appearing for liquid CFC (Fig. 8d) and REFF (Fig. 8f). Supplementary Figs. S10 and S11 provide more details on spatial distributions and corresponding levels of significance for changes in liquid clouds from CLARA-A2 and MODIS.

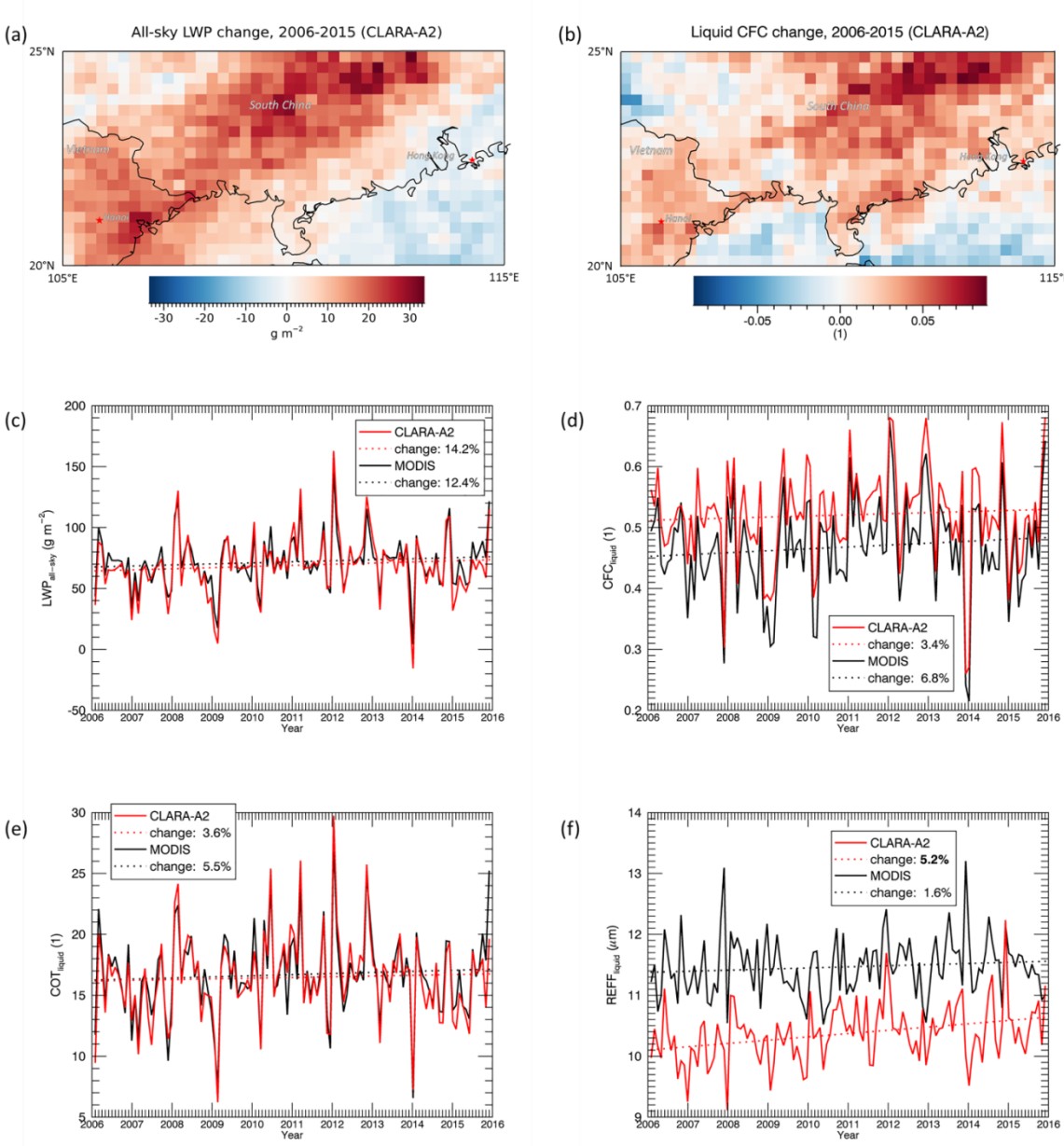

**Figure 8. Changes in liquid cloud properties over southern China from 2006 to 2015, based on CLARA-A2 and MODIS data. (a), (b) Spatial distributions of changes in all-sky LWP and liquid CFC based on CLARA-A2 data. Spatially averaged monthly deseasonalized values of all-sky LWP (c), liquid CFC (d), liquid COT (e) and REFF (f). Percent changes during the period examined are also shown, with the statistically significant ones (only CLARA-A2 liquid REFF) indicated in bold.**

The increase in all-sky LWP appears much larger than the increase in liquid CFC, suggesting an increase in cloud geometrical thickness and thus higher cloud tops. Therefore an additional analysis on Cloud Top Height (CTH) from CLARA-A2 and MODIS was performed. Results are presented in supplementary Fig. S12, showing that indeed CTH increased during the study period (Fig. S12b), and in fact this increase occurred in late autumn and early winter (Fig. S12c).

While these signs of change are consistent with the previous explanation, the lack of statistical significance in CTH changes, along with differences between the two data records, renders further conclusions dubious. Furthermore, CTH refers to all clouds, and a change in liquid CFC would also change the mean CTH, making interpretations more difficult.

The long time range available from CLARA-A2 data (34 years, starting in 1982) offers the opportunity for further evaluation of the cloud properties changes reported before, especially with respect to changes during the past three decades. For this

purpose, changes from all possible time ranges, at least 10 years long and starting from 1982 onward, were estimated for the study region. Results, shown in Fig. 9, suggest that the ranges of changes reported in Fig. 8 are not typical of the entire 34-year CLARA-A2 period. Specifically, for LWP, liquid CFC and liquid COT, the largest increases occur when the time range examined ends within the last five years of the CLARA-A2 period (2011-2015), indicating that corresponding values reached maxima during these years. Furthermore, for liquid REFF, a switch in the sign of change appears in the last years:

while liquid REFF is mainly decreasing for most start and end year combinations, only positive changes appear after 2003, indicating a consistent increase during the last years. It should be noted that abrupt changes appearing in the plots of Fig. 9 should be attributed to artifacts especially in the early years of the CLARA-A2 data record. Specifically, negative changes in liquid CFC occurring for starting years between 1988 and 1994 coincide with the period when AVHRR on NOAA-11 was operational, which caused a small discontinuity in the time series. Additionally, the switch from channel 3b (at 3.7 μm) to

channel 3a (at 1.6 μm) on NOAA-16 AVHRR during 2001-2003 caused a discontinuity in the cloud property time series, most prominently visible for REFF. A similar, long time range analysis of aerosols was not possible, due to the lack of available aerosol data.

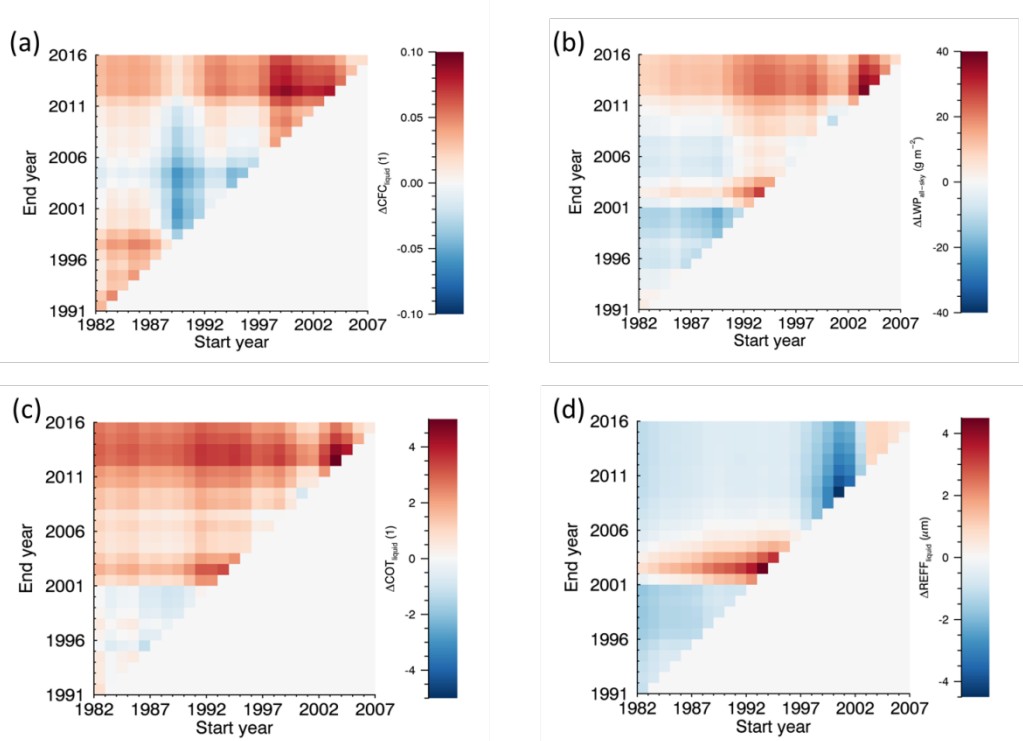

**Figure 9. Changes in liquid cloud properties over southern China, based on 34 years of CLARA-A2 data (1982-2015) and estimated for all possible combinations of start and end years, with a minimum time range of 10 years. The four plots show corresponding changes in (a) all-sky LWP, (b) liquid CFC, (c) liquid COT and (d) liquid REFF.**

As for aerosols, the seasonality of cloud property changes was also analyzed. Figure 10 shows that the overall increase in liquid clouds during the 10-year period examined can be attributed to changes occurring mainly in November and December. In fact, the patterns of seasonal changes show that CLARA-A2 and MODIS agree very well, with an increase in LWP occurring primarily in December and secondarily in November (Fig. 10a), and liquid CFC increases prevailing also in November and December (Fig. 10b). Corresponding results for liquid COT and liquid REFF (Figs. 10c and 10d) indicate the similarity in change patterns between COT and LWP, and the ambiguity in the REFF change between CLARA-A2 and MODIS, especially in November. The liquid CFC change is statistically significant in the November case, while all other cloud property changes shown in Fig. 10 are significant in December. Detailed levels of significance for all cloud properties are provided in supplementary Table S13.

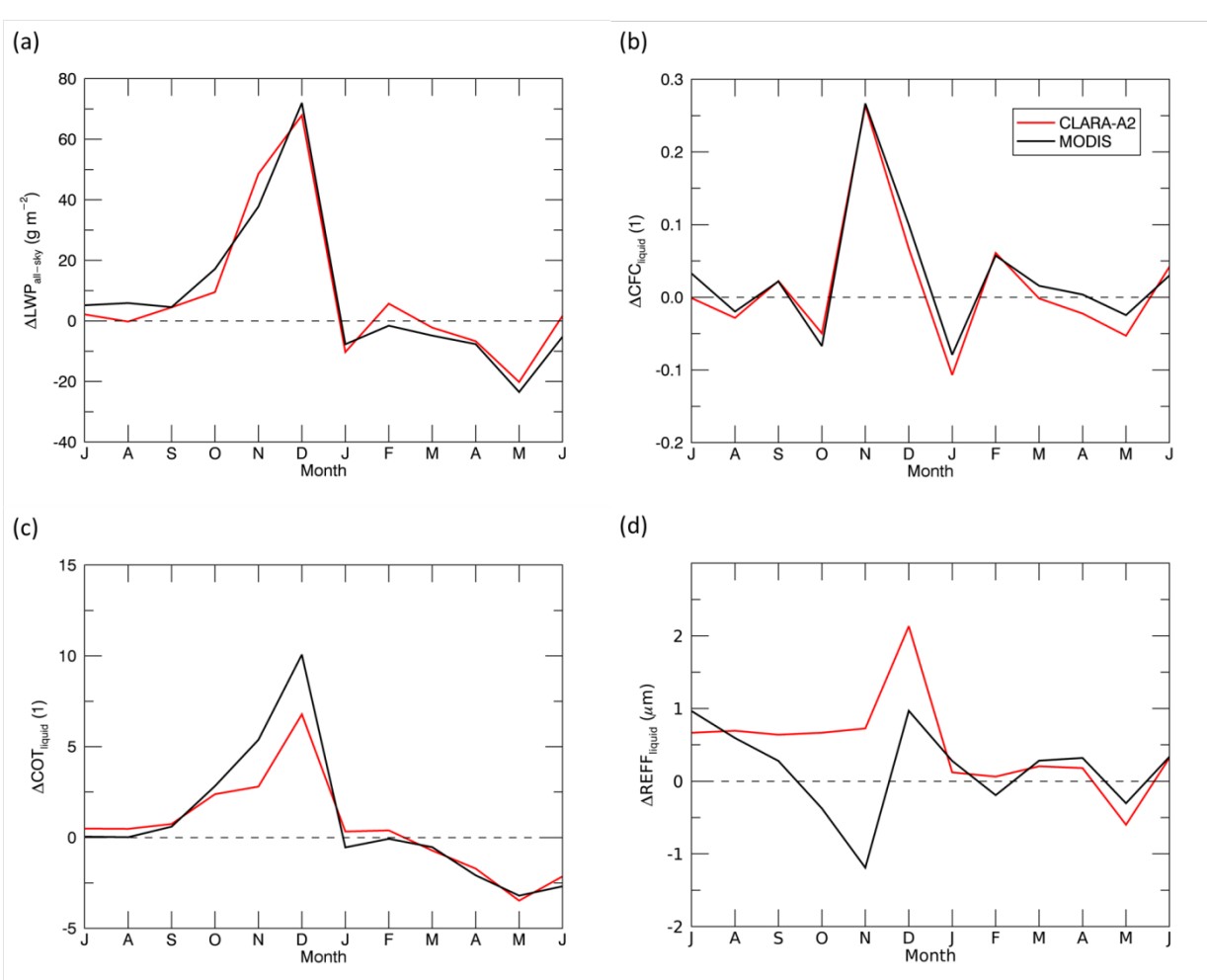

**Figure 10. Seasonal variation of changes in liquid cloud properties over southern China. (a) all-sky LWP, (b) liquid CFC, (c) liquid COT and (d) liquid REFF changes from 2006 to 2015 based on CLARA-A2 and MODIS data.**

## 3.3 Summary of aerosol and cloud seasonal changes

The results presented in the previous section show that during the 10-year study period, monthly AOD decreased mainly in autumn and early winter (Fig. 3), while GFED emissions decreased and cloud properties increased almost exclusively in November and December (Figs. 4b and 10). To add robustness to our findings, and realizing that averaging over full seasons will dilute the results too much, we have further aggregated the aerosol and cloud parameters to two-month periods. Table 1 summarizes the changes in AOD, GFED emissions, liquid clouds and precipitation on a bimonthly basis, with statistically significant changes highlighted in bold. This analysis makes clear that the period September-December drove the AOD changes during the study period, with significant decreases by about 40%, while GFED emissions only changed significantly

in November-December. As mentioned before, liquid cloud changes occurred mainly in November and December, with liquid CFC increasing by around 40% and LWP almost doubling. Precipitation also increased significantly in November and December, showing consistency with other cloud changes (increase in LWP, CTH), and providing a possible explanation for part of the aerosol reduction. Overall, there is a concurrence of substantial aerosol and cloud variations in late autumn and early winter.

**Table 1. Relative change (in % for all data except for $T_{2m}$ in K) of two-monthly aerosol, GFED BC+OC emissions, cloud, precipitation and $T_{2m}$ parameters over southern China from 2006 to 2015 (2007 to 2015 for CALIPSO AOD). Significant changes are indicated with boldface.**

| parameter | Jan+Feb | Mar+Apr | May+Jun | Jul+Aug | Sep+Oct | Nov+Dec |
|---|---|---|---|---|---|---|
| CALIPSO total AOD | -2 | -14 | -11 | -12 | **-42** | -34 |
| MODIS total AOD | -10 | 10 | 0 | -24 | **-38** | **-35** |
| MISR total AOD | -8 | 7 | 3 | -20 | **-39** | **-35** |
| MISR fine mode AOD | -11 | 2 | 3 | -19 | **-40** | **-41** |
| MISR coarse mode AOD | -6 | 16 | 5 | -24 | **-38** | -27 |
| GFED emissions | -54 | 14 | -35 | 69 | 50 | **-97** |
| CLARA liquid CFC | -3 | -1 | -1 | -3 | -3 | 35 |
| MODIS liquid CFC | -1 | 1 | 0 | 2 | -5 | **42** |
| CLARA all-sky LWP | -1 | -4 | -20 | 3 | 17 | **92** |
| MODIS all-sky LWP | -4 | -7 | -23 | 18 | 22 | **80** |
| CLARA CTH | -2 | -5 | 4 | 3 | 3 | 11 |
| MODIS CTH | -1 | -8 | 2 | 7 | 3 | **41** |
| GPCP precipitation | -22 | 13 | -10 | 1 | 36 | **208** |
| ERA $T_{2m}$ (in K) | -0.67 | -0.12 | 0.92 | -0.40 | -0.66 | -1.32 |

Further statistical analysis for November-December showed that there is indeed a strong, statistically significant anti-correlation between GFED emissions and AOD, on one side, and liquid cloud CFC and LWP, on the other. Results for all possible combinations examined are shown in Table 2, with statistically significant correlation coefficients in the 95% confidence interval highlighted in bold. These results reveal persistent anti-correlations, independently from the aerosol or cloud data sets used. The same analysis was performed for the entire seasonal cycle, showing that, apart from some spurious cases, significant correlations occur consistently only in November-December (Table S14).

**Table 2.** Linear correlation coefficients of November-December-mean GFED BC+OC emission and AOD time series with cloud properties and precipitation time series over southern China from 2006 to 2015 (2007 to 2015 for CALIPSO AOD). Significant correlations are indicated with boldface.

| parameter | GFED carbon emissions | CLARA liquid CFC | MODIS liquid CFC | CLARA all-sky LWP | MODIS all-sky LWP | CLARA CTH | MODIS CTH | GPCP precipitation |
|---|---|---|---|---|---|---|---|---|
| GFED carbon emissions | 1.00 | -0.54 | -0.54 | **-0.72** | **-0.77** | -0.03 | -0.66 | -0.62 |
| CALIPSO total AOD | 0.49 | **-0.77** | **-0.75** | **-0.69** | **-0.71** | 0.34 | -0.34 | -0.25 |
| MODIS total AOD | **0.78** | **-0.76** | **-0.81** | **-0.75** | **-0.84** | -0.14 | **-0.74** | -0.63 |
| MISR total AOD | **0.73** | **-0.66** | **-0.74** | **-0.66** | **-0.81** | -0.27 | **-0.73** | **-0.70** |
| MISR fine AOD | **0.79** | **-0.66** | **-0.74** | **-0.70** | **-0.84** | -0.30 | **-0.78** | **-0.71** |
| MISR coarse AOD | **0.63** | -0.62 | **-0.69** | -0.55 | **-0.72** | -0.21 | -0.60 | **-0.65** |

An important question is which mechanisms could explain the concurrent variation of aerosol and cloud properties. A first possibility is that large-scale meteorological variability affects both aerosols and clouds simultaneously, either through a natural cycle or affected by climate change. Secondly, local-scale ACI and/or ARI mechanisms could lead to cloud changes as a result of aerosol changes. A combination of these factors should not be excluded either. A second question arising from the previous results, is why significant cloud changes occur in November-December only, while aerosols change significantly also in September-October (Table 1). We attempt to address these questions in the following section.

## 4 Discussion

### 4.1 Possible effects of meteorological variability and large-scale phenomena

Based on the Intergovernmental Panel on Climate Change (IPCC) definition of climate change (IPCC, 2018), it is not reasonable to examine effects of climate change occurring within a 10-year only period. However, complex feedback mechanisms affected by human activities that initiated in the past could be affecting larger scale phenomena, such as seasonal patterns and large-scale circulation, and through them playing a role in the changes reported here. Aerosol regimes, in particular, are determined by processes describing emissions, atmospheric transformations and deposition. The dependency of these processes on climate change varies considerably among aerosol sources and types, while other, local factors, may play an equal or even more important role. The effect of climate change is clearer when aerosols from natural sources prevail, e.g. desert dust and marine salt. In these cases, it can affect the aerosol regime mainly through changes in atmospheric dynamics. In areas where aerosols come mainly from anthropogenic sources, however, including the wider South and Southeast Asia regions (Zhang et al., 2012), possible effects of climate change on aerosols will manifest mainly in terms of transportation and deposition, since the main factors affecting emissions are economic development and environmental policies (Chin et al. 2014). Overall, effects of climate change can be indirect, and affect aerosol transformation and deposition, through atmospheric variables like temperature and wind speed (Tegen and Schepanski, 2018).

Hence, an attempt was made to assess this kind of effects. This included first the analysis of surface air temperature ($T_{2m}$) , while natural variability was then analysed by examining changes in atmospheric circulation patterns. Changes in

atmospheric circulation related to large-scale phenomena affecting the wider South-East Asia region, namely the El Nino Southern Oscillation (ENSO) and Asian Monsoon (AM) cycles, were also examined.

The $T_{2m}$ analysis was based on reanalysis data from the ERA-Interim data set (Dee et al., 2011). No significant change was detected in $T_{2m}$ over the study region during the period examined, either in the entire time series, or when examining each

month separately. It is interesting to note, however, that a relatively strong decrease (although not statistically significant) took place in November–December, coinciding with the increase in cloud properties (Table 1). A possible explanation for this coincidence would be that more clouds over the region led to less solar radiation reaching the surface, thus reducing the surface air temperature. These findings are similar to the ones reported in Zhang et al. (2019); they also show a decrease in temperature over southern China after 1997 and practically no change in the period 2005-2015 (see their figures 5b and 3a,

respectively).

For the assessment of changes in atmospheric circulation we used surface pressure ($P_S$) and 500 hPa geopotential height ($Z_{500}$) fields from the CAMS reanalysis data record (Flemming et al., 2015; 2017). These data sets are available on a monthly basis and at 1° × 1° spatial resolution. Similarly to the aerosol and cloud properties, the analysis was based on deseasonalized linear regressions of the entire time series of monthly averages, as well as changes on a monthly basis,

focusing especially on months when aerosol and cloud changes maximize (i.e. November-December). For this analysis, however, the study area was extended by 10° in every direction, to include large-scale patterns that could be affecting the southern China region.

Results of this analysis are shown in supplementary Fig. S15, in terms of both average values of $Z_{500}$ and $P_S$ (Figs. S15a and S15c, respectively) and changes during 2006-2015 (Figs. S15b and S15d, respectively). Average values of $P_S$ and $Z_{500}$ follow

the topography of the region, with lower values over areas with higher elevation. The patterns of changes appear different, with a south-to-north gradient in $Z_{500}$ (Fig. S15b) and some $P_S$ increases and decreases over sea and land, respectively (Fig. S15d). This analysis, however, shows that $Z_{500}$ changes at the grid cell level are in the order of several meters and $P_S$ changes are just a fraction of 1 hPa. Even for specific months, $P_S$ changes are up to a few hPa, with no statistical significance. These results suggest that meteorological variability is not among the major factors contributing to the aerosol and cloud changes

reported.

Regarding possible effects of ENSO over southern China, the Oceanic Nino Index (ONI) was used to examine possible correlations between ENSO and the aerosol and cloud properties analysed here. ONI is the National Oceanic and Atmospheric Administration (NOAA) primary indicator for measuring ENSO; it is defined as the 3-month running Sea Surface Temperature (SST) anomaly in the Nino 3.4 region, based on a set of improved homogeneous SST analyses (Huang

et al., 2017). This analysis showed no particular correlation between ONI and cloud or aerosol properties; Correlation coefficients were around -0.2 for the entire time series and slightly larger for specific months. A very similar, not significant, anti-correlation between ENSO and low cloud amount was found by Liu et al. (2016), examining the entire China and the period 1951-2014.

The overall effects of AM on the area are most pronounced in summer. Although AM is known to affect aerosol

concentrations (through wet deposition during the raining season) and cloud cover, this seasonality pattern does not coincide temporally with the seasonal aerosol and cloud changes reported here. Furthermore, it is known that AM and ENSO are strongly correlated (Li et al. 2016), hence the effects of the former on these changes are expected to be similarly insignificant as those of the latter.

## 4.2 Possible effects of ACIs and ARIs

Although cause and effect mechanisms cannot be proven based on observations only, possible underlying ACI and ARI mechanisms are worth investigating, since the combination of aerosol and cloud changes can also be used to exclude some of them.

Following this approach, our results appear inconsistent with the standard definitions of the first and second aerosol indirect effects, although the possibility of multiple mechanisms occurring simultaneously cannot be excluded. Specifically, according to the first aerosol indirect effect, a decrease in aerosols would lead to an increase in cloud droplet size, under constant liquid water content. In our case, while both CLARA-A2 and MODIS indicate an overall increase in liquid REFF (Fig. 8f), these changes do not coincide seasonally with any significant aerosol change (Fig. 3). In fact, mixed signs in liquid

REFF change were observed in November (Fig. 10d). Additionally, the LWP increases considerably, suggesting that the first indirect effect mechanism does not play a major role. Furthermore, the already high aerosol loads over the region in the recent past may have led to a saturation in the role of cloud condensation nuclei (CCN) to droplet formation. According to the second aerosol indirect effect, a decrease in aerosols implies reduced cloud life time through more rapid precipitation. While an increase in precipitation coinciding with a decrease in aerosols was reported, an increase in liquid cloud fraction

was also observed, suggesting increased cloud life time, which is contrary to this mechanism.

Contrary to the first and second aerosol indirect effects, the semi-direct effect cannot be readily excluded as an explanatory process, since the signs of changes of all aerosol and cloud variables presented here are consistent with what would be expected based on this mechanism. Specifically, this effect predicts that a decreasing absorbing aerosol load inside the cloud layers would lead to reduced evaporation of cloud droplets and hence increased cloudiness and cloud water content. It is

important noting that this mechanism holds primarily for absorbing aerosols, such as biomass burning particles, while aerosols from air pollution can also be absorbing. It is also important noting that the position of the aerosols relative to the cloud layer determines the sign of the semi-direct effect: a decrease in aerosols will lead to increased cloudiness only if the aerosols are at the same level with clouds. If the aerosols are above clouds, the effect will be the opposite (Koch and Del Genio, 2010).

In order to further examine the possibility of the semi-direct effect as an underlying mechanism, an analysis of the vertically resolved changes in aerosol extinction profiles was conducted, based on CALIPSO data, combined with typical values of cloud extinction profiles for this region. September-October and November-December were selected, since they exhibit a significant decrease in aerosols, with the main difference being that in November-December cloud changes were also prominent. Figure 11a shows the typical profile of cloud extinction in autumn over southern China, available from the

LIVAS data set (Lidar climatology of Vertical Aerosol Structure for space-based lidar simulation studies; Amiridis et al., 2015) based on measurements from 2007 to 2011. It is apparent that low clouds prevail during this season. Figures 11b and 11c show, for the same height range, changes in the aerosol extinction profiles in September-October and November-December during 2007-2015. In September-October, changes occurred mainly at an elevated altitude. When compared with the cloud extinction profile, it appears that the decrease in aerosols tended to occur mostly above clouds. In November-

December, however, the decrease was more pronounced towards the surface. In fact, the shape of the profile change suggests that most of the November-December decrease occurred near or within clouds. The aerosol profile change in November-December also implies a local origin of aerosols. A decrease in aerosols from local sources is expected to be proportional to their typical profile (higher concentrations at lower atmospheric levels). It should be noted here, that the uncertainty in aerosol extinction profiles retrieval from CALIPSO increases in lower atmospheric layers (Young et al., 2013), thus

decreasing the confidence in the results towards the surface. The vertically resolved analysis of aerosol changes showed that the significance level in September-October (Fig. 11b) exceeds 95% between 1.3 km and 2.5 km altitude, while changes in November-December are significant between 0.6 km-1.0 km and 2.0-2.5 km.

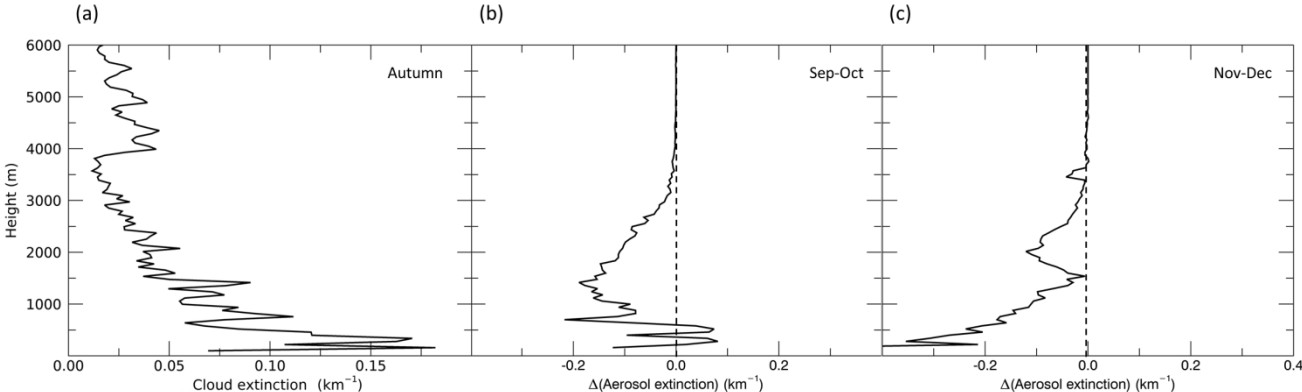

**Figure 11. Profiles of cloud and aerosol changes over southern China. (a) Cloud extinction in autumn (September-November), estimated based on LIVAS CALIPSO data from 2007-2011. Aerosol extinction change for September-October (b) and November-December (c) based on CALIPSO level 3 data from 2007-2015.**

These results show consistency with an aerosol semi-direct effect mechanism acting under decreasing aerosol loads in the November-December case. Specifically, the decrease in aerosols within clouds in these months coincides with an increase in liquid cloud fraction and water content in low liquid clouds (Figs. 10a, 10b), with a significant anti-correlation (Table 2). The decrease in aerosols above clouds (September-October case), on the other hand, has no coincidence with any significant cloud change. A possible explanation for this difference between the two periods examined is that in September and October aerosols are not strongly absorbing, compared to the November-December case.

## 5 Summary

In the present study, aerosol, emissions and cloud characteristics and changes were analysed based on a combined use of multiple independent remote sensing data sets. The study focused on the southern China region, which is characterised by intense aerosol-producing human activities, while a significant decrease in aerosol loads has previously been reported. In agreement to these previous reports, it was found that aerosol loads over the region decreased significantly in autumn and early winter months, specifically in September-December. This decrease could be partially attributed to an increase in precipitation, which occurred roughly during the same months. The decrease in aerosols also coincided with large decreases in biomass burning emissions in November and December. Concurrent changes in liquid cloud fraction and water path were observed in these two months, with notable increases in both. Possible physical mechanisms that could be causing these cloud changes were analysed, including interannual meteorological variability, the ENSO phenomenon and the Asian Monsoon, which largely drives the seasonal behaviour of clouds over the region. However, no apparent connection was found between these phenomena and the cloud changes reported here.

The possibility of interactions between aerosols and clouds having played a role in the cloud changes was also examined, although no cause-and-effect mechanism can be established based on observations only. However, the first and second aerosol indirect effects could be excluded as dominant mechanisms by noting that the signs of changes of aerosols and cloud properties are inconsistent with the predictions of these mechanisms. This approach, however, is not sufficient to exclude the possibility of a semi-direct effect occurring under decreasing aerosol loads, whereby less absorbing aerosols residing in liquid clouds would lead to a reduction in cloud evaporation and a corresponding increase in cloud cover and LWP. The aerosol and cloud changes and correlations observed in November-December are consistent with this mechanism.

While the aerosol semi-direct effect has been studied in the past through both model simulations (e.g. Allen and Sherwood, 2010; Ghan et al., 2012) and analysis of observations (e.g. Wilcox, 2012; Amiri-Farahani et al., 2017), it should be stressed here that the combined analysis of different aerosol and cloud data sets can only provide strong indications, without proving

any cause and effect mechanism. This analysis rather represents a contribution to the observational approaches in aerosol-cloud-radiation interaction studies, highlighting both the possibilities and limitations of these approaches. To overcome some of these limitations, further research should focus on model simulations of the conditions described here, in order to provide more insights regarding the underlying physical mechanism.

## Data availability

MODIS aerosol and cloud data were obtained from https://ladsweb.modaps.eosdis.nasa.gov; MISR data were obtained from ftp://ftp-projects.zmaw.de/aerocom/satellite; CALIPSO aerosol data were obtained from https://eosweb.larc.nasa.gov/project/calipso/cal_lid_l3_apro_allsky-standard-v3-00; GFED data were obtained from http://www.geo.vu.nl/~gwerf/GFED/GFED4; CLARA-A2 cloud data were obtained from http://www.cmsaf.eu; LIVAS cloud data were obtained from http://lidar.space.noa.gr:8080/livas. CAMS data were obtained from https://eccad3.sedoo.fr/#CAMS-GLOB-ANT. GPCP data were obtained from http://gpcp.umd.edu/.

## Author contributions

N.B. and J.F.M. developed the methodology and performed the analysis. All authors contributed in interpreting the results, writing, editing and finalizing the manuscript.

## Acknowledgments

We acknowledge the EUMETSAT CM SAF project for providing support to this study.

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

Figure captions

Figure 1. Seasonal variation in Aerosol Optical Depth (AOD) over southern China, based on the period 2006-2015, from MODIS, MISR and CALIPSO, including MISR fine, coarse and dust mode AOD. Note that the horizontal axis starts in July and ends in June.

Figure 2. Changes in AOD over southern China from 2006 to 2015. (a) Spatial distribution of AOD change over the study region deduced from MODIS data. Spatially averaged monthly deseasonalized values of AOD from MODIS (b), CALIPSO (c), and MISR (d). Shaded areas correspond to one standard deviation of the grid-scale monthly averages. Dotted lines correspond to linear regression fits. Percent changes during the period examined are also shown, with the statistically significant ones indicated in bold.

Figure 3. Seasonal variation of changes in AOD over southern China from 2006 to 2015 deduced from MODIS, MISR and CALIPSO data. MISR data include fine, coarse and dust mode AOD.

Figure 4. (a) Seasonal variation in organic and black carbon emissions from GFED (in Gg C) over southern China from 2006 to

2015. (b) Corresponding changes on a monthly basis (in Gg C month$^{-1}$) during the same period.

Figure 5. Emissions of aerosols and precursor gases from the Copernicus Atmosphere Monitoring Service (CAMS). The emissions have been aggregated to annual totals over the southern China study area and plotted relative to the year 2006.

Figure 6. (a) Seasonal variation of precipitation in the southern China study region based on Global Precipitation Climatology Project (GPCP) data. (b) Corresponding spatially averaged monthly deseasonalized values. The dotted line corresponds to the linear regression fit. (c) Seasonal variation of changes in GPCP precipitation. Seasonal averages and changes in (a) and (c) are based on data from the period 2006-2015.

Figure 7. Seasonal variations in cloud properties over southern China, based on CLARA-A2 and MODIS data, during the period 2006-2015. (a) Total CFC, (b) cloud phase (CPH; fraction of liquid clouds relative to total CFC), (c) COT for liquid clouds and (d) REFF for liquid clouds.

Figure 8. Changes in liquid cloud properties over southern China from 2006 to 2015, based on CLARA-A2 and MODIS data. (a),

(b) Spatial distributions of changes in all-sky LWP and liquid CFC based on CLARA-A2 data. Spatially averaged monthly deseasonalized values of all-sky LWP (c), liquid CFC (d), liquid COT (e) and REFF (f). Percent changes during the period examined are also shown, with the statistically significant ones (only CLARA-A2 liquid REFF) indicated in bold.

Figure 9. Changes in liquid cloud properties over southern China, based on 34 years of CLARA-A2 data (1982-2015) and

estimated for all possible combinations of start and end years, with a minimum time range of 10 years. The four plots show corresponding changes in (a) all-sky LWP, (b) liquid CFC, (c) liquid COT and (d) liquid REFF.

Figure 10. Seasonal variation of changes in liquid cloud properties over southern China. (a) all-sky LWP, (b) liquid CFC, (c) liquid COT and (d) liquid REFF changes from 2006 to 2015 based on CLARA-A2 and MODIS data.


Figure 11. Profiles of cloud and aerosol changes over southern China. (a) Cloud extinction in autumn (September-November), estimated based on LIVAS CALIPSO data from 2007-2011. Aerosol extinction change for September-October (b) and November-December (c) based on CALIPSO level 3 data from 2007-2015.
