# Peer review of "Satellite observations of aerosols and clouds over southern China from 2006 to 2015: analysis of changes and possible interaction mechanisms"

_Atmospheric Chemistry and Physics, 2018_

## Referee Comment (RC1) · Anonymous Referee #1 · 30 Oct 2018

Review of paper:

Satellite observations of aerosol and clouds over southern China from 2006 and 2015: analysis of changes and possible interaction meachanisms. by N.Benas et al.

Positives - exploring co-located changes in aerosol and cloud retrievals for insights - picking a (S.China) region, where aerosol loads have been changing on decadal scales - looking for consistency by exploring different aerosol and cloud products

Concerns - AOD is a poor indicator for CCN concentrations – AODf is much better (I suggest MISR) - CALIPSO typing cannot really distinguish between pollution and wildfire (and wildfire is not so effective for CCN as pollution) thus looking as GFED

(biomass, van der Werf) emissions rather than also industrial emissions (IPCC6, Smith) is not convincing - the examined region is relatively small - the opportunity to contrast impacts with aerosol increase (before 2008) and aerosol decrease (after 2012) is missed - many data limitation / inconsistencies are recognized but not further explored . . . so the value of the paper is limited and the suggested links remain speculative.

General comments:

This study examines co-located 'observational data' based on satellite retrievals for aerosol and clouds over time. Here, a relatively small region over China is picked since over the larger China region the (fine-mode) aerosol loads after many decades on continued increase have been decreasing over the last decade. Co-located cloud properties over the same period were examined. The associations suggest that for aerosol impacts on low water clouds the first indirect effects (Twomey) seem unimportant (which is not completely surprising as baseline CCN concentrations are already very high), while cloud lifetime impacts (aerosol solar absorption and heating associated evaporation for less cloud cover) seem more relevant. Hereby it is suggested that the reduced aerosol load and absorption is recent years might explain increases to low altitude cloud cover and to liquid water content. There are many (admitted) observational limitations, especially with respect the interpretation of aerosol type thus I recommend also to look at the fine-mode AOD of the MISR retrieval, which I placed on ftp ftp://ftp-projects.zmaw.de/aerocom/satellite/mis_v23/ It seems very promising to examine cloud property changes in regions where large changes in aerosol loading have occurred. However for that region an opportunity is missed by just exploring periods of a recent AOD decline (especially since 2012), whereas it was contrasted by a strong AOD increase before 2008. Thus opposing cloud property changes should have been observed, if there was an aerosol impact on clouds. Unfortunately even with these AOD changes the aerosol loading was quite high so that CCN concentrations may been already saturated with respect to droplet formation which in part explain the largely 'missing' first indirect effect. Overall, I like the paper but I am sometimes

dismayed at the recognition of shortfalls without going into further detail. The data-consistency (e.g. CLARA) also is often a major handicap so that despite of significant data-analysis often there remain relatively little useful information to work. The use of reanalysis data is an interesting aspect, and I just wonder if they the MODIS data assimilation in MACC actual changes in cloud-properties in the examined China regions are simulated. I suggest also not to look just at biofuel but also fossil fuel emission (trends) as alternate aerosol change indicator although working with actual AODf is probably best.

details in the supplement

Please also note the supplement to this comment:
https://www.atmos-chem-phys-discuss.net/acp-2018-982/acp-2018-982-RC1-supplement.pdf

**Supplement:**

Review of paper:

**Satellite observations of aerosol and clouds over southern China from 2006 and 2015: analysis of changes and possible interaction meachanisms.**

*by N.Benas et al.*

**Positives**
- exploring co-located changes in aerosol and cloud retrievals for insights
- picking a (S.China) region, where aerosol loads have been changing on decadal scales
- looking for consistency by exploring different aerosol and cloud products

**Concerns**
- AOD is a poor indicator for CCN concentrations – AODf is much better (I suggest MISR)
- CALIPSO typing cannot really distinguish between pollution and wildfire (and wildfire is not so effective for CCN as pollution) thus looking as GFED (biomass, van der Werf) emissions rather than also industrial emissions (IPCC6, Smith) is not convincing
- the examined region is relatively small
- the opportunity to contrast impacts with aerosol increase (before 2008) and aerosol decrease (after 2012) is missed
- many data limitation / inconsistencies are recognized but not further explored … so the value of the paper is limited and the suggested links remain speculative.

**General comments:**

This study examines co-located 'observational data' based on satellite retrievals for aerosol and clouds over time. Here, a relatively small region over China is picked since over the larger China region the (fine-mode) aerosol loads after many decades on continued increase have been decreasing over the last decade. Co-located cloud properties over the same period were examined. The associations suggest that for aerosol impacts on low water clouds the first indirect effects (Twomey) seem unimportant (which is not completely surprising as baseline CCN concentrations are already very high), while cloud lifetime impacts (aerosol solar absorption and heating associated evaporation for less cloud cover) seem more relevant. Hereby it is suggested that the reduced aerosol load and absorption is recent years might explain increases to low altitude cloud cover and to liquid water content.

There are many (admitted) observational limitations, especially with respect the interpretation of aerosol type thus I recommend also to look at the fine-mode AOD of the MISR retrieval, which I placed on ftp [ftp://ftp-projects.zmaw.de/aerocom/satellite/mis_v23/](ftp://ftp-projects.zmaw.de/aerocom/satellite/mis_v23/)
It seems very promising to examine cloud property changes in regions where large changes in aerosol loading have occurred. However for that region an opportunity is missed by just exploring periods of a recent AOD decline (especially since 2012), whereas it was contrasted by a strong AOD increase before 2008. Thus opposing cloud property changes should have been observed, if there was an aerosol impact on clouds. Unfortunately even with these AOD changes the aerosol loading was quite high so that CCN concentrations may been already saturated with respect to droplet formation which in part explain the largely 'missing' first indirect effect.

Overall, I like the paper but I am sometimes dismayed at the recognition of shortfalls without going into further detail. The data-consistency (e.g. CLARA) also is often a major handicap so that despite of significant data-analysis often there remain relatively little useful information to work. The use of reanalysis data is an interesting aspect, and I just wonder if they the MODIS data assimilation in MACC actual changes in cloud-properties in the examined China regions are simulated. I suggest also not to look just at biofuel but also fossil fuel emission (trends) as alternate aerosol change indicator although working with actual AODf is probably best.

**Minor comments:**

**figures**

Figure 1    biomass burning emissions are at a maximum in spring but AOD/ AODf reductions (of MISR see below) are largest in fall. I would also add seasonalities of MISR data (AOD, AODfine and AODnonsphere[dust])

Figure 2    MODIS data over land have limited accuracy and MISR data are there usually more accurate … although the reduced AOD in that region is not questioned. The component AOD type assignments of CALIPSO involve many assumptions and should not be overinterpreted

Figure 3    the AOD reductions by MODIS and CALIPSO are max in October, for which GEFD changes can be hardly used to explain. Also consider that the CALIPSO 'smoke' in fact could be 'pollution'

Figure 4    cloud fraction is relatively high (ca 80%) so that the number of successful pixels in MODIS retrievals maybe very small and that CALIPSO will miss cases with strong pollution below clouds. Why is the water cloud phase dominant in the coldest seasons (that is not very intuitive). Also I would assume the largest cloud-fraction, COT and LWP during the monsoon season in summer, which the seasonality does not show.

Figure 5    the increase to LWP and cloud cover change have a strong local signature so I wonder if working with large regional changes is too simple. Even these annual averages display strong variations over time, so that I wonder if the identical 'trends' are significant. Also COT an REF retrievals are only possible for overcast conditions – so in that context tendencies in overcast cloud conditions would be of interest as well.

Figure 6    AVHRR over the years have different overpass time (trends may reflect in part daily cycles) and different sensors, so CLARA may not have the homogeneity needed for long-term trends. On the other hand the long time series is of advantage as (industrial) fine-mode AOD likely increased until 2008 and likely decreased since 2008 over China. So if there is a cloud property response I would expect a clear bi-polar response, which I do not see.

Figure 7    the largest cloud property changes (increases to all examined cloud properties!) are in December – not the month with the largest AOD changes (October). Also the largest AOD and AODf values of MISR in that region are in spring (consistent with your Figure 1)

Figure 8    the changes in October are likely pollution rather than smoke and the category of dust mixtures is too imprecise for me to be useful.

Figure 9    Wouldn't it be better to show the histogram just for Oct and Nov when you focus on Oct and Nov changes. Actually I am bit uncomfortable just to focus on two individual months rather than on an entire season – in the context of meteorological variability

**text**

page 2 / 20    there are some issues with the total AOD, since the it is contaminated by larger dust particles, which do not contribute as smaller particles from wildfire and pollution to particle concentrations and potential CCNs. MODIS only offers to total AOD and there are issues with the accuracy over land (for which the data are applied here) and the opportunity to retrieve aerosol maybe be limited by the relatively high low cloud cover. And the aerosol typing of CALIPSO is only quantitative – especially for 'mixtures' and it is extremely difficult to distinguish between

pollution and biomass aerosol types. I think the use of MISR data also should be included in order to include AOD trends, especially since MISR also addressed the AODf.

page 2 / 40    the mentioning of the GFEDv4 data is an interesting concept to justify aerosol changes due to biomass burning, but I would argue that industrial emission reductions (talk to Steve Smith to provide his industrial emission data for that region) are much more likely to explain satellite observed AODf reductions in recent years.

page 3 / 10    the MODIS data for (liquid) cloud cover is the most reliable property (though possibly biased by the overpass time and also dependent on the non-obscurance by higher altitude clouds). Details (LWC, Reff and COT) only refer to an overcast cloud subsample – so I wonder if also changes for overcast low cloud pixels can be addressed as well

page 3 / 15    I wonder how stable the CLARA data-set is, especially involving AVHRR data with drifting overpass time and different sensors.

page 3 / 30    consistent changes of different data is certainly a help but for clouds I wonder how independent the cloud retrievals (other than cloud cover) are.

page 4 / 12    what is a pixel? (I assume a 1x1 lat-lon region), thus there are 50 pixels in the regions investigated region

page 4 / 20    domestic burning may be largest in winter … but what about industrial emissions as I also expect that mitigations (e.g. to electric vehicles) have lowered emissions?

page 4 / 35    the aerosol typing of calipso should be taken with care. What does 'polluted dust' really tell you on fractions of dust and pollution? without a good distinction between pollution and smoke how to argue via biofuel (GFED) rather than fossil fuel (IPCC6) changes?

page 4 / 38    given the potential of advection it would be useful also to look at trends including surrounding regions (at least for the aerosol data)

page 5 / 2    MODIS pixels for AOD are usually provided at 10x10km so the use of 'pixel' here for 1x1 deg is a bit misleading

page 5 / 16    please also add industrial emissions in Figure 3 (contact S.Smith, who provides industrial emissions over time – also for China – for IPCC6 simulations)

page 5 / 42    it does not seem too intuitive why there should be so much less liquid water clouds the monsoon season: is this an artifact since there are more ice clouds so that the lower altitude clouds cannot be seen?

page 6 / 5    the 1x1 region based changes to water cloud cover and water content show a lot of sub-regional signals … so maybe also aerosol data (e.g. AODf data ) should be examined on that basis…?

page 6 / 20    in figure 6 changes in the y-axis direction are relevant, so that the Reff results are rather meaningless – other than could drops were relatively large in 2003 … which however was identified in the text as an artifact. Although Figure 6 is a nice analysis the results may not be robust enough to make good cases for cloud property changes.

page 6 / 20    the changes to aerosol (earlier) and cloud (later) do not have the quite same seasonality … and the admittance of potential meteorological causes is good though limiting to the goals of the study.

page 7 / 5        unfortunately no renalaysis data are shown (in that context it would be interesting if reanalysis sees the same 'trends' for aerosol and cloud properties as the observations. The reanalysis also could address if there were changes to the monsoon (strength and location) … e.g. were there trends in precipitation and are there temporal shifts (e.g. associated potentially with temporal delay with more fine-mode aerosol)? Also do the reanalysis data see shifts in fine-mode AOD altitude as indicated by CALIPSO?

page 7 / 25      to exclude monsoon impacts the focus is on changes to aerosol and clouds in the dry season … so maybe this can be even part of the title.

page 7 / 37      add 'rapid' after 'more'

page 8 / 1        also aerosol from pollution is absorbing (ssa ~ 0.92) – although usually not so strong as from fresh biomass aerosol (ssa ~ 0.85).

page 8 / 13      if there I so much low altitude cloud cover in Oct and Nov, how reliable is the CALIPSO statistics (other than the strong trend changes within 1 month seem highly suspicious)

page 8 / 35      interesting but speculative, although possible at least to some degree

page 9 / 10      to make the stated case on absorbing aerosol decrease in that region (other than aerosol in that region is absorbing to start with) it would be useful to demonstrate that these aerosol reductions are associated with the fine-mode AOD (at this point there is a reliance on Calipso typing which is quite general and its association with biomass burning is not very convincing as also urban/industrial (fossil fuel) emissions were reduced and may be even more relevant.

page 9 / 20      the disclaimer is warranted (though disappointing)

Here the MISR data for AOD (dust +pollution + wildfire) and AODf (only pollution + wildfire) are analyzed for changes. In the context of impacts on clouds the fine-mode AOD is more relevant since it determines the aerosol number concentration (MODIS and CALIPSO data of the study only address total AOD which is contaminated by coarse mode dust contributions – so here both total AOD as well as AODf are examined. Since MISR has limited coverage in the analysis monthly data were averaged on a 12(lon)x6(lat) degree grid. Difference maps to the local 18 year *dAODf*  *(2001-2017)  - annual* ge (2000-2017) better highlight differences.

**AOD**  *(2001-2017) – annual*                    **AODf**  *(2001-2017) – annual*

[Figure]

*differences to the long-time (17 year) local averages ….*

d**AOD** *(2001-2017)  - annual*                    d**AODf**  *(2001-2017)  - annual*

[Figure]

Here the MISR data

**AOD** *(2001-2017) – annual South China*     **AODf** *(2001-2017) – annual, South China*

[Figure]

*d**AOD** (2001-2017) – annual, South China*     *d**AODf** (2001-2017) – annual, South China*

[Figure]

Over the South China region almost half for the AOD is NOT in the fine-mode. Compared to the relative large AOD between 2004 and 2008 there is reduction in AOD and AODf since then.
In 2012 and 2013 the reductions are mainly due to the fine-mode (AODf) while lower values from 2015 to 2017 are caused by both fine-mode (AODf) and coarse-mode (AOD minus AODf).

Thus anomalies are now examined on a seasonal basis.

**AOD** *(05,09,13,17) – seasonal, south China*    **AODf** *(05,09,13,17) – seasonal, south China*

[Figure]

**dAOD** *(05,09,13,17) – seasonal, south China*    **dAODf** *(05,09,13,17) – seasonal, south China*

[Figure]

The AODf reductions are larger during summer and fall, reductions in winter and fall are smaller in 2017.

**AOD** *(10,12,14,16) – seasonal, south China*    **AODf** *(10,12,14,16) – seasonal, south China*

[Figure]

**dAOD** *(10,12,14,16) – seasonal, south China*    **dAODf** *(10,12,14,16) – seasonal, south China*

[Figure]

The AODf reductions are largest in fall during fall and winter, reductions in winter and summer are smaller and there is even an spring increase in 2012 and 2016.

---

## Referee Comment (RC2) · Anonymous Referee #2 · 6 Nov 2018

This paper synthesizes monthly-averaged satellite aerosol data from MODIS and CALIPSO, biomass burning emissions data from GFED, and cloud data from MODIS and CLARA-A2 to examine annual and seasonal trends for the South China region over the past decades. The purported goal of the study is two-fold (as stated on Pg. 2, Lines 9-13): 1) to analyze aerosol and cloud characteristics and changes, and 2) to investigate the possibilities and limitations of the synergistic use of this multitude of data for assessing aerosol and cloud interaction mechanisms. Only three aerosol types are investigated from the CALIPSO dataset: Dust, Smoke, and Polluted Dust. This substantially limits the conclusions that can be drawn with regard to aerosol source attribution; although, the manuscript forges ahead and attributes changes in the decadal

timeseries of aerosol optical depth large to changes in biomass burning, particularly residential energy sources (Pg. 5, Lines 14 and 18). The authors then average the decadal cloud microphysical data by month to look at seasonal trends, and find no clear seasonal trend in aerosol optical depth even as there are pronounced increases in liquid water path, cloud fraction, cloud optical thickness, and effective radius during the November-December time period. From these relationships, they conclude that the observed seasonal trends are inconsistent with the first and second aerosol-cloud indirect effects, but possibly consistent with the semi-direct effect (Section 3.3.2).

Such strong conclusions are not supported by the underlying data, which are themselves highly averaged in both space and time. The highly-averaged nature of the data makes it hard to draw conclusions other than to say that the seasonal trend of one variable appears to correlate with the trend of another variable or that one or more variables trend up/down slightly over time – such apparent correlations are neither causal nor attributive. No statistical tests are presented to quantify the robustness or strength of such correlations. Indeed, I find all of the authors' conclusions regarding the attribution of aerosol sources to biomass burning and their effects on clouds to be highly speculative. To quote the authors (Pg. 9, Lines 19-21): "These results do not constitute evidence of any cause and effect mechanism, which cannot be proved based on observations only. They rather represent a contribution to the observational approaches in aerosol-cloud-radiation interaction studies, highlighting both the possibilities and limitations of these approaches." From reading this paper, I don't know what approaches are being referred to here. The approach employed seems to have been to take a bunch of Level 3 temporally-averaged and gridded satellite data products, plot them up, and draw strong, unsupported conclusions about aerosol-cloud interactions based on perceived annual or seasonal trends.

In my opinion, this paper does not represent a substantial contribution to scientific progress, which is the minimum criterion for the ACP scientific significance review criteria. As I do not see a path forward by which this manuscript could be revised to be a

significant contribution, I recommend to the editor that this manuscript be rejected.

---

## Author Comment (AC1) · 27 Nov 2018

We thank the referee for the constructive review. Following are our point by point replies with the referee comments in italic.

*General comments:*

*There are many (admitted) observational limitations, especially with respect the interpretation of aerosol type thus I recommend also to look at the fine-mode AOD of the MISR retrieval, which I placed on ftp ftp://ftp-projects.zmaw.de/aerocom/satellite/mis_v23/*

We thank the referee for this suggestion and for providing the data. Using fine mode AOD as a better CCN indicator (compared to total AOD) will add robustness to our findings. We intend to include this analysis in a revised version of the manuscript.

*It seems very promising to examine cloud property changes in regions where large changes in aerosol loading have occurred. However for that region an opportunity is missed by just exploring periods of a recent AOD decline (especially since 2012), whereas it was contrasted by a strong AOD increase before 2008. Thus opposing cloud property changes should have been observed, if there was an aerosol impact on clouds. Unfortunately even with these AOD changes the aerosol loading was quite high so that CCN concentrations may been already saturated with respect to droplet formation which in part explain the largely 'missing' first indirect effect. Overall, I like the paper but I am sometimes dismayed at the recognition of shortfalls without going into further detail. The data-consistency (e.g. CLARA) also is often a major handicap so that despite of significant data-analysis often there remain relatively little useful information to work.*

We acknowledge the value of performing the same analysis for larger periods of time, in view of the known alternating signs of aerosol changes over this region. However, reliable satellite-based aerosol observations are not available before the 2000s. Additionally, to minimize the effects of interannual variability on the detected changes, the latter should be computed for relatively long periods (here we required at least 10 years). These factors render the combined examination of aerosol and cloud changes over larger periods practically unfeasible. The strong AOD decrease before 2008 that the referee mentions is not apparent in our analysis (Fig. 2), probably because the latter includes only two years prior to this switching point. However, we took the opportunity to examine clouds changes over a larger time period, offered by CLARA-A2. As the referee mentions, issues of long-term consistency in this data set reduce the availability of useful information. The remaining information, however, shows that the changes reported during the more recent years are considerably different (and in some cases opposing) than those based on earlier time periods (as shown in Fig. 6), suggesting a connection with the switch in the sign of aerosol changes in recent years.

The referee's remark on possible saturation in CCN concentrations is an important one that we plan to include in the relevant discussion.

*The use of reanalysis data is an interesting aspect, and I just wonder if they the MODIS data assimilation in MACC actual changes in cloud-properties in the examined China regions are simulated. I suggest also not to look just at biofuel but also fossil fuel emission (trends) as alternate aerosol change indicator although working with actual AODf is probably best.*

Comparisons of satellite-based aerosol and cloud data with corresponding CAMS data show consistent changes. However, in the case of CAMS this is due to data assimilation, and does not prove a successful model reproduction of aerosol-cloud interactions that occurred in reality.

Working with fine mode AOD is a constructive suggestion that we intend to incorporate.

*Minor comments:*

*figures*

*Figure 1 biomass burning emissions are at a maximum in spring but AOD/ AODf reductions (of MISR see below) are largest in fall. I would also add seasonalities of MISR data (AOD, AODfine and AODnonsphere[dust])*

From MODIS and CALIPSO we also get largest AOD reductions in fall and early winter (see Fig. 3). We plan to include MISR data in the revised manuscript.

*Figure 2 MODIS data over land have limited accuracy and MISR data are there usually more accurate … although the reduced AOD in that region is not questioned. The component AOD type assignments of CALIPSO involve many assumptions and should not be overinterpreted.*

We intend to include MISR data in the analysis. CALIPSO assumptions and issues are acknowledged, and they will be further emphasized in the framework of a more conservative interpretation.

*Figure 3 the AOD reductions by MODIS and CALIPSO are max in October, for which GEFD changes can be hardly used to explain. Also consider that the CALIPSO 'smoke' in fact could be 'pollution'*

As we explain in the manuscript, GFED does not provide a full picture of aerosol emissions (page 4, lines 37-38). Hence it should not be expected to explain all changes in AOD found from MODIS or CALIPSO. CALIPSO uncertainties will also be further highlighted, although possible misclassifications of the "smoke" subtype are already mentioned (page 5, lines 29-33).

*Figure 4 cloud fraction is relatively high (ca 80%) so that the number of successful pixels in MODIS retrievals maybe very small and that CALIPSO will miss cases with strong pollution below clouds. Why is the water cloud phase dominant in the coldest seasons (that is not very intuitive). Also I would assume the largest cloud-fraction, COT and LWP during the monsoon season in summer, which the seasonality does not show.*

Because of the high cloud fraction, there are indeed relatively few successful MODIS AOD retrievals. Therefore, we required only 10 daily AOD values to be present (compared to 18 for cloud properties) to accept a MODIS monthly-mean AOD for further analysis. On the relatively coarse grid scale of the MODIS products (1 x 1 degree) this is usually fulfilled. Similarly, the even much coarser (2 x 5 degrees) spatial resolution of the CALIPSO level-3 products also yields sufficient retrievals for the monthly means.

The monsoon season in summer is characterized by a larger fraction of high clouds with ice near the top, in particular convective clouds. In winter, low stratus/stratocumulus clouds prevail. In both seasons the cloud fraction is rather high, but one can imagine that persistent stratus will give a larger mean cloud fraction than intermittent convective systems. The seasonal cycle of cloud properties is retrieved very consistently from two independent satellite records, CLARA-A2 and MODIS, so we think it can be trusted.

*Figure 5 the increase to LWP and cloud cover change have a strong local signature so I wonder if working with large regional changes is too simple. Even these annual averages display strong variations over time, so that I wonder if the identical 'trends' are significant. Also COT an REF retrievals are only possible for overcast conditions – so in that context tendencies in overcast cloud conditions would be of interest as well.*

The LWP and cloud cover changes do show spatial variations, in particular with larger (positive) changes over land, and even slightly negative changes over sea. These patterns are consistent between CLARA-A2 and MODIS. We do not want to 'cherry-pick' exactly the grid boxes where we see the largest changes, and therefore we analyze time series of monthly-mean cloud properties averaged over the full region from both satellites, even if this might give somewhat weaker effects, because some parts with weaker trends are included. COT and REF retrievals are performed for cloudy pixels under the assumption that these are overcast. This will not always be the case, but we do not have sub-pixel information to determine that.

*Figure 6 AVHRR over the years have different overpass time (trends may reflect in part daily cycles) and different sensors, so CLARA may not have the homogeneity needed for long-term trends. On the other hand the long time series is of advantage as (industrial) fine-mode AOD likely increased until 2008 and likely decreased since 2008 over China. So if there is a cloud property response I would expect a clear bi-polar response, which I do not see.*

Orbital drift in NOAA satellites is important especially in the 80s and 90s. For the 10-year period examined in this study, data from NOAA-18 and NOAA-19 were used. Specifically, only the "primary" satellite was used in each month, meaning that when NOAA-19 data became available, NOAA-18 was not used any more. In this way any possible drift would be minor. The very good agreement with MODIS ensures that drift is not an issue for the 10-year period examined. For longer periods, depicted in Fig. 6, the effects of drifts are identified and acknowledged.

*Figure 7 the largest cloud property changes (increases to all examined cloud properties!) are in December – not the month with the largest AOD changes (October). Also the largest AOD and AODf values of MISR in that region are in spring (consistent with your Figure 1)*

Not all cloud properties have the largest change in December: liquid CFC has the largest increase in November. Apart from that, we agree that the largest aerosol and cloud changes don't coincide, but we also claim that such a coincidence is not a prerequisite for possible interactions. Aerosol type and height relative to clouds are also important factors, and constituted the main reason for CALIPSO data to be included in our analysis. We also don't see why the largest changes in AOD should coincide with the largest AOD concentrations. Large changes in autumn and large amounts in spring are not inconsistent.

*Figure 8 the changes in October are likely pollution rather than smoke and the category of dust mixtures is too imprecise for me to be useful.*

The most relevant part of aerosol type in this case is the optical properties and how these might affect interactions with clouds. We agree that attributing sources of aerosols based on their types is uncertain, but it is also of secondary importance in our case. The two main important points are (i) that the aerosols are absorbing and (ii) that they reside at a certain height, which is different in these two months. We will try to clarify this even more carefully in the revised version of the manuscript.

*Figure 9 Wouldn't it be better to show the histogram just for Oct and Nov when you focus on Oct and Nov changes. Actually I am bit uncomfortable just to focus on two individual months rather than on an entire season – in the context of meteorological variability*

Yes, this suggestion is reasonable and we will incorporate it. The focus on two separate months is due to the different behavior in CALIPSO aerosol subtypes (Fig. 3) and corresponding differences in vertical changes in these months (Fig. 8). Our results suggest that these changes affected clouds, based on the same explanatory mechanism which manifested in two different ways depending on the relative positions of aerosols and clouds. These effects would be indiscernible, had we averaged these months or examined the season as a whole.

*text*

*page 2 / 20 there are some issues with the total AOD, since the it is contaminated by larger dust particles, which do not contribute as smaller particles from wildfire and pollution to particle concentrations and potential CCNs. MODIS only offers to total AOD and there are issues with the accuracy over land (for which the data are applied here) and the opportunity to retrieve aerosol maybe be limited by the relatively high low cloud cover. And the aerosol typing of CALIPSO is only quantitative – especially for 'mixtures' and it is extremely difficult to distinguish between pollution and biomass aerosol types. I think the use of MISR data also should be included in order to include AOD trends, especially since MISR also addressed the AODf.*

We acknowledge the issues in using AOD as proxy to CCN. Possible reduced retrievals due to high low cloud cover are treated by applying all the requirement thresholds in the estimation of monthly means, described in Section 2.4. Regarding the quantitative aspects of AOD from CALIPSO, our focus is primarily on their optical properties and how they can affect interactions with clouds. Inferring sources of aerosols based on their types is secondary and we will try to clarify this in the revised version.

*page 2 / 40 the mentioning of the GFEDv4 data is an interesting concept to justify aerosol changes due to biomass burning, but I would argue that industrial emission reductions (talk to Steve Smith to provide his industrial emission data for that region) are much more likely to explain satellite observed AODf reductions in recent years.*

Reductions in aerosols from biomass burning activities over this area in recent years is well documented in the literature, which we include in our study, and the GFED analysis confirms these previous findings in terms of both seasonal distribution and changes. We claim that the change in these activities explain only part of the reduction in total AOD, as is also obvious e.g. from Fig. 3 and the relevant discussion (page 5). We agree that reductions in industrial emissions may also play a role in explaining the overall AOD reduction, and it is a constructive idea to include some relevant analysis as the referee suggests. It should be noted, however, and will be emphasized in the revised manuscript version, that the main goal of this study is to examine possible aerosol-cloud-radiation interaction mechanisms manifested due to the AOD decrease, rather than explain in detail every aspect in this decrease over the region. These interactions should not be expected to correlate with the overall aerosol change reported, but depend on specific aerosol types, relative positions and overall conditions.

*page 3 / 10 the MODIS data for (liquid) cloud cover is the most reliable property (though possibly biased by the overpass time and also dependent on the non-obscurance by higher altitude clouds). Details (LWC, Reff and COT) only refer to an overcast cloud subsample – so I wonder if also changes for overcast low cloud pixels can be addressed as well*

It is true that MODIS COT, REFF and consequently LWP are retrieved for a subsample of cloud-detected pixels (e.g. cloud edges are excluded). In that sense, analyzing the fraction of "overcast"-only cloudy pixels is a useful addition.

*page 3 / 15 I wonder how stable the CLARA data-set is, especially involving AVHRR data with drifting overpass time and different sensors.*

The issues that the referee mentions are more pronounced during the first two decades of CLARA-A2 data. In fact it has been shown that CLARA-A2 data are stable from the 2000s onwards (see also Karlsson et al., 2017). We use data from the 10 most recent years of the data set, from afternoon satellites NOAA-18 and NOAA-19. For every month, the most recent satellite available was used. Hence, the rapid succession of satellites ensures that when their orbital drift could start compromising the stability of the time series, their data were already not being used in the analysis. This point will be included in the revised manuscript to clarify the issue.

*page 3 / 30 consistent changes of different data is certainly a help but for clouds I wonder how independent the cloud retrievals (other than cloud cover) are.*

We understand the referee's point. Since both CLARA-A2 and MODIS use the Nakajima-King principle to retrieve cloud optical thickness and effective radius, their retrievals are not completely independent. We will rephrase accordingly.

*page 4 / 12 what is a pixel? (I assume a 1x1 lat-lon region), thus there are 50 pixels in the regions investigated region*

Correct, it will be clarified.

*page 4 / 20 domestic burning may be largest in winter … but what about industrial emissions as I also expect that mitigations (e.g. to electric vehicles) have lowered emissions?*

Yes, industrial emissions may also have been decreased, and their seasonality pattern will be different. A relevant description will be included in the revision for completeness. However, these changes are not depicted in the GFED analysis, since they originate in a different aerosol source.

*page 4 / 35 the aerosol typing of calipso should be taken with care. What does 'polluted dust' really tell you on fractions of dust and pollution? without a good distinction between pollution and smoke how to argue via biofuel (GFED) rather than fossil fuel (IPCC6) changes?*

As we mention in other parts, the primary goal of the study is not to attribute sources on different aerosol types, especially given the uncertainty that characterizes the CALIPSO aerosol subtypes. We argue, however, that consistency in seasonal characteristics and changes in an aerosol subtype with GFED constraints the possibilities of the origin of this subtype. In this sense, we claim that the "polluted dust" subtype is probably of local origin. For its possible effects on interactions with clouds, however, fractions of dust and pollution are of less importance.

*page 4 / 38 given the potential of advection it would be useful also to look at trends including surrounding regions (at least for the aerosol data)*

While this would support conclusions on the specifics of the origins of these changes, this analysis would exceed the scope of the study, since attributing specific reasons of changes in aerosol concentrations is not the primary goal.

*page 5 / 2 MODIS pixels for AOD are usually provided at 10x10km so the use of 'pixel' here for 1x1 deg is a bit misleading*

The term here refers to level 3 "grid cells". We will use this term consistently for more clarity.

*page 5 / 16 please also add industrial emissions in Figure 3 (contact S.Smith, who provides industrial emissions over time – also for China – for IPCC6 simulations)*

This is a constructive idea adding to the completeness of the analysis regarding sources of aerosol change and will be considered in the revised manuscript.

*page 5 / 42 it does not seem too intuitive why there should be so much less liquid water clouds the monsoon season: is this an artifact since there are more ice clouds so that the lower altitude clouds cannot be seen?*

Our results are verified by two independent studies cited in this part of the analysis (Pan et al., 2015; Cai et al., 2017). Following the referee's question, the same analysis for ice clouds will be included. Please refer also to our reply on the referee's previous comment on Figure 4.

*page 6 / 5 the 1x1 region based changes to water cloud cover and water content show a lot of sub-regional signals … so maybe also aerosol data (e.g. AODf data ) should be examined on that basis…?*

Following the referee's suggestion also in other comments regarding analysis of fine mode AOD, corresponding data from MISR will be included in the revised study.

*page 6 / 20 in figure 6 changes in the y-axis direction are relevant, so that the Reff results are rather meaningless – other than could drops were relatively large in 2003 … which however was identified in the text as an artifact. Although Figure 6 is a nice analysis the results may not be robust enough to make good cases for cloud property changes.*

In Fig. 6 the x-axis position is also important since it determines the start year of the period examined each time. Hence in the $R_{eff}$ case, apart from start and end years around 2000, which are known to be artifacts, the plot shows that there was a relatively constant decrease in the period before 2000 and a relatively constant increase afterwards. We consider that even after identification of the artifacts, this analysis still contains useful information, and it is a relatively robust way to exploit the long time series available from CLARA-A2 and highlight the importance of changes during the past 10 years compared to earlier periods.

*page 6 / 20 the changes to aerosol (earlier) and cloud (later) do not have the quite same seasonality … and the admittance of potential meteorological causes is good though limiting to the goals of the study.*

The seasonality in changes is indeed different. However, it would be oversimplified for ACI to manifest through common seasonality in changes of total aerosol and cloud properties. But in cases where strong changes in clouds and specific aerosols occur simultaneously, possible interactions playing a role are worth investigating. We consider however a prerequisite in such a study to "admit" potential meteorological causes and examine if these can explain the changes reported or if they can be excluded as explanatory mechanisms. Our analysis indicates that the latter is true.

*page 7 / 5 unfortunately no renalaysis data are shown (in that context it would be interesting if reanalysis sees the same 'trends' for aerosol and cloud properties as the observations. The reanalysis also could address if there were changes to the monsoon (strength and location) … e.g. were there trends in precipitation and are there temporal shifts (e.g. associated potentially with temporal delay with more fine-mode aerosol)? Also do the reanalysis data see shifts in fine-mode AOD altitude as indicated by CALIPSO?*

All the analyses suggested by the referee would be interesting and possibly strengthen some of our conclusions, although there is always a question when reanalysis agrees with observations: does this agreement verify an underlying mechanism or is it just due to assimilation? In the case of CAMS data, probably the latter holds, since the aerosol-cloud-radiation interactions mechanisms are not included in the model (Flemming et al. 2017). Such an  analysis would extend considerably the present study, of

which the purpose is to examine what can be possibly inferred using observations only, minimizing the need for analyzing reanalysis data.

*page 7 / 25 to exclude monsoon impacts the focus is on changes to aerosol and clouds in the dry season … so maybe this can be even part of the title.*

The season of largest aerosol and cloud changes is prominently mentioned in the abstract and elsewhere. We think that including this in the title would make it excessively long.

*page 7 / 37 add 'rapid' after 'more'*

Ok.

*page 8 / 1 also aerosol from pollution is absorbing (ssa ~ 0.92) – although usually not so strong as from fresh biomass aerosol (ssa ~ 0.85).*

This shows that even if part of the aerosol change is from pollution rather than biomass burning, there is still considerable absorption. We will include this detail in the revised version.

*page 8 / 13 if there I so much low altitude cloud cover in Oct and Nov, how reliable is the CALIPSO statistics (other than the strong trend changes within 1 month seem highly suspicious)*

In October and November the cloud fraction is around 70% (Fig. 4). Therefore, there are still considerable cloud-free periods, allowing to retrieve aerosol information from CALIPSO for the full atmospheric column. The aggregation of CALIPSO data over rather coarse (2 x 5 degree) grid boxes further increases the number of valid aerosol profiles. In addition, when low clouds are present, potential aerosols above these clouds can still be identified with CALIPSO.

*page 8 / 35 interesting but speculative, although possible at least to some degree*

This mechanism is indeed mentioned as a possibility, with no claim of proof based on the present analysis.

*page 9 / 10 to make the stated case on absorbing aerosol decrease in that region (other than aerosol in that region is absorbing to start with) it would be useful to demonstrate that these aerosol reductions are associated with the fine-mode AOD (at this point there is a reliance on Calipso typing which is quite*

*general and its association with biomass burning is not very convincing as also urban/industrial (fossil fuel) emissions were reduced and may be even more relevant.*

The association with biomass burning was based on the good agreement with the GFED, rather than reliance on CALIPSO aerosol subtypes only. However, fine mode AOD analysis will be included to add robustness to our conclusions.

*page 9 / 20 the disclaimer is warranted (though disappointing)*

We agree with the referee's comment. However, one should acknowledge these limitations, which are inherent to observations, and recommend the most promising way forward, which includes modelling simulations in order to overcome these limitations.

---

## Author Comment (AC2) · 27 Nov 2018

*This paper synthesizes monthly-averaged satellite aerosol data from MODIS and CALIPSO, biomass burning emissions data from GFED, and cloud data from MODIS and CLARA-A2 to examine annual and seasonal trends for the South China region over the past decades. The purported goal of the study is two-fold (as stated on Pg. 2, Lines 9-13): 1) to analyze aerosol and cloud characteristics and changes, and 2) to investigate the possibilities and limitations of the synergistic use of this multitude of data for assessing aerosol and cloud interaction mechanisms.*

*Only three aerosol types are investigated from the CALIPSO dataset: Dust, Smoke, and Polluted Dust. This substantially limits the conclusions that can be drawn with regard to aerosol source attribution; although, the manuscript forges ahead and attributes changes in the decadal timeseries of aerosol optical depth large to changes in biomass burning, particularly residential energy sources (Pg. 5, Lines 14 and 18).*

We demonstrate that CALIPSO observes significant negative changes in smoke and polluted dust AOD (which are consistent with MODIS observed changes) and that GFED carbon emission estimates also have strong decreases. These are two independent pieces of information suggesting that AOD changes are related to changes in carbon emissions from burning, which is furthermore supported by a number of studies referenced on page 5, line 19-20. Furthermore, we acknowledge the limitation in the CALIPSO level 3 aerosol data set that the reviewer mentions (page 2, lines 31-33) and the underlying reasons (page 2, lines 33-34), as explained in the discussion of the relevant paper. The validation of this data set and its use in other studies (see page 3, lines 35-38) shows that this limitation does not necessarily invalidate our conclusions. It rather limits the scope of the conclusions that can be drawn regarding the sources of the total aerosol load over the region, something that we have already acknowledged.

*The authors then average the decadal cloud microphysical data by month to look at seasonal trends, and find no clear seasonal trend in aerosol optical depth even as there are pronounced increases in liquid water path, cloud fraction, cloud optical thickness, and effective radius during the November-December time period.*

The reviewer has apparently not read the paper correctly: there is a clear (and strong) seasonal decrease in AOD in these months (see Fig. 3).

*From these relationships, they conclude that the observed seasonal trends are inconsistent with the first and second aerosol-cloud indirect effects, but possibly consistent with the semi-direct effect (Section 3.3.2). Such strong conclusions are not supported by the underlying data, which are themselves highly averaged in both space and time. The highly-averaged nature of the data makes it hard to draw conclusions other than to say that the seasonal trend of one variable appears to*

*correlate with the trend of another variable or that one or more variables trend up/down slightly over time – such apparent correlations are neither causal nor attributive.*

Any study of long-term changes will have to use averaged data. We have used standard monthly-averaged satellite products, which are publicly available and have been used in many other studies as well. The statement of the reviewer implies that these monthly-averaged products make no sense and studies using these data are flawed. In our case, we do not find 'variables trending slightly up/down over time' but very strong changes over the decade studied. These strong changes cannot possibly be caused by the particular ways in which the data have been averaged.

*No statistical tests are presented to quantify the robustness or strength of such correlations.*

How can the reviewer state this? As written on page 4, line 14: 'Statistical significance of all calculated changes was estimated using the two-sided t-test.' The results of these tests appear in many parts of the manuscript: e.g., page 5 line 6 (AOD), page 6 lines 4-5 (liquid CFC and LWP), page 6 lines 33-34 (seasonal cloud property changes), page 7 lines 8-10 (geopotential height and surface pressure: no significant changes), page 8 line 9 (seasonal AOD), page 8 lines 21-22 (seasonal smoke/polluted dust).

*Indeed, I find all of the authors' conclusions regarding the attribution of aerosol sources to biomass burning and their effects on clouds to be highly speculative. To quote the authors (Pg. 9,Lines19-21): "These results do not constitute evidence of any cause and effect mechanism, which cannot be proved based on observations only. They rather represent a contribution to the observational approaches in aerosol-cloud-radiation interaction studies, highlighting both the possibilities and limitations of these approaches." From reading this paper, I don't know what approaches are being referred to here. The approach employed seems to have been to take a bunch of Level 3 temporally-averaged and gridded satellite data products, plot them up, and draw strong, unsupported conclusions about aerosol-cloud interactions based on perceived annual or seasonal trends.*

We have analysed observed changes in aerosols and clouds over southern China, and outlined possible aerosol-cloud interaction mechanisms that can explain these changes: not more but also not less. We have made a careful attempt to indicate limitations of an observation-based study. However, the reviewer characterizes the study as "highly speculative", then quotes a part where we acknowledge the limitation of our analysis in drawing strong conclusions, and then criticizes the study for drawing "strong conclusions". This is an obviously inadequate comment, that does not do justice to the study.

*In my opinion, this paper does not represent a substantial contribution to scientific progress, which is the minimum criterion for the ACP scientific significance review criteria. As I do not see a path forward by which this manuscript could be revised to be a significant contribution, I recommend to the editor that this manuscript be rejected.*

This review does not mention any concrete and specific flaw in our study. In contrast, it seems that the reviewer only glanced through the manuscript to reach an unsupported negative judgement. It is unfortunate that the lack of even a general suggestion for a possible way forward that could lead to any improvement in this study only corroborates this conclusion.

---

## Referee Comment (RC3) · Anonymous Referee #1 · 28 Nov 2018

thanks for the detailed responses to the reviews

all questions were addressed and most reponses are reasonable

I wiill be waiting for the updated version

---

## Referee Comment (RC4) · Anonymous Referee #2 · 12 Dec 2018

| To: | Dr. Toshihiko Takemura, Editor, *Atmospheric Chemistry and Physics* |
| --- | --- |
| From: | Anonymous Referee # 2 |
| Date: | 10 December 2018 |
| Subject: | Further explanation of recommendation to reject "Satellite Observations of Aerosols and Clouds over Southern China From 2006 to 2015: Analysis of Changes and Possible Interaction Mechanisms" for publication in *Atmospheric Chemistry and Physics*. |

In your email dated 04 December 2018, you advised that the authors' reply to my initial review of their paper did not mesh with the content of my review. You asked me to comment on the manuscript again and to more concretely indicate relevant sentences and figures supporting my initial concerns with the paper and to reply to the authors' comments. I have re-read the manuscript, my first review, and my notes from the first review. Based on this re-review, my recommendation to reject the manuscript is unchanged. In this memo, I provide the specific details that underpin this recommendation.

The fundamental flaw that I find with this paper is that the data and analyses presented in the manuscript do not support the conclusions. It is also implied that some variables must be related because they trend together, but no causal relationship is established. Statistical tests are mentioned as having been performed, but no statistics such as p-values establishing significance or correlation coefficients establishing correlation (except for the minor case discussion on Pg. 9, Line 2) are presented. The authors are good about owning up to the limitations of the data, which is appreciated. However, acknowledging the limitations doesn't make them go away or make the dataset better suited for addressing the chosen science questions. Additional depth of analysis is needed to overcome these limitations in order to draw meaningful conclusions.

Here, I highlight my concerns with specific manuscript statements (italicized and quoted). Because of the extensive nature of these concerns, I do not think that the paper should be published without new data and analyses and extensive re-writing. Since new data and a considerable increase in the depth of analysis is required beyond that typically encountered for a major revision, I recommended the paper be rejected. I did not reach this recommendation lightly and only after a thorough reading, and now re-review, of the manuscript.

*Pg. 1, Lines 10-12: "The results show a decrease in aerosol optical depth over the study area by about 20% on average, accompanied by an increase in liquid cloud cover and cloud liquid water path (LWP) by 5% and 13%, respectively."*

[Figure]

This statement in the abstract follows from Fig. 2 and Fig. 5 (panels above). It is important to see predictive statistics associated with these trend lines rather than just the percent changes. There's quite a bit of scatter in the data, some of which may be seasonal variability, some of which may be interannual variability, and then there's the uncertainty of the Level 3 data product itself. What are the slopes of the dotted lines ($yr^{-1}$)? What are the p-values for the statistical test that one can reject the null hypothesis that the slope of the dotted line is zero?

There's also quite a bit of day-to-day and sub-pixel variability that is not reflected in the Level 3 gridded monthly mean product as well as different numbers of measurements (i.e., samples) in each pixel that need to be considered and this is not really discussed in the manuscript. Some mention of area weighting of pixels is given on Pg. 3, Line 38, but is not described; how was this done? The monthly-averaged CALIPSO observations also have different numbers of observations that are averaged to yield the reported gridded mean and standard deviation – in addition to the area weighting, were these differences in number of samples accounted for when averaging across the region or across different months/years?

It would helpful for the reader to see the Level 3 standard deviations on these trendline graphs as error bars or as a shaded region. How is this additional sub-month, sub-grid-cell variability being captured in the statistical tests to assess whether or not there is a trend? Assuming there are indeed, statistically-significant trends (which I don't think hasn't been discussed very extensively at all) is the trend in AOD is related to the trend in CFC or LWP or are they coincidental? The italics statement above from the abstract implies that there is a non-coincidental relationship.

*Pg. 1, Line 12-13: "Analysis of aerosol types and emissions suggests that the main driver for their reduction is a decrease in biomass burning aerosols. These changes occurred mainly in late autumn and early winter months..."*

The fundamental flaw with this conclusion is that it is not clear that all aerosol types have been captured, so one cannot say that the "main driver" of AOD trends over the last decade is biomass burning or continental pollution or marine aerosol or other types, because only the three aerosol types are included in the Level 3 CALIPSO product (dust, smoke, and polluted dust), and critical information about the trends of these other aerosol types is lacking. One must also ask the question, are the CALIPSO aerosol types sufficient to answer the question that's being posed, or does one need more specificity with regard to aerosol composition (e.g., sulfate, organics, dust, black carbon) that must be obtained from a model? Therefore, I would characterize this conclusion as unsupported by the underlying data and highly speculative. One way to address this criticism would be to not use the Level 3 data, but rather to use the Level 2 data that has more aerosol type classifications. Another approach would be to use model data products to explore this research question. Of course there would be uncertainties associated with any aerosol type classification scheme that would make it difficult to compare across different data sets – for example, the CALIPSO smoke aerosol probably is not only associated with biomass burning and also includes the contribution of other anthropogenic combustion sources. Another advantage of using the Level 2 data products is that they are not gridded and temporally averaged, so they capture a truer range of measured variability. This helps avoid biases, because the mean of the means is not always the same as the mean of the population if sample sizes are not constant and this unequal weighting is not accounted for properly.

Putting aside the major flaw of the missing aerosol types, it is also hard for me to see the trends in the data that the authors are using as a basis for saying that AOD changes occur in late Autumn and early Spring. From Fig. 1a and Fig.3 (shown at right), the peak in biomass burning in GFED is apparent between Nov.-Mar., while the ΔAOD traces vary quite a bit but don't really peak in this period. There is some decrease (the traces are below zero), but there is also a good bit of scatter in the data. There are no metrics of statistical variability included in this graph – only the means – so it is hard for me to assess the statistical significance of the data. The authors say they did t-tests, but on what? Where are these statistical results presented? Statements are made in multiple places that trends are statistically significant but no p-values are provided. Where variables are thought to be correlated (as in the case of biomass burning emissions and AOD), there are no correlation coefficients provided. I see this lack of scientific rigor as a major flaw in this study. It also led me to comment that most of the correlations suggested are determined by whether one or more variables trend up/down together over time, which I guess is determined visually. Having some numbers here related to the statistics, I think, is very important.

[Figure]

*Pg. 1, Line 13: "These changes occurred mainly in late autumn and early winter months and coincided with changes in cloud properties."*

The panels from Figs. 3 and 7 shown at right on this page indicate the seasonal variation in changes of AOD (top) and cloud properties (bottom three panels). There is a very clear and distinct change in cloud properties in Nov.-Dec. that does not appear to be related to the changes in AOD during this period. I don't understand the basis for the italicized statement made in the abstract that changes in AOD "coincided with changes in cloud properties".

*Pg. 1, Line 15-17: "Further analysis of changes in aerosol vertical profiles demonstrates a consistency of the observed aerosol and cloud changes with the aerosol semi-direct effect, which depends on their relative heights. Based on this mechanism, fewer absorbing aerosols in the cloud layer would lead to an overall decrease in evaporation of cloud droplets, thus increasing cloud LWP and cover."*

The semi-direct posits that solar heating of above-cloud absorbing aerosol layers changes the temperature profile of the atmosphere, reducing buoyancy, and ultimately cloud cover and liquid water path. To be consistent, then, with the semi-direct effect, I would expect to see an inverse correlation between absorbing aerosols above cloud and these cloud properties. What is shown in Figure 8 are monthly-averaged differences in the vertical profile of aerosol extinction as well as the vertical profile of cloud extinction. First, extinction is not absorption. Even relatively close to fires, the scattering-to-extinction ratio is > 0.8 (e.g., Yokelson et al., Atmos. Chem. Phys., 2009; https://doi.org/10.5194/acp-9-5785-2009), and it is known that the ratio is much higher as the smoke plumes age. No data is being presented regarding smoke age, whether or not the smoke is from urban pollution or biomass burning, or that there is or isn't any trend in absorbing aerosols over this region.

[Figure]

Second, the CALIPSO Level 3 data typing algorithm identifies smoke only when the layer is elevated – by definition! Therefore, it is not appropriate to use the positioning of this smoke product to suggest that there is some sort of vertical relationship with cloud. The smoke classification type shares many similar features to the polluted continental classification type, except that the latter is at the surface and not elevated. The polluted continental classification type has not been considered in the present analysis, which is a major gap in the analysis. Finally, I don't understand the relevance of the ISCCP classification types to this discussion – this classification scheme seems much too coarse to be meaningful. In sum, I see no conclusive evidence that aerosol changes are altering the temperature profile of the atmosphere to effect changes in clouds. Consequently, I don't think that it's appropriate for the authors to suggest that the semi-direct effect is a causal mechanism for the observed, 5-13% increase in LWP and cloud fraction from 2006-2015.

*Pg. 4, Line 32-33: "While it was not possible to pinpoint specific reasons for the March-April differences based on the data sets used here, this feature deserves further investigation."*

Why was it not possible to explore this discrepancy? How would further investigation be carried out? This is a very shallow approach to analyzing the data.

*Pg. 4, Line 35-36: "According to the CALIPSO classification, smoke aerosols originate in biomass burning activities…"*

Biomass burning aerosols do contribute to smoke layers, but so do other sources of combustion. Similarly, biomass burning, urban pollution, and fossil fuel combustion aerosols contribute to the polluted continental aerosol type (which is not accounted for in this study). A key difference between the CALIPSO smoke and polluted continental aerosol types is whether or not the layer is at the surface or elevated. Since the aerosol classification types are based on aerosol intensive and extensive parameters, there can be misclassification and some ambiguity across aerosol types, particularly for categories dominated by smoke and urban pollution because both types of aerosol are dominated by relatively small, non-depolarizing aerosols. The polluted dust category isn't necessarily a mix of biomass burning and dust – it represents the middle part of the continuum between smoke/continental-pollution (small and weakly depolarizing) and dust (large and strongly depolarizing). The satellite aerosol-typing products are very useful, but they are not unambiguous. This statement is too strong and not supported by the data.

*Pg. 4, Lines 36-37: "Biomass burning emissions and satellite-based AOD are not directly comparable."*

I agree with the authors' statement here, and yet, Figure 1 and Figure 3 attempt to make precisely this comparison.

*Pg. 5, Lines 7-8: "Based on the CALIPSO aerosol types classification, this decrease can be attributed to corresponding reductions in polluted dust and smoke aerosols"*

It is true that fitted lines to both polluted dust and smoke aerosols trend down during this period along with the overall AOD. However, it is unclear what the trend in continental pollution or marine aerosols are for this period because they have not been considered by this study. Certainly, decreases in polluted dust and smoke contribute to the decrease in AOD, but I don't think that the authors can "attribute" the change to only these two aerosol types when there are other types that are not being considered.

*Pg. 5, Lines 14-15: "Based on CALIPSO, this decrease [in AOD] is driven by biomass burning aerosols: as for the full time series (Fig. 2c), dust aerosols show no significant change."*

What evidence is there that the smoke and polluted dust aerosol types are dominated by biomass burning aerosol versus other sources of combustion or pollution aerosols? The CALIPSO aerosol type is not specific to biomass burning. Consequently, for the authors to make this conclusion, they need to provide some other evidence. Since no such evidence is apparent in this manuscript, this seems highly speculative.

*Pg. 5, Line 15-18: "analysis of the total mass of carbon particles (C) from local emissions (Fig. 3c) shows that the largest decrease in emitted particles occurs during late autumn to early spring, with a minimum in November, suggesting that this decrease should be attributed to changes in residential energy sources, which peak during the same period."*

No data on residential energy sources are provided or discussed in this manuscript, so this statement is entirely speculative, and, frankly irrelevant to the present study. The previous studies cited in the next sentence are also not sufficient to support this statement, as they are not recent enough be cover the 2006-2015 time period in this study. Even if there was a decrease in residential biomass burning emissions starting in the 1990s, such a decrease does not necessarily extend to present day. This conclusion is unfounded.

*Pg. 5, Line 20- : "Furthermore, a direct comparison of changes in satellite-based AOD and surface emissions offers additional insights into the origins of these changes: the seasonal variation of changes in C emissions partially agrees with the total AOD change pattern, while this agreement improves in the case of polluted dust."*

Again, showing the same figure as before at right, it can be seen that there is no agreement between the change in AOD and the change in C emissions (delta-AOD even becomes positive in January, while delta-C is fairly constant). I'm not sure I understand what is being meant by the term, "partially agrees". It appears that during the seasons where delta-C reaches a local minimum and is fairly stable that both MODIS and CALIPSO delta-AOD are quite variable and not at a local minimum or maximum. Finally, is it even appropriate to be trying to establish this comparison, as it was already stated on Pg. 4, Lines 36-37: "Biomass burning emissions and satellite-based AOD are not directly comparable"?

*Pg. 5, Lines 23-26: "These results suggest that part of the aerosol load over the study area (especially smoke aerosols) is transported from neighboring regions, as was also inferred from differences in seasonality patterns (Fig. 1). In such cases, AOD and local emissions do not agree well (e.g., smoke aerosols in October). Forest fires and biomass burnig activities in Indochina could be such sources."*

What evidence is there to assert that the aerosols or smoke observed over this region is transported from neighboring regions such as Indochina (versus long-range transport or local emissions)? No data on fire activity in neighboring regions is presented, nor is any information on air mass back trajectories. What about the confounding influences of local, urban pollution and non-biomass combustion aerosols on the CALIPSO types? This italicized statement seems highly speculative.

[Figure]

*Pg. 5, Lines 29-33: "It should be noted here that misclassification of aerosol type in the CALIPSO data set does occur. In particular, smoke aerosols may be confused with other small aerosol types such as urban pollution (Burton et al., 2013)…Hence, some ambiguity probably exists regarding the origin of the aerosols, especially for the smoke aerosol type."*

I agree 100% with this statement. The problem is that this ambiguity undercuts many of the conclusions put forward in this manuscript. This is made even more problematic in that the continental pollution aerosol type is not included in this analysis. Given these uncertainties, a fundamental question that must be asked is, is this satellite-based AOD aerosol type data set appropriate to address source attribution? Given that some major aerosol types are missing, I think the answer is that it is not appropriate.

*Pg. 5, Lines 38-40:* A minor comment is that COT and REFF should be defined.

*Pg. 6, Lines 8-10: "Cloud changes appear statistically significant at the 95% level over large areas of the study region, especially over land, when studied on a pixel basis. Analysis of spatially averaged values, however, over the entire (5x10-degree) study region, reduces this significance to levels below 95% in most cases of Fig. 5."*

I'm struggling to interpret the meaning of this statement. Spatial distributions of the change in LWP and cloud fraction are presented that demonstrate large increases in LWP and cloud fraction over land and decreases over water. No indication is given in Figure 5 of the areas where these changes are or are not statistically significant (often I have seen this done with speckling overlaid on the statistically significant portions). It also sounds like from this statement that most of the cases in Fig. 5 are not statistically significant. Yet, these are the numbers that are being quoted in the abstract for changes in liquid cloud cover and liquid water path of "5% and 13%, respectively" (Pg. 1, Line 12) and for drawing other conclusions later on. Are these numbers statistically significant?

*Pg. 6, Lines 33-34: "The liquid CFC change is statistically significant in the November case, while all other cloud property changes shown in Fig. 7 are significant in December."*

Do I understand this statement correctly to say that there is only one month where each of the cloud property changes is statistically different from zero, and that only cloud fraction is statistically different from zero in November? Why would there be a trend in cloud properties in only a single month? Reading between the lines here, does this mean that the overall change in cloud property changes is not statistically significant, or is the effect size of this single month enough to drive the entire trend?

*Pg. 6, Lines 37-38: "The results presented in the previous section show that during the study period, aerosols decreased over southern China particularly in autumn and early winter, while liquid clouds increased mainly in late autumn and early winter. Hence, there is a concurrence of substantial aerosol and cloud changes during the same months, namely in late autumn and early winter."*

I don't think this statement is correct. Liquid clouds increased only in November on a cloud fraction basis and only in December on a LWP or COT basis (based on the statistical significance discussion in the previous comment). The seasonal pattern of the AOD changes and the cloud properties changes are not similar (either correlated or anti-correlated). This statement suggests that they are anti-correlated, which is not true.

*Pg. 7, Line 10-11: "These results suggest that meteorological variability is not among the major factors contributing to the aerosol and cloud changes reported."*

First, statistical significance of the cloud changes is unclear (discussed previously). There does appear to be a decreasing trend in AOD, which the authors assert is statistically significant. That surface pressure and geopotential height do not show a statistically significant trend is insufficient to rule out meteorological drivers. The atmospheric temperature profile, moisture, and lower tropospheric stability are also important variables that do not appear to have been considered. Even if these variables fail to demonstrate a statistically significant trend that does not in and of itself rule out the existence of such a trend. All it means is that the available data are insufficient to reject the null hypothesis, but there may indeed be a trend that might be uncovered by additional data and/or a longer timeseries. The italicized statement is not demonstrated conclusively by the data presented, which are rather superficial.

*Pgs. 7-8, Section 3.3.2 "Possible effects of ACIs and ARIs:"; "our results appear to be inconsistent with the standard definition of the first and second indirect effects, although the possibility of multiple mechanisms occurring simultaneously cannot be excluded."; "Contrary to the first and second aerosol indirect effects, the semi-direct effect cannot be excluded as an explanatory process, since the signs of changes of all aerosol and cloud variables presented here are consistent with what would be expected based on this mechanism."; "It is important noting that this mechanism holds primarily for absorbing aerosols, such as biomass burning particles, which is the case in this study. It is also important noting that the position of the aerosols relative to the cloud layer determines the sign of the semi-direct effect: a decrease in aerosols will lead to increased cloudiness only if the aerosols are at the same level with clouds".*

I think that this paragraph is not at all supported by the underlying data, which until this point has focused on trends and changes over time. In this paragraph, process-level explanations are invoked, but are done at a highly-averaged level spanning months and 5 x 10 degree area. These are not the scales at which aerosol-cloud interactions would be expected to be evident (e.g., McComiskey and Feingold, ACP, 2012, https://www.atmos-chem-phys.net/12/1031/2012/), so the failure to see ACI effects in the trend data is not surprising. Saying that the authors' findings are "inconsistent" with the first and second indirect effects is too strong a statement. These effects may very be visible in this region if a more appropriate data set is used (e.g., aircraft, balloon, surface remote sensor scales measuring clouds over

minutes to hours or a model with better space and time resolution). One cannot know. The same is true for the discussion on the semi-direct effect, with the additional comments that have been described above in this review that the attribution of the particles to biomass burning, as absorbing particles, and that these particles are at cloud level are all not established by this data set. In fact, the CALIPSO smoke type is not unambiguous as a marker for biomass burning. That the smoke type often occurs near cloud level is unsurprising given that the layer must be elevated by definition of that aerosol type in the CALIPSO scheme. Similarly, the continental pollution aerosol type is very similar to the smoke type, but it is not in an elevated layer. Since the data set used in this manuscript and its analyses is so highly averaged in space and time, it is of little utility for discussing ACI effects. Consequently, the conclusions as stated are not definitive and this entire paragraph should be removed.

*Pg. 8, Line 10: "Figure 8a shows the typical profile of cloud extinction in autumn over southern China…"*

Are the aerosol and cloud profiles shown in Figure 8 an average profile or an individual, typical profile for each? What is meant by "autumn" or "Fall" for the cloud extinction profile – both October and November? If they are averaged profiles, how was that averaging carried out (e.g., was a weighted average of the sample numbers in each pixel used)? Are there meaningful differences in the profiles across the spatial area? A single set of averaged profiles over the entire spatial domain seem difficult to meaningfully interpret to me, as I would expect these profile changes to be very different over land and over water. How should the reader interpret these profiles with regard to representativeness?

Are the changes in Fig. 8b and 8c statistically significant at all height levels? The commentary on Pg. 8, Line 21-23 suggests that only certain layers are statistically significant and in different months (e.g., 1-1.5 km altitude for smoke in October and 0.7-1.2km for polluted dust in November). It would appear that the CALIPSO smoke aerosol change is not statistically significant in November when the cloud fraction change is statistically significant. Conversely, the smoke change is statistically significant in October when the cloud fraction change is not statistically significant. What about the December profiles where the other cloud property changes are statistically significant? It is very difficult to unravel what is being presented here, but it certainly does not seem to be suggestive of an aerosol-cloud semi-direct effect (as is stated on Pg. 1, Lines 15-18).

*Figure 9 and related discussion.*

Why is necessary to break out the cloud optical thickness data by cloud type? It has already been established from Figure 7 and associated discussion that delta-COT is not statistically significant in October or November. This is mentioned in passing on Pg. 8, Line 28. Yet, there is then extensive discussion on the coincidence between decreased biomass burning, increased liquid cloud fraction and water content in November and a decrease in smoke aerosols in October (Pg. 8, Lines 28-32). I find this discussion very confusing, but much of it appears to be based on source attribution that has already been discussed in this review as being speculative. The ISCCP cloud type classifications does not bring any additional clarity or information to the major flaws in the prior conclusions.

*Pg. 8, Lines 40 – Pg. 9, Line 1: "It should be noted here, as was also mentioned in Section 3.1, that possible misclassifications in CALIPSO aerosol types add ambiguity to our conclusions regarding the origin of these aerosols loads, especially smoke aerosols. They would not affect, however, our findings regarding possible interaction mechanisms."*

I fundamentally disagree with second part of this statement. The manuscript is saying that smoke aerosols are from biomass burning and are absorbing, and therefore, an association between smoke aerosol height and cloud height in Figure 8 is somehow related to the semi-direct effect. If the aerosols are misclassified or smoke dominated urban aerosols that are weakly absorbing then this would indeed call the conclusions of the manuscript into question. The missing CALIPSO aerosol types and the compositional ambiguity provided by this typing method make this particular dataset less capable for addressing the types of science questions and drawing the types of conclusions that are sought in this manuscript. This limitation is a significant one that cannot be overcome without new data and new analyses.

*Pg. 9, Lines 1-4: "Further analysis of the monthly time series showed that the liquid CFC and the polluted dust AOD are anti-correlated in November and December, with correlation coefficients around -0.7 to -0.8. This anti-correlation is not apparent in other months and the decreasing pattern in the polluted dust profile in November is not present for the other aerosol types or months.*

[Figure]

Here are the monthly timeseries from Fig. 1 and Fig. 4. I think that it's hard to make the case as is done here that there is a meaningful anti-correlation between polluted dust AOD and liquid CFC only in November and December (circled regions) and that such an anti-correlation can be used to draw a conclusion. There is just too much variability (and that's before the additional requested information of standard deviation error bars or shaded regions are added to the graph). The decreasing pattern in the polluted dust aerosol is present for the smoke aerosol and does occur during other months. Given the scale, it's difficult to discern the trend for the dust trace.

*Pg. 9, Line 8-10: "it was found that absorbing aerosol loads over the region decreased significantly, and this decrease was attributed mainly to changes in biomass burning activities."*

No data on aerosol absorption are presented in this study. Some CALIPSO aerosol types are missing, and those that are included cannot be unambiguously attributed to biomass burning emissions. This conclusion is not true.

*Pg. 9, Line 10-11: "Concurrent changes in liquid cloud fraction and thickness were observed, with notable increases and decreases in different months."*

Technically this statement just says that cloud fraction and thickness both change across months. The changes are neither correlated nor anti-correlated (Figure 4a and 4c). At best, this sentence doesn't reach a meaningful conclusion, and at worst it misleads the reader into thinking that concurrent changes somehow track each other. I recommend that this sentence be removed.

*Pg. 9, Line 11-13: "Further analysis of vertical profiles of both aerosols and clouds showed that the signs of cloud changes depended on the position of aerosols relative to clouds, being in agreement with the predictions of the aerosol semi-direct effect, under different aerosol and cloud configurations."*

No data actually relating the position of aerosols to clouds is presented. Instead, the vertical distribution of cloud extinction and the vertical distribution of aerosol temporal change are presented in Figure 8. These are not the same thing. It is not shown that the sign of cloud changes is determined by the position of aerosol relative to clouds. This statement is not true.

*Pg. 9, Lines 17-18: "Here, the combined analysis of different aerosol and cloud data sets showed a high level of consistency with predictions of [the semi-direct effect]."*

It is very difficult to relate the trend analysis changes for the aerosol AOD and cloud changes to a process-level causal mechanism. Even comparing the trend changes of aerosol and cloud in this manuscript are difficult because they appear to vary differently across monthly and with different (and poorly explained) levels of statistical significance. It is not true to suggest that the data show a "high level of consistency" with the semi-direct or any ACI effect. This is because the data used here are highly averaged in both space and time, and are, therefore, less ideal for tackling these sorts of science questions.

*Pg. 9, Lines 18-20: "It should be stressed however, that apart from strong indications, these results do not constitute evidence of any cause and effect mechanism, which cannot be proved based on observations only. They rather represent a contribution to the observational approaches in aerosol-cloud-radiation interaction studies, highlighting both the possibilities and limitations of these approaches. To overcome some of these limitations, further research will focus on model simulations of the conditions described here, in order to provide more insights regarding the underlying physical mechanism."*

This statement succinctly highlights the lack of depth of the analyses in this manuscript. There are a large number of critical limitations associated with the use of these data to try to draw these sorts of conclusions. The limitations are acknowledged in the manuscript, but there is no real attempt to overcome them. The model analyses discussed in this last sentence hold promise for being able to attribute aerosol to sources as well as to link those aerosol to clouds. I recommend that those tools be brought to bear on the questions being tackled here, perhaps with some context being provided by more complete set of aerosol types in the Level 2 version of the satellite data that is discussed here.

Authors' Comments (in italics) and reviewer's reply (in plain text):

1) *Multiple pieces of information are provided that are suggestive of the authors' conclusion that AOD changes are related to carbon emissions from burning, and the authors' conclusion is supported by a number of previous studies.*

   While the data may be suggestive, there are some significant gaps (e.g., missing aerosol types, lack of unambiguity in aerosol typing) that prevent direct attribution of AOD changes to specific aerosol emissions sources. If previous studies have shown a link between biomass burning and AOD changes, then it is not clear to me how this paper meaningfully contributes to that body of work.

2) *The manuscript acknowledges the limitations associated with the CALIPSO Level 3 data. Because this data has been validated and used by other papers, the limitations do not necessarily invalidate their conclusions. It just limits the scope of conclusions that can be drawn regarding the sources of the aerosol load over the region.*

   My concerns are not with the underlying data, which are useful for many different purposes. I do not think that these data are sufficient, however, to draw the conclusions that are presented in this paper. I appreciate that the authors have been upfront in acknowledging the data limitations; perhaps a future study employing a model or additional datasets/tools could move beyond just identifying them to overcoming them.

3) *The reviewer read the paper incorrectly when commenting that the seasonal trend in AOD does not track with the cloud properties during the November-December time period. There is a strong decrease in AOD during these months.*

   Yes, but the seasonality of delta-AOD and delta-Cloud Properties do not look particularly similar. I also don't know what is a strong vs. weak decrease in AOD. Some additional discussion on what these changes are (I assume the decadal change) and what the annual trend slopes are would be helpful for elucidating strong vs. weak changes.

4) *The averaged data are standard and have been used by many studies. This paper finds strong changes that cannot be caused by data averaging.*

   Again, the strength of the changes and their statistical significance is not clear to me from the manuscript. The Level 3 data has more than just the mean values, and includes standard deviations and numbers of observations. These need to be accounted for in the statistics. My concerns are not with the data themselves, but rather with the conclusions that are being drawn using them.

5) *A two-sided t-test was used to calculate the statistical significance of all calculated changes. Statements in the text that the results being discussed are statistically significant are made at multiple places throughout the manuscript.*

It would be good to include p-values and correlation coefficients (and their p-values) to support the qualitative statements that are being made in the text. Is the t-test being applied to the 2006 data and the 2015 data to evaluate a difference or is it being applied to the entire timeseries? Some additional detail on the trend analysis and statistical methodology would be welcome here so that these details are not easily overlooked.

6) *Characterizing the conclusion regarding the attribution of aerosol sources to biomass burning in the manuscript as "highly speculative" and the study more broadly as drawing "strong, unsupported conclusions" is unfair. The authors have been careful in indicating limitations.*

I appreciate that the authors have been careful in indicating limitations. I do think that the limitations limit the ability to draw conclusions based on this dataset. Consequently, strong conclusions like those detailed above are going to be at least somewhat speculative without better data or tools for source attribution and unambiguous aerosol typing.

7) *No concrete or specific flaws are mentioned. A path toward publication is not articulated.*

I hope that this re-review provides an appropriate level of specificity. While I do not see a path toward publication for this particular manuscript, I think that a study employing preferably both a model and Level 2 satellite data (with the complete set of aerosol types) to look at aerosol trends and the source attribution of those trends to specific emissions sources would be a very worthwhile paper. However, using that type of data to try to establish a connection to long-term trends in cloud properties and trying to relate cloud trends to ACI effects seems like a step too far to me given the known scale issues in identifying and quantifying ACIs.

---

## Editor Comment (EC1) · Takemura (Editor) · 18 Dec 2018

Referee #2 gives critical and concrete comments on methods and analyses in this study. Before submitting the revised manuscript, the authors should post responses to the referee's comments one by one. If it is difficult to response reasonably and theoretically or if the responses will be beyond revision of the manuscript, I recommend the authors to withdraw the review process.
* * *

---

## Author Comment (AC3) · 16 Jan 2019

We thank the Anonymous Referee #2 for this second review. In contrast to the first review it gives concrete criticism, which we can respond to. Most comments are either based on misunderstanding and/or can be addressed by providing additional information. However, some comments are plainly wrong. Specifically, the referee mentions a wrong range of months to dismiss some of our findings (comment on page 1, lines 12-13 of the manuscript), describes wrongly the semi-direct effect for aerosols above clouds, thus expecting results opposite to our findings (comment on page 1, lines 15-17), and refers to irrelevant figures to disprove some of our results (comments on page 9, lines 1-4 and page 9, lines 10-11). Following are our point-by-point replies with the referee comments in italic.

*Pg. 1, Lines 10-12: This statement in the abstract follows from Fig. 2 and Fig. 5 (panels above). It is important to see predictive statistics associated with these trend lines rather than just the percent changes. There's quite a bit of scatter in the data, some of which may be seasonal variability, some of which may be interannual variability, and then there's the uncertainty of the Level 3 data product itself. What are the slopes of the dotted lines (yr$_{-1}$)? What are the p-values for the statistical test that one can reject the null hypothesis that the slope of the dotted line is zero?*

Figures 2 and 5 provide information on the statistical significance of the plotted lines in the 95% confidence intervals, with corresponding percent changes highlighted in bold. This is explained in the caption of Fig. 2. It was omitted from Fig. 5 caption, but it should indeed be included for clarity. Regarding Fig. 5, it is also clearly stated (page 6, line 10) that in most of the cases the statistical significance level is below the 95% confidence interval. This is the reason why the analysis proceeds further to monthly changes, where the seasonal variability is removed. Reporting percent changes instead of (absolute) slopes was selected as a more intuitive measure of change. Regarding p-values, a table could also be added for completeness. The following table provides the information requested by the referee:

| Parameter | Unit | CALIPSO | MODIS | CLARA-A2 |
|---|---|---|---|---|
| | | Change (%)/slope ( yr$^{-1}$)/p-value | Change (%)/slope ( yr$^{-1}$)/p-values | change (%)/slope ( yr$^{-1}$)/p-value |
| Total AOD | 1 | -23.3/-0.013/0.013 | -17.6/-0.010/0.002 | |
| Dust AOD | 1 | +8.4/0.0003/0.797 | | |
| Smoke AOD | 1 | -22.5/-0.006/0.071 | | |
| Polluted Dust AOD | 1 | -33.5/-0.008/0.003 | | |
| | | | | |
| All-sky LWP | g m$^{-2}$ | | +12.4/0.837/0.204 | +14.2/0.913/0.242 |
| Liquid CFC | 1 | | +6.8/0.003/0.219 | +3.4/0.002/0.465 |
| Liquid COT | 1 | | +5.5/0.089/0.399 | +3.6/0.058/0.607 |
| Liquid REFF | μm | | +1.6/0.018/0.239 | +5.2/0.034/0.0003 |

The following figure depicts, on a pixel basis, the level of statistical significance for MODIS AOD changes (corresponding to Fig. 2a). For similar maps corresponding to the changes shown in Fig. 5a and 5b, the reviewer is referred to one of our later replies (page 11 of this document).

[Figure]

*There's also quite a bit of day-to-day and sub-pixel variability that is not reflected in the Level 3 gridded monthly mean product as well as different numbers of measurements (i.e., samples) in each pixel that need to be considered and this is not really discussed in the manuscript. Some mention of area weighting of pixels is given on Pg. 3, Line 38, but is not described; how was this done? The monthly-averaged CALIPSO observations also have different numbers of observations that are averaged to yield the reported gridded mean and standard deviation – in addition to the area weighting, were these differences in number of samples accounted for when averaging across the region or across different months/years?*

The area weighting mentioned by the referee concerns the differences in surface areas of grid boxes due to different latitudes. Because of the small size of the domain these differences are minor. This could be clarified in the statement of page 3, line 38. The different number of observations was accounted for by applying a threshold on the minimum number of days used in the monthly mean calculation (on a pixel basis) before estimating the spatial average (see also Section 2.4). In the case of CALIPSO, averages were weighted by the number of samples used, which is available in the level 3 data. Data sets from different sources will of course have different numbers of observations being averaged, as the referee mentions. The same concern led us to apply the thresholds described in Section 2.4, in order to minimize ensuing discrepancies. While we agree that sub-pixel variability is not reflected in the gridded monthly mean products, we consider that some rephrasing could answer the referee points previously mentioned.

*It would helpful for the reader to see the Level 3 standard deviations on these trendline graphs as error bars or as a shaded region. How is this additional sub-month, sub-grid-cell variability being captured in the statistical tests to assess whether or not there is a trend? Assuming there are indeed, statistically-significant trends (which I don't think hasn't been discussed very extensively at all) is the trend in AOD is related to the trend in CFC or LWP or are they coincidental? The italics statement above from the abstract implies that there is a non-coincidental relationship.*

The requested information could be added in the graphs. While we considered that the statistical significance of trends was adequately discussed, this discussion could also be extended, with an addition of a relevant table of p-values, as shown before. However, based on the present discussion in the paper, it is explicitly mentioned that some changes are statistically significant and some are not, as was also explained in our first reply. Hence, we don't understand why the referee would still "assume" that "there are indeed, statistically significant trends". Regarding the question on the relation or coincidence of changes in aerosols and clouds, it is one of the main science questions of this study, as described in the Introduction (page 2, lines 11-12), and we attempt to address it based on the analysis described in Section 3.3. Our results imply that there is indeed a non-coincidental relationship.

*Pg. 1, Line 12-13: The fundamental flaw with this conclusion is that it is not clear that all aerosol types have been captured, so one cannot say that the "main driver" of AOD trends over the last decade is biomass burning or continental pollution or marine aerosol or other types, because only the three aerosol types are included in the Level 3 CALIPSO product (dust, smoke, and polluted dust), and critical information about the trends of these other aerosol types is lacking. One must also ask the question, are the CALIPSO aerosol types sufficient to answer the question that's being posed, or does one need more specificity with regard to aerosol composition (e.g., sulfate, organics, dust, black carbon) that must be obtained from a model? Therefore, I would characterize this conclusion as unsupported by the underlying data and highly speculative. One way to address this criticism would be to not use the Level 3 data, but rather to use the Level 2 data that has more aerosol type classifications. Another approach would be to use model data products to explore this research question. Of course there would be uncertainties associated with any aerosol type classification scheme that would make it difficult to compare across different data sets – for example, the CALIPSO smoke aerosol probably is not only associated with biomass burning and also includes the contribution of other anthropogenic combustion sources. Another advantage of using the Level 2 data products is that they are not gridded and temporally averaged, so they capture a truer range of measured variability. This helps avoid biases, because the mean of the means is not always the same as the mean of the population if sample sizes are not constant and this unequal weighting is not accounted for properly.*

It is true that the lack of the full set of aerosol subtypes is prohibitive for the attribution of their overall decrease to a "main driver". This expression should be corrected accordingly. Our results, however, show a statistically significant decrease in total AOD and in an aerosol subtype. This may indeed not be the main driver, but the lacking information on how some other subtypes change is not "critical" for further analyzing this specific subtype. Regarding the sufficiency of the CALIPSO data used "to answer

the question that's being posed", and based on the referee's suggestions (level 2 data, model data) and some later comments, it seems that there is a misunderstanding on the reason of using CALIPSO in this study, which should be clarified also in the manuscript: it is neither to attribute aerosol sources, nor to unveil ACIs in their process level. It is to include information on the vertical distribution of aerosols, and its possible changes, which is critical for their position relative to clouds. This is why, while the CALIPSO subtypes are analyzed in terms of changes and compared to other data sets (e.g. GFED), the ambiguity regarding their origin is repeatedly stressed throughout the manuscript.

Following a suggestion by referee #1 we have collected CEDS anthropogenic emission estimates compiled for CMIP6. As shown in the figure below, these estimates suggest that both the organic carbon and black carbon emissions cannot explain the decrease in AOT in southern China, because both show slight increases in the 2006-2014 period.

[Figure]

Regarding the referee's remark on the unequal weighting in averaging, as explained in our previous reply, the unequal number of observations is actually accounted for.

*Putting aside the major flaw of the missing aerosol types, it is also hard for me to see the trends in the data that the authors are using as a basis for saying that AOD changes occur in late Autumn and early Spring. From Fig. 1a and Fig.3 (shown at right), the peak in biomass burning in GFED is apparent between Nov.-Mar., while the ΔAOD traces vary quite a bit but don't really peak in this period. There is some decrease (the traces are below zero), but there is also a good bit of scatter in the data.*

The combination of plots provided here by the referee to support the above criticism actually compares the seasonal variation in biomass burning emissions from GFED with the *changes* in the seasonal variation of AOD and biomass burning emissions. It is not obvious why the maximum change in a parameter should also coincide with its maximum average value, as the referee seems to expect. It is also worth noting that the original statement, cited by the referee (page 1, lines 12-13) reads "changes occurred mainly in late autumn and early winter months*", not "early Spring*". In fact, nowhere in the

manuscript is said that "*AOD changes occur in late Autumn and early Spring*". The sentence cited by the referee mentions "early winter", and actually refers to AOD from MODIS and CALIPSO, and their changes in October, November and December (Figs 3a and 3b). We would be happy to rephrase this part in order to clarify it. However, the referee has drawn here a red box ranging from November to March, to show that (indeed) ΔAOD does not peak in this period. The same mistake is repeated in a later comment (page 9 in this document). Hence, this criticism is rather superficial and obviously unsupported.

*There are no metrics of statistical variability included in this graph (Fig. 3) – only the means – so it is hard for me to assess the statistical significance of the data. The authors say they did t-tests, but on what? Where are these statistical results presented? Statements are made in multiple places that trends are statistically significant but no p-values are provided. Where variables are thought to be correlated (as in the case of biomass burning emissions and AOD), there are no correlation coefficients provided. I see this lack of scientific rigor as a major flaw in this study. It also led me to comment that most of the correlations suggested are determined by whether one or more variables trend up/down together over time, which I guess is determined visually. Having some numbers here related to the statistics, I think, is very important.*

We appreciate the referee's request for more details on the metrics of statistical variability. There are indeed statements in multiple places that changes are statistically significant, with corresponding p-values not provided, and a concentrated report of corresponding metrics could help. However, describing the method used for the assessment of statistical significance, and then reporting results individually, is a rather common practice in similar cases and does not constitute "lack of scientific rigor". In fact, it is a matter of a simple revision for these metrics to be provided. However, the "visual determination" of correlations, mentioned here by the referee, does not qualify as a valid scientific approach. Hence it is surprising that this is the referee's "guess" regarding our methodology, and unfortunately it is not even accompanied by the "benefit of the doubt". In our opinion, such a serious statement in a review process should at least give to the authors the opportunity to disprove it, instead of leading directly to such a negative judgement. Nevertheless, we take the opportunity here and provide the requested metrics:

- Page 5, line 6: p-value=0.002 for MODIS, 0.013 for CALIPSO.
- Page 6, line 4: please refer to the maps provided here in page 11.
- Page 6, line 10: please refer to the table provided here in page 1.
- Page 6, lines 33-34: please refer to the table provided here in pages 11-12.
- Page 8, line 9: the referred decrease is not "significant" in a statistical sense (95% confidence interval). The term should be replaced to avoid misunderstandings.
- Page 9, line 9: p-value=0.03. This decrease refers to polluted dust aerosols, should be replaced for clarification.

The following table shows the Pearson's correlation coefficients, on a monthly basis, of each CALIPSO aerosol subtype with GFED emissions. Please note that nowhere in the text is the total AOD correlated

with GFED, as mentioned by the referee. Please also note the correlation between GFED and polluted dust AOD in November, which led to the discussion in page 5, lines 24-29 of the manuscript.

| | Dust AOD | Smoke AOD | Polluted dust AOD |
|---|---|---|---|
| January | -0.27 | -0.06 | -0.10 |
| February | -0.28 | 0.37 | 0.18 |
| March | 0.14 | 0.59 | 0.16 |
| April | -0.38 | -0.50 | 0.30 |
| May | 0.42 | -0.07 | 0.27 |
| June | -0.37 | 0.24 | -0.07 |
| July | -0.31 | 0.16 | -0.23 |
| August | 0.28 | 0.00 | 0.06 |
| September | -0.44 | -0.37 | -0.37 |
| October | -0.09 | -0.07 | 0.37 |
| November | -0.43 | -0.05 | 0.74 |
| December | -0.30 | 0.62 | 0.25 |

*Pg. 1, Line 13: The panels from Figs. 3 and 7 shown at right on this page indicate the seasonal variation in changes of AOD (top) and cloud properties (bottom three panels). There is a very clear and distinct change in cloud properties in Nov.-Dec. that does not appear to be related to the changes in AOD during this period. I don't understand the basis for the italicized statement made in the abstract that changes in AOD "coincided with changes in cloud properties".*

The term "late autumn and early winter months" should probably be replaced by "November and December" to clarify the issue. It should be obvious also from the red box that the referee has drawn that in these months there are changes in AOD and cloud properties that coincide. It is not clear what the referee means by "*does not appear to be related to the changes in AOD during this period*". The term "related" was not used in the statement that the referee cites, and no "relation" was established based on the plots that the referee has compiled. The fact that in other months (e.g. October) AOD and cloud property changes do not coincide, does not negate our statement. It is actually the main finding of our paper that the AOD changes in October occur at a higher level and thus have different effects on clouds.

*Pg. 1, Line 15-17: The semi-direct posits that solar heating of above-cloud absorbing aerosol layers changes the temperature profile of the atmosphere, reducing buoyancy, and ultimately cloud cover and liquid water path. To be consistent, then, with the semi-direct effect, I would expect to see an inverse correlation between absorbing aerosols above cloud and these cloud properties. What is shown in Figure 8 are monthly-averaged differences in the vertical profile of aerosol extinction as well as the vertical profile of cloud extinction. First, extinction is not absorption. Even relatively close to fires, the scattering-to-extinction ratio is > 0.8 (e.g., Yokelson et al., Atmos. Chem. Phys., 2009; https://doi.org/10.5194/acp-*

*9-5785-2009), and it is known that the ratio is much higher as the smoke plumes age. No data is being presented regarding smoke age, whether or not the smoke is from urban pollution or biomass burning, or that there is or isn't any trend in absorbing aerosols over this region.*

*Second, the CALIPSO Level 3 data typing algorithm identifies smoke only when the layer is elevated – by definition! Therefore, it is not appropriate to use the positioning of this smoke product to suggest that there is some sort of vertical relationship with cloud. The smoke classification type shares many similar features to the polluted continental classification type, except that the latter is at the surface and not elevated. The polluted continental classification type has not been considered in the present analysis, which is a major gap in the analysis. Finally, I don't understand the relevance of the ISCCP classification types to this discussion – this classification scheme seems much too coarse to be meaningful. In sum, I see no conclusive evidence that aerosol changes are altering the temperature profile of the atmosphere to effect changes in clouds. Consequently, I don't think that it's appropriate for the authors to suggest that the semi-direct effect is a causal mechanism for the observed, 5-13% increase in LWP and cloud fraction from 2006-2015.*

The referee's statement that the semi-direct "posits that solar heating of above-cloud absorbing aerosol layers changes the temperature profile of the atmosphere, reducing buoyancy, and ultimately cloud cover and liquid water path" contradicts the widely described semi-direct effect mechanism for absorbing aerosols above stratocumulus clouds (e.g., Koch and Del Genio, 2010). The buoyancy is indeed reduced but this will lead to less entrainment at the cloud top and consequently an **increase** in cloud cover / liquid water path. Hence, a decrease in absorbing aerosols above clouds would be consistent with a corresponding decrease in stratocumulus clouds below. This is exactly what is shown in Figs. 8b and 9b.

The purpose of Fig. 8b and c is to indicate at which height the changes in aerosol occur. The CALIPSO extinction profile suits that purpose. The fact that "extinction is not absorption" is irrelevant for the conclusions regarding the vertical location of the aerosol changes. We acknowledge in the manuscript that the CALISPO smoke classification is accompanied with some uncertainty. As a result we do not know for sure how absorbing these elevated aerosols are. However, for the polluted dust aerosols, showing largest decreases in November, we have solid indications that they are strongly absorbing because their decrease goes together with a decrease in GFED biomass burning emissions (Fig. 3c) while anthropogenic emissions did not show a decrease (see figure on page 4 of this reply).

The ISCCP classification was included to highlight the changes in low clouds for October and November. We don't see why this classification, which has been used widely in the past, does not serve this purpose.

*Pg. 4, Line 32-33: Why was it not possible to explore this discrepancy? How would further investigation be carried out? This is a very shallow approach to analyzing the data.*

Our statement reads "…*it was not possible to pinpoint specific reasons for the March-April differences based on the data sets used here*". Contrary to the referee's understanding, this statement denotes that this discrepancy was actually explored, based on the data sets used here, but no explanation was found. Hence, further investigation would require further analysis of additional data sets, focusing on these months. This would extend beyond the scope of this study, which focuses on the October and November months.

*Pg. 4, Line 35-36: Biomass burning aerosols do contribute to smoke layers, but so do other sources of combustion. Similarly, biomass burning, urban pollution, and fossil fuel combustion aerosols contribute to the polluted continental aerosol type (which is not accounted for in this study). A key difference between the CALIPSO smoke and polluted continental aerosol types is whether or not the layer is at the surface or elevated. Since the aerosol classification types are based on aerosol intensive and extensive parameters, there can be misclassification and some ambiguity across aerosol types, particularly for categories dominated by smoke and urban pollution because both types of aerosol are dominated by relatively small, non-depolarizing aerosols. The polluted dust category isn't necessarily a mix of biomass burning and dust – it represents the middle part of the continuum between smoke/continental-pollution (small and weakly depolarizing) and dust (large and strongly depolarizing). The satellite aerosol-typing products are very useful, but they are not unambiguous. This statement is too strong and not supported by the data.*

We thank the referee for these clarifications, which could be added in the relevant discussion along with the appropriate references. We also agree that the aerosol-typing products are useful, but they are not unambiguous. In fact, the ambiguity regarding especially the smoke aerosol type is explicitly stated in page 5, lines 31-32. This statement could also be rephrased in accordance with these ambiguities.

*Pg. 4, Lines 36-37: I agree with the authors' statement here, and yet, Figure 1 and Figure 3 attempt to make precisely this comparison.*

This statement, along with the next sentence (page 4, lines 37-39), explain why biomass burning emissions and satellite-based AOD are expected to differ (i.e. not being directly comparable). However, including them as subplots in Figs. 1 and 3 is useful, in our opinion, since the former can help explain the latter.

*Pg. 5, Lines 7-8: It is true that fitted lines to both polluted dust and smoke aerosols trend down during this period along with the overall AOD. However, it is unclear what the trend in continental pollution or marine aerosols are for this period because they have not been considered by this study. Certainly, decreases in polluted dust and smoke contribute to the decrease in AOD, but I don't think that the*

*authors can "attribute" the change to only these two aerosol types when there are other types that are not being considered.*

We understand that the term "can be attributed" may be misunderstood as rather definitive, when other possibilities are not excluded. However, the total AOD from the three CALIPSO categories matches quite closely with the total AOD from MODIS, so other categories appear to play a minor role. Moreover, anthropogenic emission estimates (including continental pollution) do not show a decrease over the investigated time period (see page 4 of this reply).

*Pg. 5, Lines 14-15: What evidence is there that the smoke and polluted dust aerosol types are dominated by biomass burning aerosol versus other sources of combustion or pollution aerosols? The CALIPSO aerosol type is not specific to biomass burning. Consequently, for the authors to make this conclusion, they need to provide some other evidence. Since no such evidence is apparent in this manuscript, this seems highly speculative.*

One piece of evidence that biomass burning emissions play a major role, already provided in the paper, is GFED. Another piece of evidence are the CMIP emission inventories, mentioned here in page 4, which could be added to the manuscript.

*Pg. 5, Line 15-18: No data on residential energy sources are provided or discussed in this manuscript, so this statement is entirely speculative, and, frankly irrelevant to the present study. The previous studies cited in the next sentence are also not sufficient to support this statement, as they are not recent enough be cover the 2006-2015 time period in this study. Even if there was a decrease in residential biomass burning emissions starting in the 1990s, such a decrease does not necessarily extend to present day. This conclusion is unfounded.*

Information on the seasonal peak in residential energy sources is provided in He et al., (2011). While this reference should be added here, it should also be clear that this is not a conclusion of the present study, since no relevant data are provided here. The way the referee connects this sentence with the next ones is rather arbitrary. We hope it is clear now that these previous studies are not provided here to support the referred statement, but in a more general discussion on how previous findings relate to ours, which is a rather common and necessary practice.

*Pg. 5, Line 20-: Again, showing the same figure as before at right, it can be seen that there is no agreement between the change in AOD and the change in C emissions (delta-AOD even becomes positive in January, while delta-C is fairly constant). I'm not sure I understand what is being meant by the term, "partially agrees". It appears that during the seasons where delta-C reaches a local minimum and is fairly stable that both MODIS and CALIPSO delta-AOD are quite variable and not at a local minimum or maximum. Finally, is it even appropriate to be trying to establish this comparison, as it was already*

*stated on Pg. 4, Lines 36-37: "Biomass burning emissions and satellite-based AOD are not directly comparable"?*

Biomass burning emissions constitute part of the total aerosol load. Hence, they are expected to agree better with the total aerosol load concentration (or changes) when they dominate compared to other sources, rather than when other sources (or their corresponding changes) dominate the total aerosol load. This is the intended meaning of the term "partial agreement". ΔC actually reaches a minimum in November, when MODIS and CALIPSO ΔAOD also exhibit large decreases, but not necessarily their minimum values (since they are not expected to always agree). It is obvious that the term "not directly comparable" causes a misunderstanding, and could be replaced by "not expected to always agree", which is probably more appropriate for the intended meaning. Regarding the appropriateness of the comparison between biomass burning emissions and satellite-based AOD, the referee is referred to our previous reply (page 8 of this document). In short: yes, comparing a total with its part gives insights on the relative contribution of the latter to the former.

*Pg. 5, Lines 23-26: What evidence is there to assert that the aerosols or smoke observed over this region is transported from neighboring regions such as Indochina (versus long-range transport or local emissions)? No data on fire activity in neighboring regions is presented, nor is any information on air mass back trajectories. What about the confounding influences of local, urban pollution and non-biomass combustion aerosols on the CALIPSO types? This italicized statement seems highly speculative.*

This statement is not an "assertion" based on "evidence". It is a "suggestion", a possible explanation for the reported results, not excluding other possibilities. We agree that there are alternative possibilities, and we thank the referee for suggesting methods that could lead to an assertion based on evidence.

*Pg. 5, Lines 29-33: I agree 100% with this statement. The problem is that this ambiguity undercuts many of the conclusions put forward in this manuscript. This is made even more problematic in that the continental pollution aerosol type is not included in this analysis. Given these uncertainties, a fundamental question that must be asked is, is this satellite-based AOD aerosol type data set appropriate to address source attribution? Given that some major aerosol types are missing, I think the answer is that it is not appropriate.*

We agree that there are serious limitations in using this data set to address source attribution (see page 8, lines 40-42). As stated in page 2, lines 26-27, this data set was not used here for this purpose. It could also be (further) clarified that possible relations of CALIPSO aerosol types with aerosol sources were based only on previous studies (page 4, lines 35-36, page 5, lines 28-29), and do not constitute "strong" conclusions of the present study. This should be obvious from page 8, line 41, but it could be further emphasized.

*Pg. 6, Lines 8-10: I'm struggling to interpret the meaning of this statement. Spatial distributions of the change in LWP and cloud fraction are presented that demonstrate large increases in LWP and cloud fraction over land and decreases over water. No indication is given in Figure 5 of the areas where these changes are or are not statistically significant (often I have seen this done with speckling overlaid on the statistically significant portions). It also sounds like from this statement that most of the cases in Fig. 5 are not statistically significant. Yet, these are the numbers that are being quoted in the abstract for changes in liquid cloud cover and liquid water path of "5% and 13%, respectively" (Pg. 1, Line 12) and for drawing other conclusions later on. Are these numbers statistically significant?*

The information requested by the referee can be easily provided. In fact, the following two maps show the level of statistical significance, on a pixel basis, of all-sky LWP (left) and liquid CFC (right):

[Figure]

In the paper, these results were summarized in the statement that the referee mentions, but they could also be included, to avoid struggles with interpretation. We think, however, that the statement "*reduces this significance to levels below 95% in most cases of Fig. 5.*" is clear. The referee is also referred to the table in the first page of this reply. It is not clear, however, what the referee means by the term "drawing other conclusions later on". After stating that the statistical significance is reduced to levels below 95% in most cases of Fig. 5 due to averaging (page 6, lines 9-10), the 34-year CLARA-A2 time series is analyzed (page 6, lines 13-26), and then changes are examined on a seasonal basis. In page 8, lines 33-34 it is stated which of the cloud properties exhibit statistically significant changes in which months. To clarify any further misunderstandings, we include here a detailed table with corresponding levels of statistical significance (in %) for every cloud property examined and every month:

|  | All-sky LWP | | Liquid CFC | | In-cloud LWP | | Liquid COT | | Liquid REFF | |
|---|---|---|---|---|---|---|---|---|---|---|
|  | CLARA | MODIS | CLARA | MODIS | CLARA | MODIS | CLARA | MODIS | CLARA | MODIS |
| Jan | 14.13 | 13.49 | 54.18 | 39.26 | 6.15 | 5.13 | 3.59 | 6.41 | 14.13 | 56.52 |
| Feb | 10.79 | 3.68 | 44.67 | 39.57 | 10.60 | 15.01 | 5.28 | 1.13 | 14.10 | 52.25 |
| Mar | 5.52 | 15.19 | 1.47 | 12.82 | 12.85 | 7.65 | 12.19 | 11.08 | 47.77 | 45.10 |
| Apr | 33.05 | 42.88 | 28.30 | 4.58 | 58.39 | 61.04 | 50.12 | 54.49 | 73.79 | 72.08 |
| May | 79.19 | 94.20 | 61.31 | 30.53 | 96.01 | 99.22 | 85.28 | 95.12 | 86.93 | 48.38 |
| Jun | 13.69 | 26.27 | 85.18 | 49.95 | 39.64 | 62.92 | 36.34 | 55.18 | 61.09 | 59.25 |
| Jul | 22.71 | 39.33 | 2.57 | 56.52 | 26.96 | 26.94 | 23.06 | 2.27 | 90.11 | 93.16 |

| Aug | 4.06 | 77.65 | 57.83 | 42.97 | 74.50 | 29.39 | 34.42 | 1.00 | 86.12 | 97.35 |
| Sep | 36.62 | 27.02 | 55.97 | 37.12 | 30.88 | 33.54 | 21.79 | 22.90 | 92.01 | 46.53 |
| Oct | 46.82 | 69.47 | 74.85 | 81.51 | 72.35 | 70.81 | 59.18 | 66.37 | 79.20 | 50.22 |
| Nov | 90.07 | 90.99 | 99.26 | 99.55 | 65.37 | 86.26 | 48.39 | 78.23 | 89.59 | 88.76 |
| Dec | 97.05 | 99.33 | 35.75 | 54.15 | 99.43 | 99.91 | 96.45 | 98.57 | 97.46 | 67.40 |

In fact, contrary to the referee's statement, no conclusion is reported based on the changes in the entire time series of Fig. 5. The reason is that "most of the cases in Fig. 5 are not statistically significant". We hope this is clearer now.

*Pg. 6, Lines 33-34: Do I understand this statement correctly to say that there is only one month where each of the cloud property changes is statistically different from zero, and that only cloud fraction is statistically different from zero in November? Why would there be a trend in cloud properties in only a single month? Reading between the lines here, does this mean that the overall change in cloud property changes is not statistically significant, or is the effect size of this single month enough to drive the entire trend?*

Yes, this is exactly what this sentence states. Regarding the referee's second question, a possible reason could be "differences in the seasonal (or monthly) characteristics and changes in factors affecting cloud properties". Such a factor is aerosols, and a large part of the present study investigates exactly this question. This single month is not enough to drive the entire trend, hence most of the cases in Fig. 5 are not statistically significant, as stated in page 6, lines 9-10. Our previous reply also provides further clarifications.

*Pg. 6, Lines 37-38: I don't think this statement is correct. Liquid clouds increased only in November on a cloud fraction basis and only in December on a LWP or COT basis (based on the statistical significance discussion in the previous comment). The seasonal pattern of the AOD changes and the cloud properties changes are not similar (either correlated or anti-correlated). This statement suggests that they are anti-correlated, which is not true.*

This statement was phrased carefully, to avoid misinterpretations. The referee correctly states that liquid clouds increased only in November on a cloud fraction basis and only in December on a LWP or COT basis, based on the statistical significance results (see also the relevant table in pages 11-12 of this document). We summarized these changes using the phrase "liquid clouds increased mainly in late autumn and early winter", but this could be rephrased, based on the referee's statement, to be more clear. We also agree that the seasonal pattern of the AOD changes and the cloud properties changes are not similar. In fact, we don't see why they should be similar (either correlated or anti-correlated) throughout the year. However, it is one of our findings that some statistically significant changes in properties of clouds and aerosols occurred in the same months, and they are indeed anti-correlated (see also the table in page 16 of this document).

*Pg. 7, Line 10-11: First, statistical significance of the cloud changes is unclear (discussed previously). There does appear to be a decreasing trend in AOD, which the authors assert is statistically significant. That surface pressure and geopotential height do not show a statistically significant trend is insufficient to rule out meteorological drivers. The atmospheric temperature profile, moisture, and lower tropospheric stability are also important variables that do not appear to have been considered. Even if these variables fail to demonstrate a statistically significant trend that does not in and of itself rule out the existence of such a trend. All it means is that the available data are insufficient to reject the null hypothesis, but there may indeed be a trend that might be uncovered by additional data and/or a longer timeseries. The italicized statement is not demonstrated conclusively by the data presented, which are rather superficial.*

We hope the statistical significance of cloud and aerosol changes in now clearer, based on our previous replies. Apart from surface pressure and geopotential height, other parameters could indeed be considered, as the referee suggests. It is not clear, however, what the referee means by the following: "*Even if these variables fail to demonstrate a statistically significant trend that does not in and of itself rule out the existence of such a trend. All it means is that the available data are insufficient to reject the null hypothesis, but there may indeed be a trend that might be uncovered by additional data and/or a longer timeseries*". This statement seems to suggest that there is no way of excluding meteorological variability as a factor of cloud and aerosol changes, since there might always be a trend waiting to be uncovered by additional data and/or longer time series. This contradicts common practices followed in similar studies.

*Pgs. 7-8, Section 3.3.2: I think that this paragraph is not at all supported by the underlying data, which until this point has focused on trends and changes over time. In this paragraph, process-level explanations are invoked, but are done at a highly-averaged level spanning months and 5 x 10 degree area. These are not the scales at which aerosol-cloud interactions would be expected to be evident (e.g., McComiskey and Feingold, ACP, 2012, https://www.atmos-chem-phys.net/12/1031/2012/), so the failure to see ACI effects in the trend data is not surprising. Saying that the authors' findings are "inconsistent" with the first and second indirect effects is too strong a statement. These effects may very be visible in this region if a more appropriate data set is used (e.g., aircraft, balloon, surface remote sensor scales measuring clouds over minutes to hours or a model with better space and time resolution). One cannot know. The same is true for the discussion on the semi-direct effect, with the additional comments that have been described above in this review that the attribution of the particles to biomass burning, as absorbing particles, and that these particles are at cloud level are all not established by this data set. In fact, the CALIPSO smoke type is not unambiguous as a marker for biomass burning. That the smoke type often occurs near cloud level is unsurprising given that the layer must be elevated by definition of that aerosol type in the CALIPSO scheme. Similarly, the continental pollution aerosol type is very similar to the smoke type, but it is not in an elevated layer. Since the data set used in this manuscript and its analyses is so highly averaged in space and time, it is of little utility for discussing ACI effects. Consequently, the conclusions as stated are not definitive and this entire paragraph should be removed.*

This paragraph examines what can (and cannot) be deduced from the previous analysis regarding ACI and clearly states what can (and cannot) be supported by the data. It is not true that results of ACI cannot be evident at scales larger than their process scale, as the referee suggests. The problem with the temporal and spatial scales used here, is one of quantification of ACI, as clearly stated in the study cited by the referee, and acknowledged also in our study (page 1, lines 33-35). However, no ACI quantification was attempted in the present study, so the referee's criticism is rather unsupported. In the same sense, if a particular mechanism dominated over a large area and period, one would expect to see the consequences in a data set covering this area and period. If the analyzed data sets show changes in a direction opposite to the one expected, then they are "inconsistent" with the previous assumption. We don't see why such a term would be "strong" or "unsupported" by the data.

We agree with the referee that ACI may be visible based on the suggested data and resolutions. However, the purpose of this study is not to provide evidence of ACI on their process level. It is (among others) to examine if their consequences can explain observed changes in a larger scale.

We acknowledge the referee's concerns on the limitations of the CALIPSO aerosol types based on their definitions. In fact, their limitations are acknowledged in several parts of the manuscript (page 5, lines 29-33, page 8, lines 18-20, page 8, lines 40-42). However, this section examines *changes* in their profiles, not concentrations. The referee finds "unsurprising" that "the smoke type often occurs near cloud level". This is a rather confusing statement, since it is nowhere made in the manuscript.

*Pg. 8, Line 10: Are the aerosol and cloud profiles shown in Figure 8 an average profile or an individual, typical profile for each? What is meant by "autumn" or "Fall" for the cloud extinction profile – both October and November? If they are averaged profiles, how was that averaging carried out (e.g., was a weighted average of the sample numbers in each pixel used)? Are there meaningful differences in the profiles across the spatial area? A single set of averaged profiles over the entire spatial domain seem difficult to meaningfully interpret to me, as I would expect these profile changes to be very different over land and over water. How should the reader interpret these profiles with regard to representativeness? Are the changes in Fig. 8b and 8c statistically significant at all height levels? The commentary on Pg. 8, Line 21-23 suggests that only certain layers are statistically significant and in different months (e.g., 1-1.5 km altitude for smoke in October and 0.7-1.2km for polluted dust in November). It would appear that the CALIPSO smoke aerosol change is not statistically significant in November when the cloud fraction change is statistically significant. Conversely, the smoke change is statistically significant in October when the cloud fraction change is not statistically significant. What about the December profiles where the other cloud property changes are statistically significant? It is very difficult to unravel what is being presented here, but it certainly does not seem to be suggestive of an aerosol-cloud semi-direct effect (as is stated on Pg. 1, Lines 15-18).*

The cloud profile shown in Figure 8 is spatially averaged over the study area and autumn (fall) months (September, October, November), based on measurements from 2007 to 2011 (see also Amiridis et al. 2015, for details on the LIVAS data set). The aerosol profiles of Figure 8 actually show changes, calculated, for each profile level, based on the method described in Section 2.4. The averaging was

indeed carried out using the numbers of averaged samples, also provided in the data set, as weights. It is not clear what the referee means by "meaningful differences". It is true that differences should be expected, especially over land and sea, and selecting two of the four pixels covering the study region would probably be more representative of the land profiles. A separate analysis could be performed to answer the question on representativeness.

The referee interprets correctly the statement in page 8, lines 21-23. To clarify further: in October, smoke changes are statistically significant between 1-1.5 km, liquid cloud changes are not; in November, polluted dust changes are statistically significant between 0.7-1.2km and liquid CFC change is also statistically significant; in December, all liquid cloud properties changes examined, except for CFC, are statistically significant, and aerosol changes are not. While some rephrasing might help, we consider reporting the results on statistical significance along with corresponding changes really crucial, hence some difficulty in unravelling the findings should be expected.

*Figure 9 and related discussion: Why is necessary to break out the cloud optical thickness data by cloud type? It has already been established from Figure 7 and associated discussion that delta-COT is not statistically significant in October or November. This is mentioned in passing on Pg. 8, Line 28. Yet, there is then extensive discussion on the coincidence between decreased biomass burning, increased liquid cloud fraction and water content in November and a decrease in smoke aerosols in October (Pg. 8, Lines 28-32). I find this discussion very confusing, but much of it appears to be based on source attribution that has already been discussed in this review as being speculative. The ISCCP cloud type classifications does not bring any additional clarity or information to the major flaws in the prior conclusions.*

It is not clear what the referee means here. Establishing a non-significant change in a parameter of liquid clouds, does not necessarily exclude the same parameter from changing significantly in a cloud sub-type. Similarly, establishing a non-significant change in a time series does not exclude significance on a monthly basis, as was shown in this study. As the referee mentions in a previous comment, establishing non-significance does not rule out the existence of a significant change that might be uncovered by additional data and/or longer time series.

The part of the discussion mentioned by the referee seems indeed confusing and should be rephrased, since it is not based on source attribution, as the referee claims, but rather on the position of the aerosols relative to clouds. The notion that a non-significant change in COT should prevent an analysis and discussion of changes in biomass burning and smoke aerosols, coinciding with changes in liquid cloud fraction and water content, is also unsupported.

*Pg. 8, Lines 40 – Pg. 9, Line 1: I fundamentally disagree with second part of this statement. The manuscript is saying that smoke aerosols are from biomass burning and are absorbing, and therefore, an association between smoke aerosol height and cloud height in Figure 8 is somehow related to the semi-direct effect. If the aerosols are misclassified or smoke dominated urban aerosols that are weakly*

*absorbing then this would indeed call the conclusions of the manuscript into question. The missing CALIPSO aerosol types and the compositional ambiguity provided by this typing method make this particular dataset less capable for addressing the types of science questions and drawing the types of conclusions that are sought in this manuscript. This limitation is a significant one that cannot be overcome without new data and new analyses.*

The referee keeps repeating the same argument. We acknowledge that CALIPSO does not give unambiguous information about certain aerosol types, in particular biomass burning smoke and urban pollution. However, as stated before, the GFED dataset demonstrates that (absorbing) biomass burning aerosol emissions have markedly decreased over the decade studied. At the same time, anthropogenic emissions have not decreased (see Figure on page 4 of this reply). These pieces of information give strong additional indications that the aerosols are not largely misclassified.

*Pg. 9, Lines 1-4: Here are the monthly timeseries from Fig. 1 and Fig. 4. I think that it's hard to make the case as is done here that there is a meaningful anti-correlation between polluted dust AOD and liquid CFC only in November and December (circled regions) and that such an anti-correlation can be used to draw a conclusion. There is just too much variability (and that's before the additional requested information of standard deviation error bars or shaded regions are added to the graph). The decreasing pattern in the polluted dust aerosol is present for the smoke aerosol and does occur during other months. Given the scale, it's difficult to discern the trend for the dust trace.*

The following table shows the Pearson's coefficients of monthly liquid CFC from CLARA-A2 and MODIS, and AOD from CALIPSO. Results are shown separately for total, dust, smoke and polluted dust AOD.

| | Total | | Dust | | Smoke | | Polluted dust | |
|---|---|---|---|---|---|---|---|---|
| | CLARA-A2 | MODIS | CLARA-A2 | MODIS | CLARA-A2 | MODIS | CLARA-A2 | MODIS |
| January | -0.21 | -0.27 | 0.26 | 0.18 | -0.23 | -0.26 | -0.11 | -0.17 |
| February | -0.32 | -0.23 | -0.10 | 0.05 | -0.45 | -0.44 | -0.15 | -0.07 |
| March | -0.42 | -0.29 | 0.01 | <0.01 | -0.60 | -0.53 | -0.09 | 0.07 |
| April | 0.02 | 0.05 | 0.30 | 0.35 | 0.35 | 0.43 | -0.48 | -0.51 |
| May | 0.49 | 0.55 | -0.13 | -0.34 | 0.39 | 0.53 | 0.34 | 0.36 |
| June | 0.15 | 0.42 | -0.06 | 0.17 | -0.07 | -0.17 | 0.11 | 0.38 |
| July | 0.36 | 0.26 | -0.24 | -0.53 | 0.21 | 0.41 | 0.47 | 0.12 |
| August | 0.79 | 0.74 | -0.04 | -0.15 | 0.69 | 0.75 | 0.41 | 0.26 |
| September | 0.02 | 0.18 | 0.18 | 0.37 | -0.31 | -0.18 | 0.08 | 0.26 |
| October | <0.01 | -0.29 | 0.42 | 0.01 | 0.02 | -0.32 | 0.05 | -0.15 |
| November | -0.53 | -0.50 | 0.21 | 0.21 | 0.08 | 0.10 | -0.73 | -0.69 |
| December | -0.83 | -0.81 | 0.22 | 0.14 | -0.69 | -0.66 | -0.80 | -0.79 |

A quick inspection of the table makes our statement obvious: *the liquid CFC and the polluted dust AOD are anti-correlated in November and December, with correlation coefficients around -0.7 to -0.8.* This statement is true for both CLARA-A2 and MODIS liquid CFC. Inclusion of this table would clarify this

issue. However, the referee inadequately uses the plots showing the monthly averages of these variables and falsely calls these plots "monthly time series", to disprove a statement that was not based on those plots. This is an unsettling level of misunderstanding.

*Pg. 9, Line 8-10: No data on aerosol absorption are presented in this study. Some CALIPSO aerosol types are missing, and those that are included cannot be unambiguously attributed to biomass burning emissions. This conclusion is not true.*

This same argument has been made many times, and we refer to our reply on page 16.

*Pg. 9, Line 10-11: Technically this statement just says that cloud fraction and thickness both change across months. The changes are neither correlated nor anti-correlated (Figure 4a and 4c). At best, this sentence doesn't reach a meaningful conclusion, and at worst it misleads the reader into thinking that concurrent changes somehow track each other. I recommend that this sentence be removed.*

This sentence should indeed be rephrased. It is apparent that the referee is confused and makes an inadequate judgement, since Figure 4a and 4c do not show changes in liquid cloud fraction and optical thickness, but monthly averages of these parameters during the period examined. The sentence refers to changes concurrent with aerosol changes, in different months.

*Pg. 9, Line 11-13: No data actually relating the position of aerosols to clouds is presented. Instead, the vertical distribution of cloud extinction and the vertical distribution of aerosol temporal change are presented in Figure 8. These are not the same thing. It is not shown that the sign of cloud changes is determined by the position of aerosol relative to clouds. This statement is not true.*

We thank the referee for this remark. Indeed, the data provided actually relates the position of aerosol *changes* relative to clouds, and this statement should read "*Further analysis of vertical profiles of aerosol changes and clouds showed that the signs of cloud changes depended on the position of aerosol changes relative to clouds…*" in order to be true.

*Pg. 9, Lines 17-18: It is very difficult to relate the trend analysis changes for the aerosol AOD and cloud changes to a process-level causal mechanism. Even comparing the trend changes of aerosol and cloud in this manuscript are difficult because they appear to vary differently across monthly and with different (and poorly explained) levels of statistical significance. It is not true to suggest that the data show a "high level of consistency" with the semi-direct or any ACI effect. This is because the data used here are highly averaged in both space and time, and are, therefore, less ideal for tackling these sorts of science questions.*

Acknowledging the limitations of the data and concerns similar to the ones of the referee, we did not attempt to establish any process-level causal mechanism, as was emphatically noted in the statement right after the one cited here by the referee. We also tried to establish consistency in space and time (see Section 2.4) before attempting any comparison of changes. We hope that the levels of statistical significance are better explained now based on our previous replies. However, we consider that suggesting "*a high level of consistency with predictions*" of the semi-direct effect is a modest statement, which takes into account the limits of our analysis. The reason invoked by the referee to dismiss this statement, namely that the data used are highly averaged in both space and time, is indeed prohibitive for quantifying any ACI effect. However, this is not what we do here. And there is no physical reason limiting the *consequences* of any ACI on their process-level space and time scales only.

*Pg. 9, Lines 18-20: This statement succinctly highlights the lack of depth of the analyses in this manuscript. There are a large number of critical limitations associated with the use of these data to try to draw these sorts of conclusions. The limitations are acknowledged in the manuscript, but there is no real attempt to overcome them. The model analyses discussed in this last sentence hold promise for being able to attribute aerosol to sources as well as to link those aerosol to clouds. I recommend that those tools be brought to bear on the questions being tackled here, perhaps with some context being provided by more complete set of aerosol types in the Level 2 version of the satellite data that is discussed here.*

It is unfortunate that the acknowledgement of limitations in a data set and/or method of analysis is characterized as "lack of depth", especially when the referee refers to another kind of study and conclusions. Specifically, in the previous comment, the referee reasons that the data used here are "*less ideal for tackling these sorts of science questions*", because they are "*highly averaged in both space and time*". Indeed, these data are inadequate to establish an ACI cause and effect mechanism. This is clearly acknowledged as a limitation, but it was never described as a science question to be tackled in this study. Similarly, using level 2 data which are closer to the ACI process-level in terms of both spatial and temporal resolution, and/or model analyses (especially the latter), could probably lead to robust conclusions on aerosol sources and links with clouds, again, in the process level. In a previous comment, the referee invokes the study by McComiskey and Feingold (2012) to justify the inadequacy of the scales used here and a predefined failure to "*see ACI effects*". That study, however, tackles the question of quantifying the ACIs. This was never a goal in the present study, exactly because this limitation was acknowledged. The same holds for the suggestion, by the referee, of a "*more appropriate data set … (e.g., aircraft, balloon, surface remote sensor scales measuring clouds over minutes to hours or a model with better space and time resolution)*".

---

## Referee Report (RR1)

Review of revised paper:

**Satellite observations of aerosol and clouds over southern China from 2006 and 2015: analysis of changes and possible interaction mechanisms.**
*by N.Benas et al.*

**Positives**
- changes in aerosols properties (now also MISR) are examined over the last decade
- changes in satellite retrieved cloud properties over last decade are analyzed
- exploring co-located changes in aerosol and cloud retrievals (if not for insights in potential aerosol-cloud processed then at least) are constraints for transient model simulations.

**Concerns**
- GFED (biomass) emissions are insufficient to explain AOD(f) trends - and industrial emissions apparently neither … but at least show both seasonal cycles and their changes over time the figures – could meteorology be an explanation (monsoon?, clouds?).
- the examined region is relatively small
- many data limitations / inconsistencies are recognized but not further explored … so the value of the paper is limited (suggested links remain speculative).

**General comments:**

This study examines co-located 'observational data' based on satellite retrievals for aerosol and clouds. By exploring monthly retrieval data over a decade in relatively small region over southern China speculation on aerosol cloud interactions/processes are made.

Compared to the initial version I am happy to notice that the poorly defined CALIPSO type has been removed from the argumentation chain, that now MISR AOD and fine mode AOD have been included and that there is a move to a coarser temporal averaging from one to two months (although only in tables, but not in the Figures).

My analysis shows for the MODIS data an AOD decrease in the region, which is at maximum in fall with -0.23 (from 0.57 to 0.35) for the AOD decadal decrease and -0.15 (from 0.34 to 0.19) for the decadal fine-mode decrease. For MISR data we have an AOD decrease in the region, which is at maximum in fall with -0.13 (from 0.40 to 0.27) for the AOD decadal decrease and -0.09 (from 0.23 to 0.14) for the decadal fine-mode decrease (see attachment with plots with special distributions of absolute values and anomalies). These are very large changes in AOD, so the idea to related responses in associated cloud properties seems to have its merit. For that time-range the decrease in seasonal MODIS based CCN data over the ocean is consistent though not associated with the fall season (annual, monthly data available on request).

The paper is about aerosol properties trends and cloud property changes over the last decade and there are still gaps in data interpretations. While it is assuring that data from different sensors often agree, the association among the different aerosol properties (AOD, AODf, AODc spatially and seasonally) and cloud properties (COT, phase, LWP, reff  spatially and seasonally) should be better harvested to draw a better basis (before trying to link aerosol and cloud data for potential processes/interactions).

The authors addressed all comments, which more focused on explaining what and why things were as they are. This way, opportunities for improvements were avoided and missed. The paper is an analysis of retrieved cloud properties and retrieval aerosol properties both for the last decade over small region over southern China. Even if significant temporal trends are identified it is still a big task to draw potential interactions from trend associations.

As long as the paper keeps focusing on a solid analysis for aerosol and cloud retrievals

and observed (relative) changes this contribution is interesting and useful, even though the applied region is relatively small. I still wonder about the changes to other related properties (e.g. cloud top height, rain, surface temperature). The interpretation certainly is tempting though speculative … and rather an element for the discussion section. Try to be more convincing!

Figure 1
I am not sure if this plots is necessary as the focus is on changes. Still I wonder why MODIS is so much different to MISR (I tend to trust more MISR retrieval capabilities over continents and I would show MODIS AODf and AODc to add up to total AOD (I do not understand the large gap between AOD and (AODf+AODdust). I suggest to plot seasonal data (if you have to show regional average instead of maps) for a period near 2006 and a period near 2015 one for MISR (total, fine, coarse AOD), one for MODIS (total, fine, coarse AOD), one for GFED and one for fossil fuel emissions.
Figure 2
the AOD change (should be 2015-2006) seem way too small – based on my analysis for this region (check!). And patterns are more informative to me than trend plots.
Figure 3
The GFED data are down not only in fall but also in winter, while AODf an AOD values are minly down in fall. Thus, lower GFED emissions are a contributing factor but not the sole explanation. I love to see differences in fossil fuel emission (S.Smith has published data). Have you considered an shift in monsoon activity (e.g. are there seasonal precipitation data?) If it was more wet in fall the this also could explain (by wet removal, lower AOD, AODf and CDNC data).
Figure 4
Same complains as in figure 1: Show typical seasonal data for periods near 2006 and periods near 2015. I rather trust relative difference (which are just needed here) than absolute retrievals.
Figure 5
changes: 2015 minus 2006. I take from this figure that the liquid water path increase is much larger than the cloud cover increase → more convection → more wet removal? I am also puzzled why the effective radius increase is much larger than the COT increases. Does that mean that cloud tops are higher (with larger droplets on top). There is more interpretation needed to understand these retrieval cloud properties … that is if we can trust them.
Figure 6
In the last 10 years for clouds reff, COT, LWP and cover all increased … what does this mean for cloud type frequency … and then we can think about potential impact on aerosol.
Figure 7
You are talking about 2 month data analysis … but plots still show monthly anomalies. Maybe you can show 3 month running averages? The largest aerosol reductions are in fall… but the largest cloud properties are in winter. Is there
really a link (e.g. do you believe in a seasonal time-lag?)
Figure 8
I am puzzled about the big changes in Calipso profiles within 2 months. Is there a good reason why these extinction profile changes are so different?

**minor comments to the responses**

in the response it is mentioned that the reasons for why properties observed as they are, are only of secondary concern, as in the end associated changes between aerosol and clouds are of interest. I disagree and I think we first should understand why satellite retrievals do change over time so we have more confidence that what we eventually do compare is meaningful.

Thanks for checking that industrial emission apparently even increased. Unfortunately it is not clear if seasonal variations are offered (as I could not find the supplement). This background information deserves to be part of the paper. Other background changing elements would be temperature, [solar] radiation, precipitation and the monsoon time-period.

In the response to reviewer 2 there is a figure with CEOS emission. Now it would be interesting if there is a seasonality to these emissions or do they just provided annual averages?

Overcast cloud conditions: since these are required for bi-spectral retrievals methods (COT, reff) I wonder if that frequency changed? Are there other properties that can provide insights on why the cloud properties have changed? Did the cloud-top change?

**minor comments to the new text**

in the abstract you talk about a 40% AOD reduction. This is an exaggeration. My seasonal analysis shows ca 30% reduction between 2006 and 2015 only for fall (other seasons are much less) and a significant part of the reduction (ca 30%) is related to coarse mode aerosol (which little link to GFED emissions).

in the abstract the last sentence comes across as a statement but is highly speculative at best.

aerosol results: why is the dust AOD so small? AODc (mainly dust over continents) and AODf should add up to AOD. … and please show industrial emission change for that region, preferably with a seasonal cycle (in the warmer/humid summer the pollution related AOD should be larger)

the discussion section is much improved. I like the idea with the reduced semi-direct effect. Would this not also imply a more unstable atmosphere and with a stronger convection a higher cloud top?

assuming the altitude assignment of AOD change is correct … what can be reason that elevated AOD is so much reduced in fall? why not at the ground?

**AOD multi-annual seasonal averages (over the SE China region)**

[Figure]

**2015 minus 2006 AOD seasonal averages (over the SE China region)**

[Figure]

MISR  total aerosol opt.depth 550nm  (seasonal for diff years)

[Figure]

MODIS  total aerosol opt.depth 550nm  (seasonal for diff years)

[Figure]

MISR fine aerosol opt.depth 550nm (seasonal for diff years)

[Figure]

MODIS fine aerosol opt.depth 550nm  (seasonal for diff years)

[Figure]

MISR diff in total AOD 550nm (seasonal for diff years) comp to multi-year

[Figure]

MODIS diff in total AOD 550nm (seasonal for diff years) comp to multi-year

[Figure]

MISR diff in fine AOD 550nm  (seasonal for diff years) comp to multi-year

[Figure]

MODIS diff in fine AOD 550nm (seasonal for diff years) comp to multi-year

[Figure]

MODIS diff in CDNC (seasonal for diff years) compared to multi-year

[Figure]

*Here seasonal averages and anomalies with respect to multi-seasonal average are compared for fine-mode AOD and for total AOD from both MODIS and MISR for the SE China region in the Benas paper. Due to the smaller swath the MISR data are much noisier.*

*in recent years during the fall-season (Sep, Oct, Nov) there are large reductions to the AOD in the SE-Asia study region. Hereby fine-mode AOD reduction account for ca 2/3 and still 1/3 can be attributed to coarse mode AOD reduction.*
*Also Modis based CDNC show reduced concentrations over the entire year over the adjacent ocean.*

---

## Author Response (AR2)

**Reply to Referee #1 (Stefan Kinne)**

We thank Stefan Kinne for reviewing our manuscript and providing comments for revision. Following are our point-by-point replies, with his comments in italic. Please note that figure, page and line numbers refer to the marked-up version of the revised manuscript and supplement, attached here after our responses.

***General comments:***

*This study examines co-located 'observational data' based on satellite retrievals for aerosol and clouds. By exploring monthly retrieval data over a decade in relatively small region over southern China speculation on aerosol cloud interactions/processes are made.*

*Compared to the initial version I am happy to notice that the poorly defined CALIPSO type has been removed from the argumentation chain, that now MISR AOD and fine mode AOD have been included and that there is a move to a coarser temporal averaging from one to two months (although only in tables, but not in the Figures).*

*My analysis shows for the MODIS data an AOD decrease in the region, which is at maximum in fall with -0.23 (from 0.57 to 0.35) for the AOD decadal decrease and -0.15 (from 0.34 to 0.19) for the decadal fine-mode decrease. For MISR data we have an AOD decrease in the region, which is at maximum in fall with -0.13 (from 0.40 to 0.27) for the AOD decadal decrease and -0.09 (from 0.23 to 0.14) for the decadal fine-mode decrease (see attachment with plots with special distributions of absolute values and anomalies). These are very large changes in AOD, so the idea to related responses in associated cloud properties seems to have its merit. For that time-range the decrease in seasonal MODIS based CCN data over the ocean is consistent though not associated with the fall season (annual, monthly data available on request).*

*The paper is about aerosol properties trends and cloud property changes over the last decade and there are still gaps in data interpretations. While it is assuring that data from different sensors often agree, the association among the different aerosol properties (AOD, AODf, AODc spatially and seasonally) and cloud properties (COT, phase, LWP, reff spatially and seasonally) should be better harvested to draw a better basis (before trying to link aerosol and cloud data for potential processes/interactions).*

*The authors addressed all comments, which more focused on explaining what and why things were as they are. This way, opportunities for improvements were avoided and missed. The paper is an analysis of retrieved cloud properties and retrieval aerosol properties both for the last decade over small region over southern China. Even if significant temporal trends are identified it is still a big task to draw potential interactions from trend associations.*

*As long as the paper keeps focusing on a solid analysis for aerosol and cloud retrievals and observed (relative) changes this contribution is interesting and useful, even though the applied region is relatively small. I still wonder about the changes to other related properties (e.g. cloud top height, rain, surface temperature). The interpretation certainly is tempting though speculative … and rather an element for the discussion section. Try to be more convincing!*

We thank the reviewer for this comment. In the revised manuscript we have expanded the analysis, including additional parameters suggested by the reviewer (cloud top height, rain, surface air temperature), and providing additional results on distributions and correlations (e.g. Figs. 5, S6, S7, S8, Table S10). While our main conclusions did not change based on these, we hope that they have improved our study, by filling some interpretation gaps and contributing to its completeness.

*Figure 1*

*I am not sure if this plots is necessary as the focus is on changes. Still I wonder why MODIS is so much different to MISR (I tend to trust more MISR retrieval capabilities over continents and I would show MODIS AODf and AODc to add up to total AOD (I do not understand the large gap between AOD and (AODf+AODdust). I suggest to plot seasonal data (if you have to show regional average instead of maps) for a period near 2006 and a period near 2015 one for MISR (total, fine, coarse AOD), one for MODIS (total, fine, coarse AOD), one for GFED and one for fossil fuel emissions.*

The purpose of this figure is to provide an overview of the seasonal behavior of AOD and emissions over the study region and during the period examined. Since the relevant changes are later examined also on a seasonal basis, we consider the information provided in Fig. 1 necessary for the completeness of the description of these aerosol and emissions characteristics.

Regarding MISR AOD, the reviewer correctly points at the large gap between total AOD and $AOD_f$ + $AOD_{dust}$, since $AOD_c$ was missing from the analysis. In the revised manuscript, we have included coarse mode AOD from MISR. Regarding AOD components from MODIS, only fine mode AOD over ocean was available in the daily level 3 product. Including this data set would probably lead to confusing results due to spatial inconsistencies, instead of clarifications.

Regarding the suggestion to plot separate maps for the beginning and end of the study period: the difficulty is that the first and last year do not necessarily reflect the changes accurately, because these changes are derived from fits to the complete time series. Therefore, we would like to stick to our original presentation of these changes.

*Figure 2*

*the AOD change (should be 2015-2006) seem way too small – based on my analysis for this region (check!). And patterns are more informative to me than trend plots.*

We have checked again the results of our analysis, and they are correct. As explained in Section 2.4, the numbers are calculated as the difference (in percent) between the first and last years, based on the linear regression fit to the deseasonalized time series. Patterns of changes and relevant significance levels related to Fig. 2 are shown in supplementary Figs. S1 and S2.

Please also note that we preferred to use "changes during 2006-2015" rather than "2015 minus 2006", since the latter suggests that properties for the year 2006 were subtracted from those for the year 2015, which is not the case. However, we have replaced "during 2006-2015" with "from 2006 to 2015" to clarify the issue.

*Figure 3*

*The GFED data are down not only in fall but also in winter, while AODf an AOD values are minly down in fall. Thus, lower GFED emissions are a contributing factor but not the sole explanation. I love to see differences in fossil fuel emission (S.Smith has published data). Have you considered an shift in monsoon activity (e.g. are there seasonal precipitation data?) If it was more wet in fall the this also could explain (by wet removal, lower AOD, AODf and CDNC data).*

Based on the reviewer's suggestion we have included in the analysis monthly precipitation data from the Global Precipitation Climatology Project (GPCP) data set. We found that there is a strong increase in precipitation during autumn and early winter, anti-correlated with AOD changes. This indeed suggests that wet removal played

a role in the AOD decrease found during the same period. This additional analysis is now included in page 6, lines 8-16 and Fig. 5 of the revised manuscript.

Changes in anthropogenic emissions, including fossil fuels, were previously presented in the supplement, but are now moved to the main text (Fig. 4). However, they show an increase for this area and period examined, hence they cannot contribute to explaining the aerosol decrease.

*Figure 4*
*Same complains as in figure 1: Show typical seasonal data for periods near 2006 and periods near 2015. I rather trust relative difference (which are just needed here) than absolute retrievals.*

As in the case of Fig. 1, our intention with Fig. 4 (now Fig. 6 in the revised manuscript) was to show the seasonal behavior of cloud properties, which is characteristic of the entire period examined, and provide the reader with information on how these properties behave within a year. For this reason, these monthly averages were computed based on the entire study period. What the reviewer suggests would be helpful when examining differences between the start and end of the study period. However, this is not the purpose of this figure.

*Figure 5*
*changes: 2015 minus 2006. I take from this figure that the liquid water path increase is much larger than the cloud cover increase -> more convection -> more wet removal? I am also puzzled why the effective radius increase is much larger than the COT increases. Does that mean that cloud tops are higher (with larger droplets on top). There is more interpretation needed to understand these retrieval cloud properties … that is if we can trust them.*

The reviewer's suggestion was verified by adding the analysis on precipitation changes (page 6, lines 8-16 and Fig. 5). Cloud top height changes were also included, showing that indeed there was an increase in cloud top heights (page 7, lines 1-7 and supplementary Fig. S8). However, the ambiguity between CLARA-A2 and MODIS results regarding effective radius and cloud top height renders this explanation more dubious.

As noted in our reply on Figure 1, we used "changes during the period 2006-2015" instead of "2015 minus 2006", since the latter suggests that properties for the year 2006 were subtracted from those for the year 2015, which is not the case. Changes were determined from linear fits to the full time series. We have replaced "during 2006-2015" with "from 2006 to 2015" to clarify the issue (caption of Fig. 7 in the revised manuscript).

*Figure 6*
*In the last 10 years for clouds reff, COT, LWP and cover all increased … what does this mean for cloud type frequency … and then we can think about potential impact on aerosol.*

The purpose of Fig. 6 (Fig. 8 in the revised manuscript) is to put changes observed in the last 10 years in the context of the full CLARA-A2 record starting in 1982. This analysis shows that the recent increases in LWP, liquid CFC and COT are unique in the time series. The increase in liquid CFC implies a more frequent occurrence of warm, low clouds including stratus and stratocumulus, and not only has the occurrence of these clouds increased, they have also become thicker. Therefore, the discussion focusses on potential cloud-aerosol interaction mechanisms that are applicable to warm, low clouds.

*Figure 7*
*You are talking about 2 month data analysis … but plots still show monthly anomalies. Maybe you can show 3 month running averages? The largest aerosol reductions are in fall… but the largest cloud properties are in winter. Is there really a link (e.g. do you believe in a seasonal time-lag?)*

We chose to present the changes on a monthly basis in the figures in order to not anticipate and fix a certain grouping of months. For the statistical analyses bi-monthly periods are then used to improve the robustness of the results.

The possibility of a seasonal time-lag between aerosol changes and effects on clouds does not appear to be likely since aerosol-cloud interaction occurs on short timescales. As the reviewer notices, the largest aerosol reductions do not coincide with the largest cloud changes. We hypothesize that in September-October, when the largest aerosol changes occur, the aerosols are located higher in the atmosphere and there is no direct connection between aerosol and cloud changes. There is overlap of changes, however, in aerosols, emissions and cloud properties in November-December and the respective time series also show significant correlations. The aerosol semi-direct effect was investigated as a possible mechanism connecting these changes and correlations.

*Figure 8*
*I am puzzled about the big changes in Calipso profiles within 2 months. Is there a good reason why these extinction profile changes are so different?*

A possible explanation that we proposed in the discussion section relates these profile changes to the changes in biomass burning emissions. While practically no such emission change was found in September-October, large reductions were found in November-December (see also Fig. 3b). This decrease would also lead to a reduction in aerosol loads in the lower atmospheric layers, manifested here as a decrease in corresponding aerosol extinction profiles.

**minor comments to the responses**

*In the response it is mentioned that the reasons for why properties observed as they are, are only of secondary concern, as in the end associated changes between aerosol and clouds are of interest. I disagree and I think we first should understand why satellite retrievals do change over time so we have more confidence that what we eventually do compare is meaningful.*

Our point was that the good agreement between two different satellites and data sets adds confidence in our results. This is also why we explicitly mention in the text that results are dubious when the two data sets disagree. We agree with the reviewer on the importance of ensuring the quality of satellite-derived data before drawing any further conclusions.

*Thanks for checking that industrial emission apparently even increased. Unfortunately it is not clear if seasonal variations are offered (as I could not find the supplement). This background information deserves to be part of the paper. Other background changing elements would be temperature, [solar] radiation, precipitation and the monsoon time-period.*

In the revised manuscript we have added analyses on precipitation and surface air temperature. Part of the emissions analysis was also transferred from the supplement to the main text. Unfortunately, however, seasonal variations are not provided for most species (see also supplementary Fig. S4).

*In the response to reviewer 2 there is a figure with CEOS emission. Now it would be interesting if there is a seasonality to these emissions or do they just provided annual averages?*

While CEDS emissions data are available on a monthly basis, for most species there is a constant value per year, hence a seasonality analysis was not possible. The overall changes in emissions during the study period were however included in the revised manuscript (Fig. 4).

*Overcast cloud conditions: since these are required for bi-spectral retrievals methods (COT, reff) I wonder if that frequency changed? Are there other properties that can provide insights on why the cloud properties have changed? Did the cloud-top change?*

We have additionally checked atmospheric water vapor and cloud top height searching for insights on these changes. No significant change was found in water vapor concentrations, while cloud top height has increased, but again, not significantly in most cases. In fact, there are discrepancies between CLARA-A2 and MODIS regarding CTH analysis, that render further conclusions dubious (see also supplementary Fig. S8 and Table S10).

***minor comments to the new text***

*In the abstract you talk about a 40% AOD reduction. This is an exaggeration. My seasonal analysis shows ca 30% reduction between 2006 and 2015 only for fall (other seasons are much less) and a significant part of the reduction (ca 30%) is related to coarse mode aerosol (which little link to GFED emissions).*

We thank the reviewer for this remark. This number refers to the highest increase in AOD found, which occurred in September-October only. In November-December, to which this sentence in the abstract refers, the decrease was less (~35% based on MODIS and MISR, see also Table 1). We have corrected the abstract accordingly.
Coarse mode aerosol from MISR was previously not included, but it is now. Indeed it explains a part of the overall AOD reduction, but only in autumn, and specifically in September and October, when the change is statistically significant (see also Fig. 3a of the revised manuscript).

*In the abstract the last sentence comes across as a statement but is highly speculative at best.*

The purpose of this sentence was to explain how the proposed mechanism would work. However, we understand the reviewer's concern and we have edited the last part of the abstract, stating explicitly that this is not a statement proven by our results.

*Aerosol results: why is the dust AOD so small? AODc (mainly dust over continents) and AODf should add up to AOD. … and please show industrial emission change for that region, preferably with a seasonal cycle (in the warmer/humid summer the pollution related AOD should be larger).*

A possible explanation for the low dust AOD found is the long distance of the study region from the main dust sources, i.e. the Gobi and Taklimakan deserts. This is included in page 5, lines 11-12 of the revised manuscript. Coarse mode AOD was previously not included. It is now, and indeed along with fine mode they add up to total AOD. We have also moved the analysis results on emissions from the supplement to the main text. Unfortunately, the CEDS data set does not provide seasonal information for most emitted species (see also supplementary Fig. S4).

*The discussion section is much improved. I like the idea with the reduced semi-direct effect. Would this not also imply a more unstable atmosphere and with a stronger convection a higher cloud top?*

Indeed we found a higher cloud top based on both CLARA-A2 and MODIS data (page 7, lines 1-7 and Fig. S8). However, since there are some discrepancies between the two data sets, we avoided further conclusions based on this increase.

*Assuming the altitude assignment of AOD change is correct … what can be reason that elevated AOD is so much reduced in fall? why not at the ground?*

A possible explanation already mentioned would be that these aerosols were transported from other regions. Hence, their concentrations and changes would be disentangled from local sources and corresponding changes. However, we have not been able to identify the causes of the reduction of elevated AOD in September-October.

**Reply to Referee #3**

We thank the anonymous referee for reviewing our manuscript and providing comments for revision. Following are our point-by-point replies, with the referee comments in italic. Please note that figure, page and line numbers refer to the marked-up version of the revised manuscript and supplement, attached here after our responses.

*Their study highlighted biomass burning as the major source to cause the changes in cloud, while the contribution of other sources was not discussed. Past studies have reported, during the study period (2006-2015), the reduction in emissions was observed not only for biomass burning but also for other sources. Because the emissions in southern China were very complicated, the contribution of other emission sources should be considered. Also, in Figure 3, the emissions of biomass burning in J-S showed an increase but the AOD decreased; From Oct to Nov, the emissions decreased but the AOD increased. These results do not seem to support their findings.*

Other emission sources and their changes are indeed important and were added for completeness following the referee's suggestion. Specifically, we expanded the analysis regarding emissions from the Community Emissions Data System (CEDS) and transferred respective results from the supplement to the main text (Fig. 4 of the revised manuscript). However, the reduction in emissions reported elsewhere over China is not verified for this specific region and period based on CEDS data. In addition to other emission sources, precipitation data from the Global Precipitation Climatology Project (GPCP) were also analyzed, revealing coincidences in increasing precipitation and decreasing aerosols. Hence, wet removal could partially explain the reported decrease in AOD.

Figure 3 should be interpreted on an "individual month" basis. For example, in November both AOD and biomass burning emissions have decreased substantially, as in December. In this sense, the referee correctly points that AOD and biomass burning emissions changes are not always correlated. For example, in October the change in AOD is substantial, whereas there is practically no change in biomass burning emissions. We attribute these differences to the fact that biomass burning emissions alone cannot always characterize the full aerosol load, as the referee also pointed above.

*Second, natural climate conditions should be considered in the analyses. Their findings claimed the changes in cloud were mainly due to aerosols. But natural climate variability must be a critical factor to affect cloud. Without considering natural climate variability, their results may not fully reflect the situation.*

Natural climate variability was considered by analyzing the surface pressure and 500 hPa geopotential height in Section 4.1. In the revised manuscript we expanded this analysis with relevant maps of average values and changes included in the supplement (Fig. S11, and page 8, lines 26-34 of the revised manuscript). We also included analysis of precipitation, which can affect aerosol and cloud conditions through natural variability. A statistically significant increase in precipitation was found in November-December, providing an additional possible explanation for the decrease in aerosols during these months. However, no significant correlation was found with the changes in cloud optical properties. This new precipitation analysis in discussed in page 6, lines 8-16 of the revised manuscript.

*Third, climate change was definitely a critical factor to influence annual variability of aerosol distribution and concentration. However, climate change was not assessed systemically in their analyses.*

Climate change can indeed affect aerosol and cloud distributions through various ways, e.g. by altering large scale phenomena, such as the Asian Monsoon and the ENSO, which were examined in Section 4.1. In the revised manuscript, we also included an analysis on surface air temperature, which is the main parameter describing climate change. Data from the ERA-Interim reanalysis data set were used for this purpose. No significant change was detected in surface air temperature during the 10-year study period. Interestingly, however, a strong decrease is found in November-December, coinciding with the significant increase in cloud properties. These results are discussed in page 9, lines 8-14 of the revised manuscript.

*The robustness of the results should be investigated. I suggest to conduct statistically significance tests for the trend of cloud (spatial distribution). This would help readers to better understand the results.*

Apart from the Tables 1 and 2, which report on the statistical significance of our results and their correlations, respectively, we have also included in the supplement Table S3, which provides, among others, p-values of changes in AOD and cloud properties during the period examined, Table S9, with levels of significance of cloud changes on a monthly basis, and Table S10, showing significant correlations among AOD and cloud properties. Following the reviewer's suggestion, in the revised manuscript we have included maps with information on the statistical significance of changes in all cloud variables examined, separately from CLARA-A2 and MODIS. Due to the number of maps involved, these were added in the supplement (Figs. S6 and S7).

*I agree with the point mentioned in line 3-5 in page 3. Considering local emissions alone may cause significant problems. So, considering long-range transport is necessary. "It should be noted that, due to the long-range transport of aerosols, local aerosol emissions are not expected to fully explain corresponding properties and characteristics of aerosol types and loads in the atmosphere of the same region."*

Considering local biomass burning emissions alone when discussing the origin of the aerosol load over an area would indeed be an incomplete approach. For this reason, information on other emission sources was also included. However, when focusing on a specific area and discussing possible reasons of changes in aerosol load, we consider it sufficient to examine changes in local emissions, and also acknowledge possible effects from long-range transport; analyzing long-range transport on an equivalent basis as local emissions, would require analyses of both air mass trajectories and emission changes in nearby areas, extending the study beyond its specified scope.

[revised manuscript text omitted]

**Figure S2. Changes in total,  fine and coarse mode AOD over southern China during 2006-2015 deduced from MISR data. (a, c, e) Spatial distributions of total  fine and coarse mode AOD change over the study region and (b, d, f) corresponding levels of statistical significance. The black dots in (b), (d) and (f) highlight the grid cells where the level of statistical significance is higher than 95%. White areas correspond to grid cells where the threshold regarding time series completeness was not met.**

**Table S3. Statistical measures of changes in AOD and cloud properties over southern China in 2006-2015 based on data products from CALIPSO, MISR, MODIS and CLARA-A2. Measures comprise percent changes, slopes and p-values.**

| Parameter | Unit | CALIPSO | MISR | MODIS | CLARA-A2 |
|---|---|---|---|---|---|
| | | Change (%)/slope ( $yr^{-1}$)/p-value | Change (%)/slope ( $yr^{-1}$)/p-value | Change (%)/slope ( $yr^{-1}$)/p-values | change (%)/slope ( $yr^{-1}$)/p-value |
| Total AOD | 1 | -23.3/-0.013/0.013 | -17.9/-0.008/0.007 | -17.6/-0.010/0.002 | |
| Fine AOD | 1 | | -20.7/-0.005/0.005 | | |
| Coarse AOD | 1 | | -14.3/<-0.001/0.042 | | |
| Dust AOD | 1 | | -13.1/-0.001/0.332 | | |
| | | | | | |
| All-sky LWP | g m $^{-2}$ | | | +12.4/0.837/0.204 | +14.2/0.913/0.242 |
| Liquid CFC | 1 | | | +6.8/0.003/0.219 | +3.4/0.002/0.465 |
| Liquid COT | 1 | | | +5.5/0.089/0.399 | +3.6/0.058/0.607 |
| Liquid REFF | μm | | | +1.6/0.018/0.239 | +5.2/0.034/0.0003 |

[Figure]

**Figure S4. Seasonal variation of anthropogenic emissions of aerosols and precursor gases in the southern China study region from the Community Emissions Data System (CEDS). Monthly emissions have been averaged over the period 2006-2014. Reported values refer to full molecular mass for $SO_2$, $NO_x$, and $NH_3$ and to carbon mass for black and organic carbon.**

[Figure]

Figure S5. Emissions of aerosols and precursor gases from the Community Emissions Data System (CEDS). The emissions have been aggregated to annual totals over the southern China study area and plotted relative to the year 2006.

[Figure]

**Figure S56. Levels of statistical significance for CLARA-A2 all-sky LWP and liquid CFC changes over southern China, calculated for the period 2006-2015. The black dots highlight the grid cells where the level of statistical significance is higher than 95%.**

[Figure]

**Figure S6. Spatial distributions (a, c) and levels of statistical significance (b, d) for CLARA-A2 liquid COT and REFF changes over southern China, calculated for the period 2006-2015. The yellow dots in (b) and (d) highlight the grid cells where the level of statistical significance is higher than 95%.**

[Figure]

**Figure S7. Spatial distributions (a, c, e ,g) and levels of statistical significance (b, d, f, h) for MODIS all-sky LWP, liquid CFC, COT and REFF changes over southern China, calculated for the period 2006-2015. The yellow dots in (b), (d), (f) and (h) highlight the grid cells where the level of statistical significance is higher than 95%.**

[Figure]

**Figure S8. (a) Seasonal variation of Cloud Top Height (CTH) in the southern China study region based on CLARA-A2 and MODIS data. (b) Corresponding spatially averaged monthly deseasonalized values. The dotted lines correspond to the linear regression fits. (c) Seasonal variation of changes in CTH. Seasonal averages and changes in (a) and (c) are based on data from the period 2006-2015.**

**Table S79**. Levels of statistical significance of changes in cloud properties on a monthly basis, calculated from the period 2006-2015 separately for CLARA-A2 and MODIS.

| | All-sky LWP | | Liquid CFC | | In-cloud LWP | | Liquid COT | | Liquid REFF | |
|---|---|---|---|---|---|---|---|---|---|---|
| | CLARA | MODIS | CLARA | MODIS | CLARA | MODIS | CLARA | MODIS | CLARA | MODIS |
| Jan | 14.13 | 13.49 | 54.18 | 39.26 | 6.15 | 5.13 | 3.59 | 6.41 | 14.13 | 56.52 |
| Feb | 10.79 | 3.68 | 44.67 | 39.57 | 10.60 | 15.01 | 5.28 | 1.13 | 14.10 | 52.25 |
| Mar | 5.52 | 15.19 | 1.47 | 12.82 | 12.85 | 7.65 | 12.19 | 11.08 | 47.77 | 45.10 |
| Apr | 33.05 | 42.88 | 28.30 | 4.58 | 58.39 | 61.04 | 50.12 | 54.49 | 73.79 | 72.08 |
| May | 79.19 | 94.20 | 61.31 | 30.53 | 96.01 | 99.22 | 85.28 | 95.12 | 86.93 | 48.38 |
| Jun | 13.69 | 26.27 | 85.18 | 49.95 | 39.64 | 62.92 | 36.34 | 55.18 | 61.09 | 59.25 |
| Jul | 22.71 | 39.33 | 2.57 | 56.52 | 26.96 | 26.94 | 23.06 | 2.27 | 90.11 | 93.16 |
| Aug | 4.06 | 77.65 | 57.83 | 42.97 | 74.50 | 29.39 | 34.42 | 1.00 | 86.12 | 97.35 |
| Sep | 36.62 | 27.02 | 55.97 | 37.12 | 30.88 | 33.54 | 21.79 | 22.90 | 92.01 | 46.53 |
| Oct | 46.82 | 69.47 | 74.85 | 81.51 | 72.35 | 70.81 | 59.18 | 66.37 | 79.20 | 50.22 |
| Nov | 90.07 | 90.99 | 99.26 | 99.55 | 65.37 | 86.26 | 48.39 | 78.23 | 89.59 | 88.76 |
| Dec | 97.05 | 99.33 | 35.75 | 54.15 | 99.43 | 99.91 | 96.45 | 98.57 | 97.46 | 67.40 |

**Table S810**. Linear correlation coefficients of two-month-mean emission and AOD time series with cloud property time series over southern China during the period 2006-2015 (2007-2015 for CALIPSO AOD). Significant correlations are indicated with boldface.

| parameter | Jan-Feb/Mar-Apr/May-Jun/Jul-Aug/Sep-Oct/Nov-Dec | |
|---|---|---|
| | | |
| | CLARA liquid CFC | MODIS liquid CFC |
| GFED carbon emissions | -0.60/**-0.79**/0.01/0.47/-0.07/-0.51 | **-0.64**/**-0.72**/-0.32/0.34/0.01/-0.51 |
| CALIPSO total AOD | -0.22/-0.59/0.34/**0.77**/-0.06/**-0.77** | -0.22/-0.50/0.44/0.39/-0.16/**-0.75** |
| MODIS total AOD | -0.45/0.33/0.57/0.15/0.27/**-0.76** | -0.44/0.16/0.46/0.36/0.38/**-0.81** |
| MISR total AOD | -0.51/0.30/0.29/-0.20/0.23/**-0.66** | -0.46/0.15/0.29/0.05/0.23/**-0.74** |
| MISR fine mode AOD | -0.39/0.31/0.45/-0.12/0.26/**-0.66** | -0.38/0.15/0.44/0.21/0.25/**-0.74** |
| MISR coarse mode AOD | -0.57/0.30/0.13/-0.28/0.14/-0.62 | -0.50/0.19/0.14/-0.25/0.14/**-0.69** |
| | | |
| | CLARA all-sky LWP | MODIS all-sky LWP |
| GFED carbon emissions | -0.47/-0.40/-0.25/0.59/0.05/**-0.69** | -0.46/-0.20/-0.39/0.07/0.02/**-0.75** |
| CALIPSO total AOD | -0.16/-0.31/0.55/0.41/-0.31/**-0.69** | -0.15/-0.19/0.57/-0.56/-0.35/**-0.71** |
| MODIS total AOD | -0.45/-0.13/0.52/-0.06/-0.21/**-0.75** | -0.41/-0.34/0.29/-0.02/-0.37/**-0.84** |
| MISR total AOD | -0.36/-0.25/0.38/-0.31/-0.24/**-0.66** | -0.27/-0.49/0.28/0.08/-0.48/**-0.81** |
| MISR fine mode AOD | -0.17/-0.16/0.52/-0.22/-0.27/**-0.70** | -0.11/-0.36/0.37/0.15/-0.50/**-0.84** |
| MISR coarse mode AOD | -0.55/-0.29/0.22/-0.43/-0.27/-0.55 | -0.47/-0.55/0.16/-0.12/-0.49/**-0.72** |
| | | |
| | CLARA CTH | MODIS CTH |
| GFED carbon emissions | 0.22/0.13/-0.10/-0.30/-0.21/-0.06 | -0.06/0.26/0.03/-0.40/-0.38/**-0.66** |
| CALIPSO total AOD | 0.44/0.62/0.21/-0.40/-0.01/0.34 | 0.20/0.25/0.13/-0.63/0.10/-0.34 |
| MODIS total AOD | 0.08/-0.04/-0.32/-0.34/-0.53/-0.14 | -0.44/0.11/0.09/-0.39/-0.49/**-0.74** |
| MISR total AOD | 0.30/-0.29/0.04/-0.13/-0.43/-0.27 | -0.14/-0.50/0.59/-0.06/-0.36/**-0.73** |
| MISR fine mode AOD | 0.19/-0.19/-0.10/-0.25/-0.46/-0.30 | -0.12/-0.33/0.41/-0.13/-0.39/**-0.78** |

| MISR coarse mode AOD | 0.45/-0.35/0.19/0.16/-0.35/-0.21 | -0.07/-0.61/**0.73**/0.11/-0.26/-0.60 |
| --- | --- | --- |
| | GPCP Precipitation | |
| CALIPSO total AOD | -0.03/0.35/0.50/-0.59/-0.44/-0.25 | |
| MODIS total AOD | -0.22/0.44/0.56/-0.12/-**0.82**/-0.63 | |
| MISR total AOD | 0.10/0.50/0.61/0.20/-**0.66**/-**0.70** | |
| MISR fine mode AOD | 0.19/0.41/0.63/0.09/-**0.68**/-**0.71** | |
| MISR coarse mode AOD | -0.02/0.57/0.49/0.34/-0.55/-**0.65** | |

[Figure]

Figure S11. Average spatial distributions and corresponding changes in 500 hPa geopotential height (a and b, respectively) and surface pressure (c and d, respectively) from the Copernicus Atmospheric Monitoring Service (CAMS) reanalysis over a wide area of southeast Asia, centered around the southern China study region. Averages were computed from monthly values during the period 2006–2015. Pixel–level changes were computed based on linear regressions fits to the deseasonalized monthly time series.

---

## Author Response (AR3)

We would like to thank the reviewer for reviewing again our manuscript. Following are our point-by-point replies, with the reviewer's comments in italic. Please note that figure, page and line numbers refer to the marked-up version of the revised manuscript and supplement, attached here after our responses.

*My comments have not been well addressed. In addition, it is not easy to follow the changes in the manuscript. My recommendation of major changes is still remained.*

Our replies below provide updates and clarifications regarding the way that the initial comments, repeated here by the reviewer, are addressed in the revised manuscript. Fully addressing the comments by both reviewers may indeed lead to parts of the study becoming less clear. We tried to minimize this effect by including most of the additional, detailed results in the supplement. In this way, the flow of the manuscript remains consistent, while the reader retains the option to digress to a higher level of detail provided in the supplementary material.

*"Their study highlighted biomass burning as the major source to cause the changes in cloud, while the contribution of other sources was not discussed. Past studies have reported, during the study period (2006-2015), the reduction in emissions was observed not only for biomass burning but also for other sources. Because the emissions in southern China were very complicated, the contribution of other emission sources should be considered. Also, in Figure 3, the emissions of biomass burning in J-S showed an increase but the AOD decreased; From Oct to Nov, the emissions decreased but the AOD increased. These results do not seem to support their findings."*

*Reply: This comment is not well addressed. The abstract still highlights "Analyses of emissions and precipitation changes suggest that a decrease in biomass burning aerosols and an increase in precipitation played important roles in the overall aerosol reduction. But I don't think it is correct as the relationship between the changes in biomass burning aerosols and precipitation are not clear.*

This part of the abstract has been rewritten to clarify the possible role of biomass burning in overall reductions in emissions, and also to address the reviewer's concern on a possible relationship between changes in biomass burning and precipitation. A relationship might indeed exist in two ways. On one hand, through a "reversed" aerosol semi-direct effect, less absorbing aerosols in the cloud layer would lead to less evaporation of cloud droplets, larger cloud amount and liquid water path (hypothesized as an active mechanism in our study), and possibly more precipitation. On the other hand, more precipitation would lead to wetter conditions and therefore less biomass burning emissions as well as to a more effective removal of aerosol from the atmosphere by increased wet deposition. In either case, our previous statement would still hold, although it would indeed be incomplete.

Regarding the contributions of other emission sources, apart from biomass burning aerosols, in the overall aerosol reduction, they were considered in the previous manuscript version, based on data from the Community Emissions Data System (CEDS). This is a recent, well-documented and widely used emissions data set (Hoesly et al. 2018). Our results based on analysis of annual data showed increases in CEDS emissions. Unfortunately, CEDS emission data lack information on their seasonal variation, providing only annual values (although on a monthly basis) with no seasonality information for most species. For this reason, it was not possible to extend the analysis to changes on a monthly basis, as was done for AOD and GFED emissions. This was clearly acknowledged in our analysis.

Hence, we did not claim that the emissions of all the species except $NH_3$ are constant throughout a year, as the referee suggests in the final comment, based on Fig. S4 of the previous supplement version. We merely illustrated this lack of information in CEDS. However, we understand that this approach can lead to misinterpretations, while it also artificially highlights the role of biomass burning emissions. For these reasons, in the revised manuscript we have replaced the CEDS data set with the Copernicus Atmosphere Monitoring Service (CAMS) global anthropogenic emissions inventory (Granier et al. 2019), which provides emission information on a monthly basis for a multitude of species, including the major contributors over southern China, and, in contrast to CEDS, encompasses the final year of our study period, 2015. Combined with the data from GFED, this data set offers a more complete analysis of all emission sources in the study region.

Results on the overall emission changes during the study period from CAMS are in good agreement with the ones previously reported based on CEDS, i.e. increases in all major species (page 9, lines 11-17, Fig. 5). Based on the additional monthly information, this increase appears evenly distributed throughout the year. These results seem at first contradictory to results provided in other studies, showing decreases in emissions over China. A main reason for these discrepancies is the differences in the specific areas and time periods examined. Emission sources in China are not uniformly distributed throughout the country, and different political decisions led to changes in emission tendencies in different years (e.g. van der A et al. 2017). On the other hand, it cannot be excluded that the emission inventories do not fully capture the most recent trends, as is acknowledged in the revised version (page 9, lines 16-17).

The changes applied based on the reviewer's comment are summarized as follows:

- The part of the abstract mentioned by the reviewer was rewritten (page 1, lines 12-18).
- All major emission species were studied along with biomass burning. For this purpose, the CEDS data set was replaced with CAMS, which includes more information on monthly variations.

*"Second, natural climate conditions should be considered in the analyses. Their findings claimed the changes in cloud were mainly due to aerosols. But natural climate variability must be a critical factor to affect cloud. Without considering natural climate variability, their results may not fully reflect the situation."*

*Reply: Again, natural climate variability is critical for such long-term analysis. The authors add an analysis of precipitation as they claim precipitation can affect aerosol and cloud conditions through natural variability. However, aerosols themselves have impact on precipitation through cloud. Analyzing precipitation alone does not definitely help understand the effect of natural variability.*

Our 10-year study period is too short to effectively differentiate the effect of natural variability from other mechanisms examined here (changes in local emissions, aerosol-cloud interactions, changes in precipitation). It is generally recommended that time periods of the order of decades be used to this end. In our case, all aerosol and cloud data sets except for CLARA-A2 do not span that long periods. In the case of CLARA-A2 we show that cloud changes during the last years are indeed exceptional compared to previous decades.

We agree with the reviewer that analysis of precipitation only would not provide a complete overview of possible effects of natural variability. However, our assessment of possible effects of natural variability was not limited to precipitation only. It also included analysis of the ENSO and AM cycles, as

well as surface pressure and 500 hPa geopotential height changes. The first two phenomena largely describe natural variability over the region, and any possible change related to it would also manifest there. Changes related to atmospheric dynamics and circulation would also appear in the surface pressure and geopotential height data. Furthermore, as explained in our response to the next comment, aerosol changes over this region are probably better explained based on anthropogenic factors, rather than natural variability. We had included such an analysis and expanded it based on the reviewer's comment in the previous version. Results show that no circulation pattern change could explain the changes in clouds reported here.

Summary of study points relevant to the reviewer's comment:

- Section 4.1 discusses possible mechanisms apart from aerosol and cloud interactions that could explain the aerosol and cloud changes reported before. These mechanisms are strongly related to natural climate variability.

*"Third, climate change was definitely a critical factor to influence annual variability of aerosol distribution and concentration. However, climate change was not assessed systemically in their analyses."*

*Reply: similar to the responses in the previous two comments, this comment has not yet been addressed. Surface temperature alone may not fully reflect the effect of climate change, which can affect dynamics for example. More effort should be put on addressing this concern.*

Based on the Intergovernmental Panel on Climate Change (IPCC) definition of climate change, which refers to "changes in the mean and/or the variability of its properties that persist for an extended period, typically decades or longer" (IPCC, 2018), it is not possible to examine the effect of climate change in changes taking place within only 10 years, as in our study. However, the more commonly used United Nations Framework Convention on Climate Change (UNFCCC) definition, relates climate change to human activities that alter the composition of the global atmosphere (IPCC, 2018), and most of the times refers to $CO_2$ and consequent air temperature changes. While feedback mechanisms can be much more complicated, affecting also atmospheric dynamics, as the reviewer suggests, we consider logical to examine air temperature as a first indicator of climate change effects. We agree, however, that this is not enough, since climate change could also affect dynamics. For this reason, changes in atmospheric circulation patterns were also examined through the analysis of surface pressure and 500 hPa geopotential height changes.

Regarding aerosol distributions and concentrations and possible climate change effects: aerosol regimes are determined by processes describing emissions, atmospheric transformations and deposition. The dependency of these processes on climate change varies considerably among aerosol sources and types, while other, local factors, may play a more important role. In this sense, climate change is not always a critical factor influencing annual variability of aerosol distribution and concentration, since other, local factors, may play an equal or even more important role. Climate change is indeed more obvious where aerosols from natural sources prevail, e.g. desert dust and marine salt. In these cases, it can affect critically the aerosol regime mainly through changes in atmospheric dynamics, as the reviewer mentions. In areas where aerosols come mainly from anthropogenic sources, however, including the wider South and Southeast Asia regions (Zhang et al., 2012), possible effects of climate change on aerosols will manifest mainly in terms of transportation and deposition, since the main factors affecting emissions are economic development and

environmental policies (Chin et al. 2014). Overall, effects of climate change can be indirect, and affect aerosol transformation and deposition, through atmospheric variables like temperature and wind speed (Tegen and Schepanski, 2018).

As we mentioned before, in our study we have examined possible changes in both air temperature and atmospheric dynamics. Our results showed no significant change in either. While we had expanded this analysis and the presentation of relevant results based on the reviewer's earlier comment, we understand the reviewer's concern and we have included this more detailed discussion in the revised manuscript.

Summary of changes applied to address the reviewer's initial and updated comment:

- Analysis of surface air temperature as an indicator of possible effects of climate change in the study region (page 19, lines 3-9).
- Analysis of surface pressure and 500 hPa geopotential height to assess changes in atmospheric dynamics, also possibly affected by climate change (page 19, lines 10-24).
- Expansion of the discussion on how aerosols can be affected by climate change, justifying the approach adopted (page 18, lines 14-26).

*"The robustness of the results should be investigated. I suggest to conduct statistically significance tests for the trend of cloud (spatial distribution). This would help readers to better understand the results."*

*Reply: the explanation about the statistically significance test results is not clear that make readers hard to understand.*

Based on the reviewer's initial comment, repeated above, but also on similar concerns expressed earlier in the review process, we have decided to include results of statistical significance tests extensively in our analysis. This was also motivated by acknowledging that the relevant statistical information is crucial before drawing any conclusion, and should be accessible to the reader of the study. Trying to balance this need with the risk of leading readers to confusion, we decided to include the largest part of these statistical measures in the supplement. In this way, the reader is offered both access to this crucial information and an uninhibited read of the main study. In order to further reconcile these opposing needs and address the reviewer's new concern, we have tried to simplify the relevant discussion in parts of the results analysis (page 6, lines 13-18, page 14, lines 9-11 and page 16, lines 9-10).

*"I agree with the point mentioned in line 3-5 in page 3. Considering local emissions alone may cause significant problems. So, considering long-range transport is necessary.*
*"It should be noted that, due to the long-range transport of aerosols, local aerosol emissions are not expected to fully explain corresponding properties and characteristics of aerosol types and loads in the atmosphere of the same region.""*

*Reply: I don't agree with this response. The studied region has significant local emission sources and long-range transport, while biomass burning emissions are just one of the emission sources. This study uses the method of satellite observations of aerosols. So, it is necessary considering other sources and long-range transport.*

As explained above, other emission sources were considered by analyzing data from the CAMS emissions inventory, which replaced the CEDS data set from the previous manuscript version. In the revised manuscript, this analysis is combined with the biomass burning emissions analysis, to provide a complete overview of local emissions.

A long-range transport analysis was also performed, based on the reviewer's comment. For this purpose, the Hybrid Single-Particle Lagrangian Integrated Trajectory (HYSPLIT) model was used. HYSPLIT is a public domain model (ready.arl.noaa.gov/HYSPLIT.php), suitable for analyzing air mass trajectories (Draxler and Hess, 1998). For the present study, the analysis setup was as follows: October was selected to be analyzed in terms of long-range transport, as it is the month with the largest discrepancies between changes in AOD (large decrease, see also Fig. 3a) and local emissions (small increase, see also Figs. 3b and 3c). For October 2007 and 2015, representing conditions close to the beginning and end of the study period respectively, HYSPLIT was run for trajectories starting every 6 hours and lasting 24 hours each, during the whole month, at two heights, 500 m and 1500 m above sea level. Both forward- and back-trajectories were analyzed, starting and ending at the center of the study region. The forward-trajectories analysis was performed to give insights into the degree of dispersion of locally-emitted aerosol loads outside of the study region, while the back-trajectory analysis reveals how frequently the air masses found in the study region originate outside of it. For the quantification of these properties, results are displayed in terms of frequency of trajectory occurrence, on a pixel basis, relative to the total number of trajectories.

The results of this analysis reveal that in all cases more than 90% of the forward trajectories end up within the study region and more than 90% of the back-trajectories originate inside the study region. In fact, these high-probability areas are much smaller than the study region, suggesting that even for points near the edge of the study region, instead of its center, the contributions from adjacent areas will be much lower than the local emissions.

The changes applied based on the reviewer's comment are summarized as follows:

- Analysis of other sources, apart from biomass burning, was conducted based on CAMS data, which replaced the CEDS data set.
- An air mass trajectory analysis was conducted to assess possible contributions of adjacent regions in the local AOD (page 10, lines 1-2, page 11, lines 1-15 and supplementary Figs. S5 and S6).

*One new comment. Figure S4 does not sound to be correct. First of all, the emissions of all the species except NH3 are constant throughout a year. References should be provided to support this. Also, averaging each month in the period of 2006-2014 is misleading as the monthly variations in far past and recent years may be different.*

As explained in a previous comment, the purpose of Fig. S4 was to highlight the lack of seasonal information in emissions from CEDS, and the inadequacy of this data set to perform an analysis of changes on a monthly basis, rather than show that emissions are constant throughout a year. In order to avoid similar misinterpretations, and also to include seasonal variation of all emission sources apart from biomass burning, we have decided to replace CEDS with the CAMS emissions inventory (Granier et al. 2019), which contains this additional information. Hence, this figure was replaced in the revised supplement by the corresponding information from CAMS data, which includes monthly information.

Regarding the reviewer's last remark, we agree that monthly values in far past and recent years may be different. The purpose of this figure, however, is to give information on the variation of emissions within a typical year of this period, which can be represented by average values.

[revised manuscript text omitted]

**Figure S2. Changes in total, fine and coarse mode AOD over southern China during 2006-2015 deduced from MISR data. (a, c, e) Spatial distributions of total, fine and coarse mode AOD change over the study region and (b, d, f) corresponding levels of statistical significance. The black dots in (b), (d) and (f) highlight the grid cells where the level of statistical significance is higher than 95%. White areas correspond to grid cells where the threshold regarding time series completeness was not met.**

**Table S3.** Statistical measures of changes in AOD and cloud properties over southern China in 2006-2015 based on data products from CALIPSO, MISR, MODIS and CLARA-A2. Measures comprise percent changes, slopes and p-values.

| Parameter | Unit | CALIPSO | MISR | MODIS | CLARA-A2 |
|---|---|---|---|---|---|
| | | Change (%)/slope ( yr$^{-1}$)/p-value | Change (%)/slope ( yr$^{-1}$)/p-value | Change (%)/slope ( yr$^{-1}$)/p-values | change (%)/slope ( yr$^{-1}$)/p-value |
| Total AOD | 1 | -23.3/-0.013/0.013 | -17.9/-0.008/0.007 | -17.6/-0.010/0.002 | |
| Fine AOD | 1 | | -20.7/-0.005/0.005 | | |
| Coarse AOD | 1 | | -14.3/<-0.001/0.042 | | |
| Dust AOD | 1 | | -13.1/-0.001/0.332 | | |
| | | | | | |
| All-sky LWP | g m$^{-2}$ | | | +12.4/0.837/0.204 | +14.2/0.913/0.242 |
| Liquid CFC | 1 | | | +6.8/0.003/0.219 | +3.4/0.002/0.465 |
| Liquid COT | 1 | | | +5.5/0.089/0.399 | +3.6/0.058/0.607 |
| Liquid REFF | μm | | | +1.6/0.018/0.239 | +5.2/0.034/0.0003 |

[Figure]

**Figure S4. Seasonal variation in aerosols and precursor gases from the Copernicus Atmosphere Monitoring Service (CAMS), relative to the annual emission of each species, over southern China during 2006-2015.**

[Figure]

**Figure S5. Forward trajectories frequency calculated based on HYSPLIT model for October 2007 (a, c) and 2015 (b, d), starting at the center of the study region (22.5° N, 110.0° E) and at 500 m (c ,d) and 1500 m (a, b) above sea level (a.s.l.). The patterns show the frequency of trajectory occurrence (in % relative to the total number of trajectories), that end in each pixel. The study region is indicated by a black rectangle.**

[Figure]

**Figure S6.** Back-trajectories frequency calculated based on HYSPLIT model for October 2007 (a, c) and 2015 (b, d), ending at the center of the study region (22.5° N, 110.0° E) and at 500 m (a, b) and 1500 m (c, d) above sea level (a.s.l.). The patterns show the frequency of trajectory occurrence (in % relative to the total number of trajectories), that start in each pixel. The study region is indicated by a black rectangle.

Figure S4. Seasonal variation of anthropogenic emissions of aerosols and precursor gases in the southern China study region from the Community Emissions Data System (CEDS). Monthly emissions have been averaged over the period 2006-2014. Reported values refer to full molecular mass for $SO_2$, $NO_x$ and $NH_3$ and to carbon mass for black and organic carbon.

[Figure]

**Figure S75. Levels of statistical significance for CLARA-A2 all-sky LWP and liquid CFC changes over southern China, calculated for the period 2006-2015. The black dots highlight the grid cells where the level of statistical significance is higher than 95%.**

[Figure]

**Figure S86. Spatial distributions (a, c) and levels of statistical significance (b, d) for CLARA-A2 liquid COT and REFF changes over southern China, calculated for the period 2006-2015. The yellow dots in (b) and (d) highlight the grid cells where the level of statistical significance is higher than 95%.**

[Figure]

**Figure S79. Spatial distributions (a, c, e ,g) and levels of statistical significance (b, d, f, h) for MODIS all-sky LWP, liquid CFC, COT and REFF changes over southern China, calculated for the period 2006-2015. The yellow dots in (b), (d), (f) and (h) highlight the grid cells where the level of statistical significance is higher than 95%.**

[Figure]

**Figure S108.** (a) Seasonal variation of Cloud Top Height (CTH) in the southern China study region based on CLARA-A2 and MODIS data. (b) Corresponding spatially averaged monthly deseasonalized values. The dotted lines correspond to the linear regression fits. (c) Seasonal variation of changes in CTH. Seasonal averages and changes in (a) and (c) are based on data from the period 2006-2015.

**Table S11. Levels of statistical significance of changes in cloud properties on a monthly basis, calculated from the period 2006-2015 separately for CLARA-A2 and MODIS.**

| | All-sky LWP | | Liquid CFC | | In-cloud LWP | | Liquid COT | | Liquid REFF | |
|------|-------|-------|-------|-------|-------|-------|-------|-------|-------|-------|
| | CLARA | MODIS | CLARA | MODIS | CLARA | MODIS | CLARA | MODIS | CLARA | MODIS |
| Jan | 14.13 | 13.49 | 54.18 | 39.26 | 6.15 | 5.13 | 3.59 | 6.41 | 14.13 | 56.52 |
| Feb | 10.79 | 3.68 | 44.67 | 39.57 | 10.60 | 15.01 | 5.28 | 1.13 | 14.10 | 52.25 |
| Mar | 5.52 | 15.19 | 1.47 | 12.82 | 12.85 | 7.65 | 12.19 | 11.08 | 47.77 | 45.10 |
| Apr | 33.05 | 42.88 | 28.30 | 4.58 | 58.39 | 61.04 | 50.12 | 54.49 | 73.79 | 72.08 |
| May | 79.19 | 94.20 | 61.31 | 30.53 | 96.01 | 99.22 | 85.28 | 95.12 | 86.93 | 48.38 |
| Jun | 13.69 | 26.27 | 85.18 | 49.95 | 39.64 | 62.92 | 36.34 | 55.18 | 61.09 | 59.25 |
| Jul | 22.71 | 39.33 | 2.57 | 56.52 | 26.96 | 26.94 | 23.06 | 2.27 | 90.11 | 93.16 |
| Aug | 4.06 | 77.65 | 57.83 | 42.97 | 74.50 | 29.39 | 34.42 | 1.00 | 86.12 | 97.35 |
| Sep | 36.62 | 27.02 | 55.97 | 37.12 | 30.88 | 33.54 | 21.79 | 22.90 | 92.01 | 46.53 |
| Oct | 46.82 | 69.47 | 74.85 | 81.51 | 72.35 | 70.81 | 59.18 | 66.37 | 79.20 | 50.22 |
| Nov | 90.07 | 90.99 | 99.26 | 99.55 | 65.37 | 86.26 | 48.39 | 78.23 | 89.59 | 88.76 |
| Dec | 97.05 | 99.33 | 35.75 | 54.15 | 99.43 | 99.91 | 96.45 | 98.57 | 97.46 | 67.40 |

Table S12. Linear correlation coefficients of two-month-mean GFED BC and OC emission and AOD time series with cloud property time series over southern China during the period 2006-2015 (2007-2015 for CALIPSO AOD). Significant correlations are indicated with boldface.

| parameter | Jan-Feb/Mar-Apr/May-Jun/Jul-Aug/Sep-Oct/Nov-Dec | |
|---|---|---|
| | CLARA liquid CFC | MODIS liquid CFC |
| GFED  emissions | -0.62/**-0.75**/0.19/0.39/-0.03/-0.54 | **-0.6****4**/**-0.6****72**/-0.092/0.264/0.09/-0.54 |
| CALIPSO total AOD | -0.22/-0.59/0.34/**0.77**/-0.06/**-0.77** | -0.22/-0.50/0.44/0.39/-0.16/**-0.75** |
| MODIS total AOD | -0.45/0.33/0.57/0.15/0.27/**-0.76** | -0.44/0.16/0.46/0.36/0.38/**-0.81** |
| MISR total AOD | -0.51/0.30/0.29/-0.20/0.23/**-0.66** | -0.46/0.15/0.29/0.05/0.23/**-0.74** |
| MISR fine mode AOD | -0.39/0.31/0.45/-0.12/0.26/**-0.66** | -0.38/0.15/0.44/0.21/0.25/**-0.74** |
| MISR coarse mode AOD | -0.57/0.30/0.13/-0.28/0.14/-0.62 | -0.50/0.19/0.14/-0.25/0.14/**-0.69** |
| | CLARA all-sky LWP | MODIS all-sky LWP |
| GFED  emissions | -0.547/-0.340/-0.095/0.52/0.04/**-0.72** | -0.49/-0.17/-0.02/0.09/-0.02/**-0.77** |
| CALIPSO total AOD | -0.16/-0.31/0.55/0.41/-0.31/**-0.69** | -0.15/-0.19/0.57/-0.56/-0.35/**-0.71** |
| MODIS total AOD | -0.45/-0.13/0.52/-0.06/-0.21/**-0.75** | -0.41/-0.34/0.29/-0.02/-0.37/**-0.84** |
| MISR total AOD | -0.36/-0.25/0.38/-0.31/-0.24/**-0.66** | -0.27/-0.49/0.28/0.08/-0.48/**-0.81** |
| MISR fine mode AOD | -0.17/-0.16/0.52/-0.22/-0.27/**-0.70** | -0.11/-0.36/0.37/0.15/-0.50/**-0.84** |
| MISR coarse mode AOD | -0.55/-0.29/0.22/-0.43/-0.27/-0.55 | -0.47/-0.55/0.16/-0.12/-0.49/**-0.72** |
| | CLARA CTH | MODIS CTH |
| GFED  emissions | 0.022/0.22/-0.210/-0.16/-0.30/-0.03 | -0.206/0.27/-0.04/-0.34/-0.44/**-0.66** |
| CALIPSO total AOD | 0.44/0.62/0.21/-0.40/-0.01/0.34 | 0.20/0.25/0.13/-0.63/0.10/-0.34 |
| MODIS total AOD | 0.08/-0.04/-0.32/-0.34/-0.53/-0.14 | -0.44/0.11/0.09/-0.39/-0.49/**-0.74** |
| MISR total AOD | 0.30/-0.29/0.04/-0.13/-0.43/-0.27 | -0.14/-0.50/0.59/-0.06/-0.36/**-0.73** |
| MISR fine mode AOD | 0.19/-0.19/-0.10/-0.25/-0.46/-0.30 | -0.12/-0.33/0.41/-0.13/-0.39/**-0.78** |
| MISR coarse mode AOD | 0.45/-0.35/0.19/0.16/-0.35/-0.21 | -0.07/-0.61/**0.73**/0.11/-0.26/-0.60 |
| | GPCP Precipitation | |
| GFED emissions | -0.12/-0.19/0.48/-0.53/-0.27/-0.62 | |
| CALIPSO total AOD | -0.03/0.35/0.50/-0.59/-0.44/-0.25 | |
| MODIS total AOD | -0.22/0.44/0.56/-0.12/**-0.82**/-0.63 | |
| MISR total AOD | 0.10/0.50/0.61/0.20/**-0.66**/**-0.70** | |
| MISR fine mode AOD | 0.19/0.41/0.63/0.09/**-0.68**/**-0.71** | |
| MISR coarse mode AOD | -0.02/0.57/0.49/0.34/-0.55/**-0.65** | |

[Figure]

**Figure S11.** Average spatial distributions and corresponding changes in 500 hPa geopotential height (a and b, respectively) and surface pressure (c and d, respectively) from the Copernicus Atmospheric Monitoring Service (CAMS) reanalysis over a wide area of southeast Asia, centered around the southern China study region. Averages were computed from monthly values during the period 2006–2015. Pixel–level changes were computed based on linear regressions fits to the deseasonalized monthly time series.